# Adapt via Bayesian Nonparametric Clustering: Fine-Grained Classification for Model Recycling Under Domain and Category Shift

**Zeya Wang**[*]                                                                      *zeya.wang@uky.edu*
*Dr. Bing Zhang Department of Statistics*
*University of Kentucky*

**Longwen Shang**                                                                   *longwenshang@uvic.ca*
*Department of Mathematics and Statistics*
*University of Victoria*

**Yang Ni**                                                                 *yang.ni@austin.utexas.edu*
*Department of Statistics and Data Sciences*
*The University of Texas at Austin*

**Chenglong Ye**                                                                    *chenglong.ye@uky.edu*
*Dr. Bing Zhang Department of Statistics*
*University of Kentucky*

**Reviewed on OpenReview:** *https://openreview.net/forum?id=J5B4yt7C37*

## Abstract

Recycling pretrained classification models for new domains, known as Source-Free Domain Adaptation (SFDA), has been extensively studied under the closed-set assumption that source and target domains share identical label spaces. However, this assumption does not hold when unseen classes appear in the target domain. Addressing this category shift is challenging, as unknown target classes usually arise with no prior knowledge of their identities or number, and becomes particularly difficult in the source-free setting, where access to source data is unavailable. Most existing methods treat all unknown classes as a single group during both training and evaluation, limiting their capacity to model the underlying structure within the unknown class space. In this work, we present Adapt via Bayesian Nonparametric Clustering (ABC), a novel framework designed for SFDA scenarios where unknown target classes are present. Unlike prior methods, ABC explicitly achieves fine-grained classification of unknown target classes, offering a more structured vision of the problem. Our method first identifies high-confidence target samples likely to belong to known source classes. Using these as guidance, we develop a guided Bayesian nonparametric clustering approach that learns distinct prototypes for both known and unknown classes without requiring the number of unknown classes *a priori*, and assigns target samples accordingly. We further introduce a training objective that refines the source model by encouraging prototype-based discriminability and local prediction consistency. Experiments show that our method achieves competitive performance on standard benchmarks while simultaneously providing effective clustering of unknown classes.

---

[*]Corresponding author. Address: 725 Rose Street, Lexington, KY 40536, United States. Email: `zeya.wang@uky.edu`.

# 1 Introduction

Deep learning models usually require large labeled datasets, but due to costly annotation (Ganin et al., 2016), labeled data in a source domain is often used to train models for unlabeled data in target domains. However, due to domain shift (Torralba & Efros, 2011), these models often fail to generalize, motivating the development of Unsupervised Domain Adaptation (UDA) techniques to bridge the gap. Traditional UDA methods typically assume no category shift, meaning the label sets of the source and target domains, denoted as $\mathcal{C}_s$ and $\mathcal{C}_t$, are identical, i.e., $\mathcal{C}_s = \mathcal{C}_t$, a setting known as closed-set domain adaptation. With the progress of UDA, increasing attention has been paid to the more realistic category shift scenarios that relax this assumption, allowing $\mathcal{C}_s \not\supseteq \mathcal{C}_t$, and thus permit the presence of unknown target classes. While many methods have been proposed to handle different category shift settings, most still follow traditional UDA approaches that require access to source data. This requirement limits the practicality of UDA methods due to factors such as privacy concerns, bandwidth constraints in distributed systems, and the lack of access to source data, for example when it is sensitive, too large to transmit, or lost due to corruption (Kundu et al., 2020). In such cases, it is desirable to recycle the source model without access to the original source data. These challenges have motivated the development of source-free domain adaptation (SFDA) methods, which perform UDA without the source data. Despite their potential, existing SFDA research has primarily focused on the closed-set setting. More realistic scenarios with label mismatches, where unknown target classes may be present (as illustrated in Figure 1), remain underexplored due to their increased complexity and ambiguity, with only a few works addressing these challenges (Liang et al., 2021; Qu et al., 2023; Liang et al., 2020).

While these prior works have demonstrated effective performance in classifying known classes and effectively detecting samples from unknown classes, they typically frame the problem within a standard classification setting, treating all unknown classes as a single unified "unknown" category. However, collapsing all unknown classes into one label risks overlooking the latent structure within them, potentially hindering transfer learning performance. Furthermore, with the growing interest and development in deep clustering (Min et al., 2018), where unlabeled samples are automatically grouped based on their learned embeddings, there is increasing practical value in partitioning samples from unknown classes into finer-grained clusters. This not only enhances interpretability, but also supports downstream tasks that benefit from more granular classification. For example, in medical imaging, it may reveal previously unobserved patient subpopulations when adapting a diagnosis model to data from a new cohort; in robotics, it could help systems distinguish novel sensory patterns in unfamiliar settings; and in security, it can facilitate identifying and separating unseen attacks or anomalous behaviors when deploying a recognition model to a new environment, enabling more effective handling of different unknown patterns.

A promising strategy is to integrate clustering into the adaptation process, enabling the model to simultaneously classify known classes and cluster unknown samples (i.e., samples from unknown classes) into distinct latent groups. This joint task presents additional challenges: not only is the identity of unknown classes unavailable during adaptation, but the number of unknown classes is also unknown. The number of ways to cluster $n$ samples into $k$ non-empty subsets is given by the Stirling number of the second kind $S(n, k)$, whereas allowing an unknown number of subsets yields the Bell number $B_n = \sum_{k=0}^{n} S(n, k)$, which grows much more rapidly with $n$. Traditional methods that tune the number of clusters $K$ (e.g., using internal measures such as the Silhouette score (Rousseeuw, 1987) or Elbow methods (Thorndike, 1953)) often require repeatedly refitting the clustering model across a set of candidate $K$ values, which can introduce non-negligible computational overhead and make integration into the SFDA pipeline less straightforward. More critically, such tuning procedures require defining a search space for the number of clusters $K$, e.g., for a grid search, which relies heavily on prior knowledge of the true $K$ and whose range depends on the dataset scale. For instance, target-domain datasets with many classes demand much broader search ranges than those with only a few, making grid design not very practical for real-world scenarios where prior knowledge of the number of target-domain classes is usually unavailable.

In this paper, we propose a novel algorithm to *Adapt via Bayesian Nonparametric Clustering (ABC)*, which simultaneously classifies known-class samples and clusters unknown ones, while recycling a source model when unknown classes arise in the new domain. ABC first identifies target samples with high confidence of belonging to known classes. Using these detected samples as anchors, a guided Bayesian nonparametric clustering (BNC)

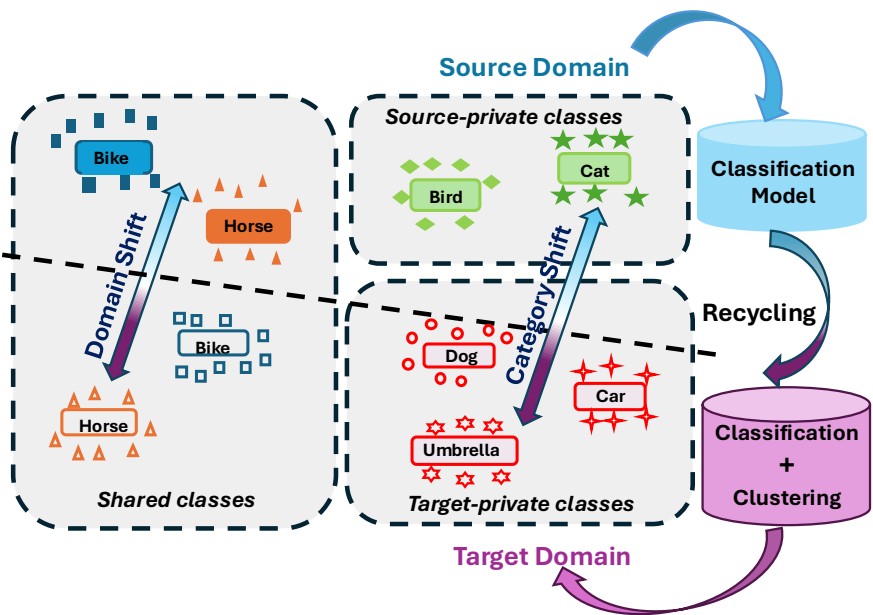

Figure 1: An illustration of the SFDA problem with unknown target classes (with both source- and target-private classes displayed) and the goal of our method.

model is introduced to assign the remaining samples either to existing known classes or to newly formed unknown classes, while simultaneously learning prototypes for all classes, both known and unknown. We further incorporate a prototype regularizer, which aligns samples with their assigned prototypes, and a local consistency regularizer, which promotes prediction agreement within local neighborhoods, together facilitating iterative refinement of the source model for adaptation to target samples. In our model, the BNC component automatically learns the number of unknown classes, allowing their structure to emerge *a posteriori* and eliminating the need for costly grid searches to determine $K$. We conduct extensive experiments and analyses, and demonstrate its effectiveness and solid performance compared to competitive baselines.

## 2 Related Work

**Unsupervised Domain Adaptation (UDA):** UDA methods improve the generalization ability of deep neural networks across different domains and mitigate the performance degradation caused by domain shift. Among the most widely adopted approaches are discrepancy minimization methods(Tzeng et al., 2014; Long et al., 2015; Sun et al., 2016; Long et al., 2016; 2017; Chen et al., 2019) and adversarial training techniques (Ganin & Lempitsky, 2015; Ganin et al., 2016; Tzeng et al., 2017; Zhang et al., 2018; Wang et al., 2020). To reduce domain shift, discrepancy minimization methods introduce a divergence measure between data distributions across domains and minimize it, while adversarial training, inspired by generative adversarial networks (GANs)(Goodfellow et al., 2014), leverages a domain discriminator to classify samples as originating from the source or target domain, thereby encouraging the learning of domain-invariant features (Ganin & Lempitsky, 2015; Ganin et al., 2016). More recent research has begun to address the UDA problem under category shift. Research has examined partial(Cao et al., 2018a;b; Wang et al., 2020; Li et al., 2021), open-set(Liu et al., 2022; Panareda Busto & Gall, 2017; Saito et al., 2018; Jang et al., 2022; Li et al., 2021), and open-partial domain adaptation scenarios(You et al., 2019; Li et al., 2021). Furthermore, there is a growing interest in developing general methods that can handle different settings (Chen et al., 2022; Saito et al., 2020; Li et al., 2021). Many of these approaches detect samples from unknown classes by identifying uncertain instances based on indicators derived from prediction outputs, such as entropy values (You et al., 2019; Jang et al., 2022). Other approaches leverage clustering-based strategies to identify shared classes. For example, Domain Consensus Clustering (Li et al., 2021) clusters samples in both source and target

domains and then evaluates semantic consensus among cluster centers to identify shared classes. Global and Local Clustering (Qu et al., 2023) performs one-vs-all clustering by comparing target samples to source-class prototypes and negative prototypes obtained via clustering, assigning known-class labels based on higher similarity to source prototypes. While these methods provide flexible clustering-based frameworks that aid identifying shared classes, most still rely on standard clustering algorithms like KMeans (MacQueen et al., 1967) and require repeated refitting across candidate $K$ values.

**Source-free Domain Adaptation (SFDA):**  SFDA focuses on adapting a model pre-trained on the source domain to a target domain without requiring access to the source data. Many methods have been proposed to address this problem, often concentrating on conventional settings where no category shift is assumed between source and target domains (Liang et al., 2020; Qu et al., 2022; Yang et al., 2021b; Ding et al., 2022; Yang et al., 2022). Some approaches utilize the source model to extract anchor features and then assign pseudo-labels based on these anchors (Ding et al., 2022), while others adapt the source model by promoting consistency among local neighborhood features in the target domain (Yang et al., 2022). Only a few works have tackled SFDA under category shift conditions, where target classes may not completely overlap with source classes (Qu et al., 2023; Kundu et al., 2020; Liang et al., 2021). Although these methods represent significant progress, they often treat all unknown classes uniformly in both modeling and evaluation, and seldom explicitly explore the clustering behavior of samples from unknown classes in the target domain.

**Bayesian Nonparametric Clustering (BNC):**  Clustering aims to group similar observations into a fixed number of clusters. Common methods like K-Means and finite mixture models require specifying the number of clusters, $K$, in advance (JAIN et al., 1999; Reynolds, 2015). While internal validation approaches can help tune this parameter by evaluating internal metrics across different values of $K$, including in deep clustering settings (Wang & Ye, 2026), they often require repeated model refitting, selection of evaluation criteria, and careful design of a search range. BNC addresses this limitation by inferring the number of clusters as part of the model (Hjort et al., 2010), allowing clustering without predefining the number of clusters. BNC encompasses a flexible family of mixture models, such as Dirichlet Process Mixtures (Ferguson, 1973; Antoniak, 1974), Pitman–Yor Process Mixtures (Pitman & Yor, 1997), and Normalized Generalized Gamma Process Mixtures (Lijoi et al., 2007), which can approximate complex data distributions. Advances in scalable inference techniques, such as variational inference (Blei & Jordan, 2006; Ni et al., 2020b) and consensus Monte Carlo (Ni et al., 2020c;a), have enabled BNC to handle large datasets, making it applicable in large-scale scenarios. Recent advances have integrated BNC with deep neural networks, enabling automatic discovery of the number of clusters when clustering high-dimensional data(Wang et al., 2021; Ronen et al., 2022). Together, these developments make BNC accessible to a wide range of large-scale and high-dimensional data settings. Compared to conventional clustering strategies in SFDA methods such as Domain Consensus Clustering and Global and Local Clustering, BNC offers a clear advantage: it can directly estimate the number of unknown target-domain classes while clustering, eliminating the need for repeated model refitting or manual grid searches over $K$. This allows the unknown class structure to emerge *a posteriori*, reduces computational overhead, facilitates seamless integration into SFDA pipelines, and makes the approach more robust to datasets where the number of unknown classes varies widely. Unlike other clustering-based approaches, which primarily use clustering to assist in identifying shared classes, our method leverages BNC to directly discover individual unknown classes, supporting more precise and fine-grained adaptation.

## 3  Methodology

In SFDA, neither source data $\mathcal{X}_s$, source labels $\mathcal{Y}_s$, nor target labels $\mathcal{Y}_t$ are accessible during adaptation. To address scenarios with unknown classes, we consider the setting where the source and target class sets $\mathcal{C}_s$ and $\mathcal{C}_t$ satisfy $\mathcal{C}_s \cap \mathcal{C}_t \neq \emptyset$, and $\mathcal{C}_t \setminus \mathcal{C}_s \neq \emptyset$. For clarity, we define $\mathcal{C} = \mathcal{C}_s \cap \mathcal{C}_t$ as the *common label space*, $\mathcal{C}_s^c = \mathcal{C}_s \setminus \mathcal{C}_t$ as the *source-private label space*, and $\mathcal{C}_t^c = \mathcal{C}_t \setminus \mathcal{C}_s$ as the *target-private label space*. We will refer to the classes within $\mathcal{C}_s$ as known classes and those within $\mathcal{C}_t^c$ as unknown classes. While prior work typically treats classes in $\mathcal{C}_t^c$ as a single category $c_{\text{unk}}$, we aim to distinguish these classes individually, denoting the set of discovered unknown classes by $\hat{\mathcal{C}}_t^c$. Our learning objective is twofold: to classify target instances as the estimated label $\hat{y}_i^t \in \mathcal{C}$ if they belong to known classes, and to assign them to discovered groups $\hat{\mathcal{C}}_t^c$ if they

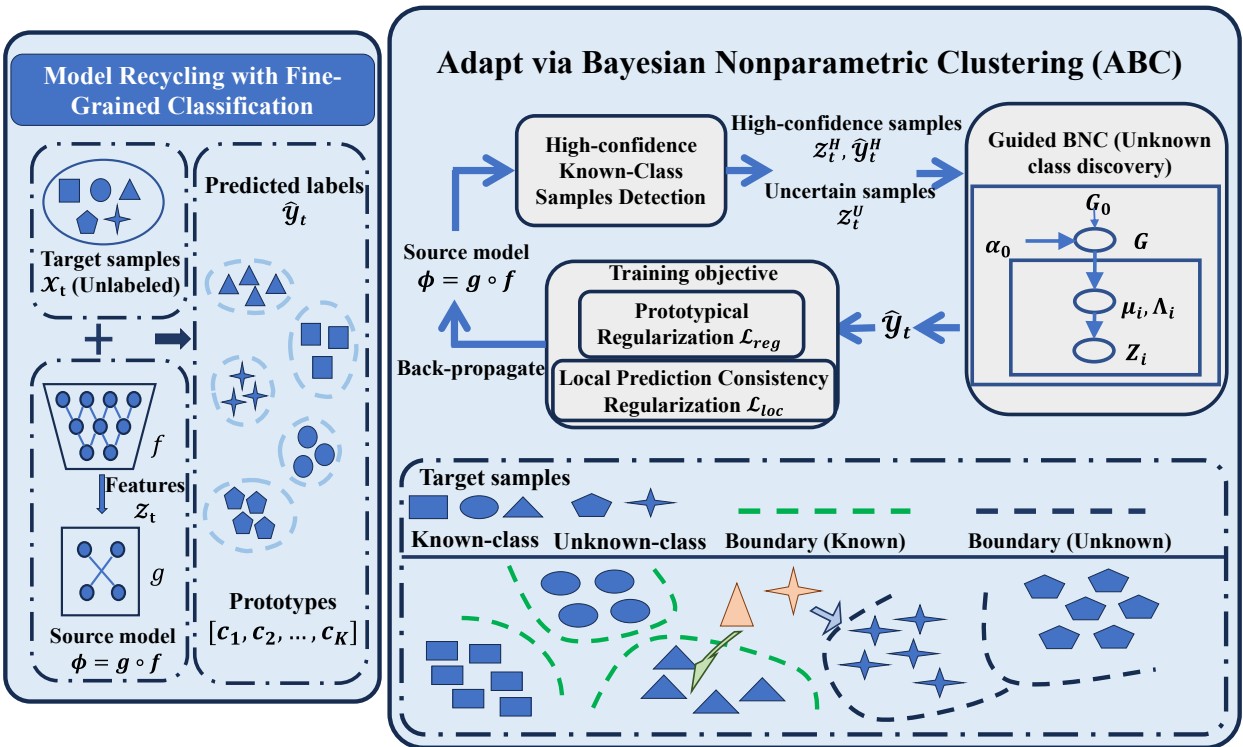

Figure 2: An overview of Adapt via Bayesian Nonparametric Clustering (ABC).

belong to unknown classes. The overall workflow of the proposed algorithm is illustrated in Figure 2, with detailed steps described in the following sections.

## 3.1 High-Confidence Common Class Samples

We first focus on detecting samples most likely to belong to the common label set $\mathcal{C}$. Let the source model be denoted as $\phi = g \circ f$, where $f$ represents the feature extractor mapping $x_i \in \mathcal{X}_s$ to a feature representation $z_i \in \mathbb{R}^m$, and $g$ is the classifier that maps $z_i$ to source classes $\mathcal{C}_s = \{1, 2, \cdots, K_s\}$, where $K_s$ represents the number of source classes. Let $\mathsf{w}^g \in \mathbb{R}^{m \times K_s}$ denote the weight matrix of the classifier $g$, where each column $\mathsf{w}_k^g \in \mathbb{R}^m$ corresponds to the weight vector associated with source class $k$. Each $\mathsf{w}_k^g$ can be interpreted as an anchor representing the $k$-th source class in the feature space, and has been leveraged to initialize source cluster centers for spherical k-means clustering in previous SFDA methods, where target instances are assigned to source classes by maximizing cosine similarity to these centers (Ding et al., 2022). The cosine similarity between a target feature $z_i$ and each source class anchor $\mathsf{w}_k^g$, defined as $cos(z_i, w_k^g) = \frac{z_i^\top \mathsf{w}_k^g}{\|z_i\|\|\mathsf{w}_k^g\|}$, measures proximity to known class $k$. Consequently, the maximum similarity over all source classes provides an interpretable measure of the sample's affiliation with the known classes. A low maximum similarity indicates the sample may not belong to any known class.

The source model's output is often used to initialize the detection and classification of known-class samples and can provide informative indicators, denoted as $\kappa$, for identifying known-class versus unknown-class samples (Jang et al., 2022; Qu et al., 2023). Building on the discussion above, we define $\kappa_i = 1 - \max_k cos(z_i, w_k^g)$ and model its marginal distribution using a mixture of two Beta distributions, a choice that is both natural and effective in many prior UDA work targeting unknown classes.

$$p(\kappa) = \pi_{\text{known}} p(\kappa \mid y \in \mathcal{C}_s) + \pi_{\text{unknown}} p(\kappa \mid y \notin \mathcal{C}_s) \tag{1}$$

where $\pi_{\text{known}}$ and $\pi_{\text{unknown}}$ represent the probabilities $P(y \in \mathcal{C}_s)$ and $P(y \notin \mathcal{C}_s)$, respectively. Given this Beta mixture model, we can estimate the posterior probability $P(y \in \mathcal{C}_s \mid \kappa)$, which represents the confidence that a sample belongs to the source classes. The parameters of the Beta distributions, along with $\pi_{\text{known}}$ and $\pi_{\text{unknown}}$, are estimated by fitting the Beta mixture model using the Expectation-Maximization (EM) algorithm (Arazo et al., 2019; Jang et al., 2022). Once the parameters are obtained, the posterior probability $P(y \in \mathcal{C}_s \mid \kappa)$ is given by: $P(y \in \mathcal{C}_s \mid \kappa) = \pi_{\text{known}} p(\kappa \mid y \in \mathcal{C}_s) / [\pi_{\text{known}} p(\kappa \mid y \in \mathcal{C}_s) + \pi_{\text{unknown}} p(\kappa \mid y \notin \mathcal{C}_s)]$. The decision boundary $P(y \in \mathcal{C}_s \mid \kappa) > P(y \notin \mathcal{C}_s \mid \kappa)$, which indicates that the posterior probability of belonging to the known classes exceeds that of the unknown classes, corresponds to $P(y \in \mathcal{C}_s \mid \kappa) > 0.5$. Consequently, we can identify high-confidence common-class samples by applying a threshold $P(y \in \mathcal{C}_s \mid \kappa) > \delta$ with $\delta = 0.5$ as the default. The threshold $\delta$ can be slightly adjusted around 0.5 using an auto-tuning strategy, and the details of this implementation will be discussed in Section 4.3. For samples that meet this criterion, we assign their predicted label to the class with the highest cosine similarity, computed as $\arg\max_k cos(z_i, w_k^g)$. We denote the identified *high-confidence samples* and their estimated labels as $\mathcal{X}_t^H$ and $\hat{\mathcal{Y}}_t^H$, respectively. Based on $\hat{\mathcal{Y}}_t^H$, we can derive an estimate of the common label set $\mathcal{C}$, denoted as $\hat{\mathcal{C}}$. The remaining target samples, for which the source-class confidence is insufficient, are considered as *uncertain samples* and denoted by $\mathcal{X}_t^U$. We use $\mathcal{Z}_t^H$ and $\mathcal{Z}_t^U$ to denote the embeddings extracted by the encoder corresponding to $\mathcal{X}_t^H$ and $\mathcal{X}_t^U$, respectively.

## 3.2 Guided Bayesian Nonparametric Clustering

Uncertain samples may include both instances outside the common label set (i.e., from target-private label space) and those within it that received low-confidence predictions. High-confidence common-class samples can serve as anchors or references to guide the clustering of uncertain samples, making the problem more aligned with weakly-supervised clustering. Furthermore, clustering samples from unknown classes requires a method that does not rely on a predefined number of clusters, as this information is usually not available. To this end, we propose a guided Bayesian nonparametric clustering (GBNC) framework that automatically discovers previously unseen classes while simultaneously classifies samples into known classes.

Given the embeddings $\mathcal{Z}_t^U = \{z_1^U, \ldots, z_{n_U}^U\}$ of uncertain samples with size $n_U$, we consider leveraging Bayesian nonparametric clustering (BNC) technique to estimate the corresponding cluster labels $\mathcal{Y}_t^U = \{y_1^U, \ldots, y_{n_U}^U\}$, as it can both handle unknown class discovery and incorporate weak supervision effectively. Specifically, we build our model on the Dirichlet Process Mixture (DPM) framework, a widely used Bayesian nonparametric approach known for its simplicity of implementation and extensive literature support. The DPM-based formulation naturally accommodates a flexible and potentially unbounded number of clusters, allowing the model to adjust automatically to the number of clusters as the data grows. Our method can also be extended to other Bayesian nonparametric mixture models, such as Pitman–Yor process mixtures and Normalized Generalized Gamma process mixtures, which, like the DPM, allow the number of clusters to emerge from the data. The Dirichlet Process (DP) is a stochastic process that defines a distribution over distributions, meaning that each draw from it is itself a probability distribution. A key property of the DP is its almost surely discrete realizations, which naturally induces clustering by allowing parameter sharing among individual data points and eliminates the need to specify the number of clusters in advance. A DPM constructed from the observed data $\mathcal{Z}_t^U$ can be formulated within a Bayesian hierarchical framework,

$$z_i^U | \eta_i \sim p(z_i^U | \eta_i), \quad \eta_i | G \sim G, \quad G \sim DP(\alpha, G_0). \tag{2}$$

$p(\cdot \mid \eta_i)$ denotes the likelihood function, following a probability distribution parameterized by $\eta_i$, where $\eta_i$ is drawn from a DP-distributed prior $G$. The DP prior is characterized by the concentration parameter $\alpha$ and the base distribution $G_0$, which specifies a prior probability distribution over the parameters. The base distribution $G_0$ is typically chosen to be conjugate to the likelihood to make the inference tractable and thus simplify computation. Due to the discrete nature of $G$, multiple $\eta_i$ associated with different observations may take the same value, naturally forming clusters. Cluster assignments arise from these shared values, meaning that $y_i^U = y_j^U$ when $\eta_i = \eta_j$. The concentration parameter $\alpha$ is directly related to the prior expected number of clusters, which can be set in a noninformative manner, as discussed in our Section 4.3.

We denote the labels generated in Section 3.1 for the high-confidence samples as $\hat{\mathcal{Y}}_t^H = \{\hat{y}_1^H, \ldots, \hat{y}_{n_H}^H\}$, where $n_H$ is their number. These labels are assumed to be reliably assigned to known classes and are therefore treated as fixed. Accordingly, for the embeddings $\mathcal{Z}_t^H = \{z_1^H, \ldots, z_{n_H}^H\}$, we can directly model their distribution given these fixed labels as $z_i^H \sim p(z_i^H | \eta_{\hat{y}_i^H}^\star)$, where $\eta_{\hat{y}_i^H}^\star$ denotes the parameters associated with the cluster corresponding to the predicted label $\hat{y}_i^H$. Let $|\hat{\mathcal{C}}|$ denote the number of estimated common classes, i.e., the number of unique classes in $\hat{\mathcal{Y}}_t^H$. To address the potential mismatch in cluster indices between $\hat{\mathcal{Y}}_t^H$ and $\mathcal{Y}_t^U$, we assume that all $\hat{y}_i^H$ take values from the index set $\{1, 2, \ldots, |\hat{\mathcal{C}}|\}$. In our experiments, we include an additional step to align and remap the estimated labels $\hat{\mathcal{Y}}_t^H$ to this index range, facilitating the subsequent prototype-based regularization. Further details on this step are provided in Appendix B.

By marginalizing out $G$ in Equation 2, the hierarchical model can be reformulated as an infinite mixture model, as in Equation A2, also known as the stick-breaking representation (with more details in Appendix A). We then integrate this representation with the distribution of $z_i^H$ to yield:

$$z_i^U | y_i^U = k \sim p(z_i^U | \eta_k^\star), \quad z_i^H \sim p(z_i^H | \eta_{\hat{y}_i^H}^\star), \quad \eta_k^\star \sim G_0$$

$$y_i^U | \pi \sim Cat(\boldsymbol{\pi}), \quad \boldsymbol{\pi} = (\pi_1, \pi_2, \cdots, \pi_{|\hat{\mathcal{C}}|}, \cdots), \quad \pi_k = v_k \prod_{j=1}^{k-1}(1 - v_j), \quad v_k \sim Beta(1, \alpha) \tag{3}$$

where $\boldsymbol{\pi}$ is a probability vector over clusters for $k = 1, 2, \ldots$. The categorical distribution $Cat(\boldsymbol{\pi})$ assigns cluster memberships to the uncertain samples with probabilities $P(y_i^U = k) = \pi_k$. $Beta(\cdot)$ denotes the beta distribution.

We model the mixture components using a multivariate Gaussian distribution, $p(\cdot | \eta_k^\star) = N(\cdot | \mu_k, \Lambda_k)$, where $\Lambda_k$ is the inverse covariance matrix (i.e., precision matrix). While deep embeddings may deviate from ideal normality, Gaussian mixture models are flexible, can approximate a wide range of distributions in practice, and have been widely used in deep clustering literature (Wang & Jiang, 2018; Wang et al., 2021; Ronen et al., 2022). We adopt a conjugate Normal-Wishart prior, $G_0 = N(\mu_k | \zeta, \tau \Lambda_k) \times W(\Lambda_k | b, \Omega)$, where $\tau$ is a scaling factor, $b$ is the degrees of freedom, and $\Omega$ is the scale matrix. The overall model is therefore parameterized by $\{v_k, \mu_k, \Lambda_k\}_{k=1}^\infty$ and latent cluster labels $\{y_i^U\}_{i=1}^{n_U}$. The constructive stick-breaking representation allows us to derive a tractable joint distribution for posterior estimation. For computational efficiency, we choose Variational Bayes (VB) to approximate the posterior of these parameters. The joint probability distribution of all random variables can be factorized as follows, where boldface symbols denote vectorized forms of the corresponding variables in Equation 4.

$$p(\mathbf{Z}^U, \mathbf{Z}^H, \boldsymbol{Y}^U, \boldsymbol{v}, \boldsymbol{\mu}, \boldsymbol{\Lambda}) = p(\mathbf{Z}^U | \boldsymbol{Y}^U, \boldsymbol{\mu}, \boldsymbol{\Lambda}) p(\mathbf{Z}^H | \boldsymbol{\mu}, \boldsymbol{\Lambda}) p(\boldsymbol{Y}^U | \boldsymbol{v}) p(\boldsymbol{v}) p(\boldsymbol{\mu} | \boldsymbol{\Lambda}) p(\boldsymbol{\Lambda}) \tag{4}$$

The goal of VB is to find a variational distribution that minimizes the Kullback–Leibler (KL) divergence to the true posterior. We assume a fully factorized variational family, as in Equation 5, which allows us to apply mean-field approximation to yield tractable updates in our VB implementation (Bishop & Nasrabadi, 2006).

$$q(\{v_k\}_{k=1}^{K^*-1}, \{\mu_k\}_{k=1}^{K^*}, \{\Lambda_k\}_{k=1}^{K^*}, \{y_i^U\}_{i=1}^{n_U}) = \prod_{k=1}^{K^*-1} q(v_k) \prod_{k=1}^{K^*} q(\mu_k, \Lambda_k) \prod_{i=1}^{n_U} q(y_i^U), \tag{5}$$

where $q(v_k)$, $q(\mu_k, \Lambda_k)$, and $q(y_i^U)$ denote the variational distributions for the corresponding variables and form the factors of the factorized variational distribution. This structure enables efficient optimization via coordinate ascent. Following the standard approach of applying a truncated stick-breaking representation in VB for DPMs (Blei & Jordan, 2006), we truncate the stick-breaking process at level $K^*$ such that $q(v_{K^*} = 1) = 1$. The value $K^*$ serves as a practical upper bound, fundamentally different from the number of clusters $K$ in tuning-based methods which is repeatedly adjusted, and consistent results can be expected with any sufficiently large $K^*$ (e.g., 100). In our model, the variational distributions are specified as: $q(v_k) = Beta(v_k | \gamma_{k1}, \gamma_{k2})$, $q(\mu_k, \Lambda_k) = N(\mu_k | m_k, \tau_k \Lambda_k) \times W(\Lambda_k | c_k, D_k)$, and $q(y_i^U) = Cat(\phi_i)$, where $\phi_i = (\phi_{i1}, \phi_{i2}, \ldots)$. The variational parameters $\{m_k, \tau_k, c_k, D_k, \gamma_{k1}, \gamma_{k2}\}_k$ and $\{\phi_i\}_i$ are updated via the VB

algorithm. Both the full conditional and variational distributions belong to the exponential family, allowing the variational distributions for the parameters to be derived from their full conditionals (Blei & Jordan, 2006). Building on this, we derive our VB algorithm. Let $\hat{n}_k^H = \sum_{i=1}^{n_H} I(\hat{y}_i^H = k)$. The iterative update steps for each variational parameter in our inference procedure are presented below, with detailed derivations and further implementation details provided in Appendix B. Finally, with the estimated $\phi_i$, each uncertain sample is assigned a predicted label $\hat{y}_i^U = \arg\max_k \phi_{ik}$, forming $\hat{\mathcal{Y}}_t^U = \{\hat{y}_1^U, \ldots, \hat{y}_{n_U}^U\}$.

**Update steps within each iteration:**

1. Compute $(\hat{n}_k, \overline{z}_k^*, S_k^*)$:

$$\hat{n}_k = E_q[n_k] = \sum_{i=1}^{n_U} \phi_{ik}; \quad \overline{z}_k^* = \frac{1}{\hat{n}_k + \hat{n}_k^H}[\sum_{i=1}^{n_U} \phi_{ik} z_i^U + \sum_{i=1}^{n_H} I(\hat{y}_i^H = k) z_i^H]$$

$$S_k^* = \frac{1}{\hat{n}_k + \hat{n}_k^H}[\sum_{i=1}^{n_U} \phi_{ik} \left(z_i^U - \overline{z}_k^*\right)\left(z_i^U - \overline{z}_k^*\right)^{\mathrm{T}} + \sum_{i=1}^{n_H} I(\hat{y}_i^H = k)\left(z_i^H - \overline{z}_k^*\right)\left(z_i^H - \overline{z}_k^*\right)^{\mathrm{T}}]$$

2. Update variational parameters $(\gamma_{k1}, \gamma_{k2})$:

$$\gamma_{k1} = E_q[n_k + 1] = \hat{n}_k + 1, \quad \gamma_{k2} = E_q[\sum_{j>k} n_j + \alpha] = \sum_{j>k} \hat{n}_j + \alpha$$

3. Update variational parameters $(\tau_k, m_k, c_k, D_k)$:

$$\tau_k = \tau + \hat{n}_k + \hat{n}_k^H; \quad m_k = \frac{\tau\zeta + \sum_{i=1}^{n_U} \phi_{ik} z_i^U + \sum_{i=1}^{n_H} z_i^H I(\hat{y}_i^H = k)}{\tau + \hat{n}_k + \hat{n}_k^H}$$

$$c_k = \hat{n}_k + \hat{n}_k^H + b; \quad D_k^{-1} = \Omega^{-1} + (\hat{n}_k + \hat{n}_k^H) S_k^* + \frac{(\hat{n}_k + \hat{n}_k^H)\tau}{\tau + \hat{n}_k + \hat{n}_k^H}(\overline{z}_k^* - \zeta)(\overline{z}_k^* - \zeta)^{\top}$$

4. Update variational parameters $\phi_i = (\phi_{i1}, \phi_{i2}, \ldots)$:

$$\log(\phi_{ik}) \propto \psi(\gamma_{k1}) - \psi(\gamma_{k1} + \gamma_{k2}) + \sum_{j=1}^{k-1}[\psi(\gamma_{j2}) - \psi(\gamma_{j1} + \gamma_{j2})]$$

$$+ \frac{1}{2}\left[\psi_d\left(\frac{c_k}{2}\right) + \log|D_k|\right] - \frac{1}{2}[d\tau_k^{-1} + c_k(z_i^U - m_k)^{\top} D_k(z_i^U - m_k)]$$

where $\psi(\cdot)$ and $\psi_d(\cdot)$ represent the digamma and multivariate digamma functions, respectively.

### 3.3 Optimization and Inference

**Prototype-based Regularization:** To enhance target sample separation, we alternate clustering updates with end-to-end training using pseudo-cluster labels generated in Sections 3.1 and 3.2, optimized under a clustering-based objective. Inspired by prior work on non–source-free settings (Li et al., 2021), we incorporate a prototype-based regularizer that promotes intra-cluster compactness and inter-cluster separation. Specifically, we first compute a centroid $c_k$ for each cluster $k$, used as its prototype, based on all target samples and their predicted labels $\hat{\mathcal{Y}}_t = [\hat{\mathcal{Y}}_t^H, \hat{\mathcal{Y}}_t^U]$, where $\hat{\mathcal{Y}}_t^H$ and $\hat{\mathcal{Y}}_t^U$ are obtained from Sections 3.1 and 3.2, respectively. Let $\mathbf{c} = [c_1, c_2, \ldots, c_K]$ represent the estimated prototypes, computed as the L2-normalized centroids of the obtained target clusters. These prototypes are dynamically updated during each training iteration, and recomputed after each update of $\hat{\mathcal{Y}}_t = \{\hat{y}_1^t, \hat{y}_2^t, \cdots\}$. Before applying the regularizer, we convert each predicted label $\hat{y}_i^t$ into its one-hot representation, where $\hat{y}_{i,k}^t = 1$ if $\hat{y}_i^t = k$. The prototypical regularizer for a minibatch $B$ is then defined as:

$$\mathcal{L}_{\mathrm{reg}} = -\sum_{i \in B} \sum_{k=1}^{K} \hat{y}_{i,k}^t \log \hat{p}_{(i,k)}^t \tag{6}$$

where, $\hat{p}_{(i,k)}^t = \frac{\exp\left((\tilde{z}_i^t)^\top c_k/0.1\right)}{\sum_{k=1}^K \exp\left((\tilde{z}_i^t)^\top c_k/0.1\right)}$, and $\tilde{z}_i^t$ denotes the L2-normalized feature $z_i$ of target sample $i$. To gradually regulate the influence of $\mathcal{L}_{\text{reg}}$ and avoid premature over-expansion, we adopt the ramp-up schedule from (Li et al., 2021), using a dynamic weight $\lambda_{\text{reg}} = e^{-3\cdot(1-\frac{l}{L})}$, where $l$ is the current step and $L$ the total training steps, yielding the term $\lambda_{\text{reg}}\mathcal{L}_{\text{reg}}$ in the training objective.

**Local Prediction Consistency Regularization:**  To further refine the source classifier $g$ and align similar target samples, we introduce a local consistency loss $\mathcal{L}_{\text{loc}}$, which enforces prediction consistency within local neighborhoods.

$$\mathcal{L}_{\text{loc}} = -\sum_{i \in B} \sum_{j \in \mathcal{N}_i} \log(g_i^t)^\top g_j^t \tag{7}$$

$g_i^t$ denotes the softmax prediction of a target sample from the source classifier $g$, representing a probability distribution over source classes. The set $\mathcal{N}_i$ refers to the local neighborhood of target sample $i$, defined as its top $K_{\text{nn}}$ nearest neighbors based on cosine similarity in the feature space. To implement the nearest-neighbor computations during minibatch training, we follow prior work (Yang et al., 2021a; 2022; 2021b) and maintain two memory banks: one storing all target features and the other storing their corresponding softmax predictions, both updated incrementally using minibatch data. During training, $g_j^t$ is retrieved from the memory bank and treated as a fixed reference, so gradients are only backpropagated through $g_i^t$. Similar to methods that align neighboring samples by maximizing the dot product of their predictions (e.g., Yang et al. (2022)), this regularizer maximizes the dot product between a query sample's log prediction and the predictions of its neighbors. This encourages consistency among nearby samples while reducing the tendency toward overly peaked (i.e., one-hot-like) softmax outputs over the source classes, so that high-entropy (less confident) predictions are aligned more softly. Since gradients are only propagated through $g_i^t$, our objective also resembles $\text{KL}(g_j^t \| g_i^t)$, minimizing the KL divergence between a fixed neighbor distribution and the query sample.

**Training Objective and Inference:**  The final training objective integrates two components: the prototypical regularizer $\mathcal{L}_{\text{reg}}$ and the local prediction consistency regularizer $\mathcal{L}_{\text{loc}}$, balanced by their respective weighting coefficients $\lambda_{\text{reg}}$ and $\lambda_{\text{loc}}$. The overall loss function is formulated as: $\mathcal{L} = \lambda_{\text{reg}}\mathcal{L}_{\text{reg}} + \lambda_{\text{loc}}\mathcal{L}_{\text{loc}}$. With this overall objective, the workflow of ABC, comprising the iterative steps detailed in Sections 3.1, 3.2, and 3.3, is outlined in Algorithm 1. During inference, with the final set of prototypes $[c_1, c_2, \ldots, c_K]$, each target sample is assigned the label of the most similar prototype based on cosine similarity between its L2-normalized feature and the normalized prototypes, according to $\hat{y}_i^t = \arg\max_k \langle \tilde{z}_i^t, c_k \rangle$.

---

**Algorithm 1** Adapt via Bayesian Nonparametric Clustering

---

 1: **Input:** Unlabeled target samples $\mathcal{X}_t$; pretrained classification model $\phi = g \circ f$
 2: **Output:** Predicted labels $\hat{y}_1^t, \hat{y}_2^t, \cdots$; prototypes $\{c_1, c_2, \ldots, c_K\}$
 3: **for** $epoch = 1, 2, \ldots, T$ **do**
 4:     a. Identify high-confidence samples $\mathcal{X}_t^H$ with pseudo-labels $\hat{\mathcal{Y}}_t^H$
 5:     b. Identify uncertain samples $\mathcal{X}_t^U$
 6:     c. Estimate labels $\hat{\mathcal{Y}}_t^U$ for $\mathcal{X}_t^U$ via guided Bayesian nonparametric clustering
 7:     d. Generate prototypes $\mathbf{c} = [c_1, \ldots, c_K]$
 8:     **if** $epoch < T$; **for** $iteration = 1, 2, \ldots$ **do**
 9:         Backpropagate $\phi$ w.r.t. the loss $\mathcal{L}$ and update $\mathbf{c}$
10:     **end for**
11: **end for**
12: **Inference:** Assign final labels $\hat{y}_1^t, \hat{y}_2^t, \ldots$

---

Table 1: H-score (%) on Office-Home, Office-31, and VisDA under OPDA settings. 'SF' indicates source-free methods, and 'FG' refers to fine-grained discovery of unknown classes. **Bold blue numbers** highlight top non-SF methods, and **bold red numbers** highlight leading SF methods (excluding the GBNC-only module).

(a) Office-Home

| Methods | SF | FG | A2C | A2P | A2R | C2A | C2P | C2R | P2A | P2C | P2R | R2A | R2C | R2P | Avg |
|---|---|---|---|---|---|---|---|---|---|---|---|---|---|---|---|
| UAN (You et al., 2019) | ✗ | ✗ | 51.6 | 51.7 | 54.3 | 61.7 | 57.6 | 61.9 | 50.4 | 47.6 | 61.5 | 62.9 | 52.6 | 65.2 | 56.6 |
| CMU (Fu et al., 2020) | ✗ | ✗ | 56.0 | 56.9 | 59.2 | 67.0 | 64.3 | 67.8 | 54.7 | 51.1 | 66.4 | 68.2 | 57.9 | 69.7 | 61.6 |
| DCC (Li et al., 2021) | ✗ | ✗ | 58.0 | 54.1 | 58.0 | **74.6** | 70.6 | 77.5 | 64.3 | **73.6** | 74.9 | **81.0** | **75.1** | 80.4 | 70.2 |
| OVANet (Saito & Saenko, 2021) | ✗ | ✗ | 62.8 | 75.6 | 78.6 | 70.7 | 68.8 | 75.0 | 71.3 | 58.6 | 80.5 | 76.1 | 64.1 | 78.9 | 71.8 |
| GATE (Chen et al., 2022) | ✗ | ✗ | **63.8** | **75.9** | **81.4** | 74.0 | **72.1** | **79.8** | **74.7** | 70.3 | **82.7** | 79.1 | 71.5 | **81.7** | **75.6** |
| **Best Epoch (Best H-score)** | | | | | | | | | | | | | | | |
| SHOT-O (Liang et al., 2020) | ✓ | ✗ | 43.2 ±0.8 | 56.1 ±0.2 | 57.4 ±1.7 | 57.5 ±0.8 | 54.5 ±1.6 | 60.3 ±0.9 | 56.7 ±1.2 | 45.7 ±0.8 | 62.2 ±0.2 | 57.6 ±0.5 | 44.4 ±0.4 | 58.5 ±0.2 | 54.5 |
| GLC (Qu et al., 2023) | ✓ | ✗ | 53.8 ±0.5 | 78.0 ±0.6 | **89.6** ±0.2 | 64.5 ±0.2 | 74.7 ±3.4 | **89.0** ±0.4 | **75.8** ±0.8 | 54.3 ±0.6 | **88.2** ±0.1 | 78.8 ±0.7 | 52.4 ±0.4 | **84.3** ±0.0 | 73.6 |
| ABC | ✓ | ✓ | **56.7** ±0.9 | **80.6** ±1.0 | 84.5 ±0.4 | 71.4 ±0.7 | **80.3** ±1.3 | 82.9 ±0.2 | 71.2 ±1.1 | **55.1** ±0.8 | 83.8 ±0.4 | 71.1 ±0.9 | **56.9** ±0.3 | 79.8 ±0.8 | 72.9 |
| **Final Epoch** | | | | | | | | | | | | | | | |
| SHOT-O (Liang et al., 2020) | ✓ | ✗ | 31.9 ±0.7 | 30.8 ±0.9 | 39.6 ±1.9 | 49.8 ±0.8 | 29.1 ±0.8 | 38.3 ±1.2 | 50.8 ±1.3 | 34.9 ±0.3 | 43.0 ±0.8 | 51.6 ±1.9 | 32.6 ±0.4 | 35.9 ±0.9 | 39.0 |
| GLC (Qu et al., 2023) | ✓ | ✗ | 10.6 ±0.5 | 77.5 ±1.0 | 88.6 ±0.5 | 16.8 ±1.0 | 74.6 ±3.5 | 88.3 ±0.6 | 74.8 ±0.3 | 13.4 ±0.5 | 87.8 ±0.5 | 76.1 ±0.9 | 14.5 ±0.6 | 83.5 ±0.0 | 58.9 |
| GBNC-only | - | - | 52.1 ±0.7 | 74.9 ±1.4 | 82.7 ±0.7 | 61.9 ±3.1 | 66.2 ±2.7 | 78.0 ±1.6 | 61.0 ±1.8 | 50.9 ±2.0 | 80.1 ±1.1 | 62.8 ±0.5 | 53.3 ±0.6 | 77.5 ±0.8 | 66.8 |
| ABC | ✓ | ✓ | **54.5** ±0.4 | 77.1 ±2.2 | 82.1 ±0.8 | **70.3** ±1.9 | 72.9 ±3.0 | 81.3 ±0.4 | 69.7 ±0.7 | **53.6** ±1.0 | 83.2 ±0.8 | 69.1 ±0.3 | **55.1** ±2.2 | 75.5 ±2.8 | **70.4** |

(b) Office-31 and VisDA

| Methods | SF | FG | VisDA | Office-31 | | | | | | |
|---|---|---|---|---|---|---|---|---|---|---|
| | | | S2R | A2D | A2W | D2A | D2W | W2A | W2D | Avg |
| UAN (You et al., 2019) | ✗ | ✗ | 34.8 | 59.7 | 58.6 | 60.1 | 70.6 | 60.3 | 71.4 | 63.5 |
| CMU (Fu et al., 2020) | ✗ | ✗ | 32.9 | 68.1 | 67.3 | 71.4 | 79.3 | 72.2 | 80.4 | 73.1 |
| DCC (Li et al., 2021) | ✗ | ✗ | 43.0 | **88.5** | 78.5 | 70.2 | 79.3 | 75.9 | 88.6 | 80.2 |
| OVANet (Saito & Saenko, 2021) | ✗ | ✗ | 53.1 | 85.8 | 79.4 | 80.1 | **95.4** | **84.0** | **94.3** | 86.5 |
| GATE (Chen et al., 2022) | ✗ | ✗ | **56.4** | 87.7 | **81.6** | **84.2** | 94.8 | 83.4 | 94.1 | **87.6** |
| **Best Epoch (Best H-score)** | | | | | | | | | | |
| SHOT-O (Liang et al., 2020) | ✓ | ✗ | 31.5 ±0.5 | 61.5 ±0.8 | 58.6 ±0.6 | 64.5 ±0.8 | 78.8 ±1.5 | 60.8 ±2.7 | 73.6 ±0.9 | 66.3 |
| GLC (Qu et al., 2023) | ✓ | ✗ | **65.4** ±2.0 | 81.6 ±0.3 | 83.6 ±0.5 | 87.8 ±0.3 | 90.1 ±0.3 | 86.0 ±0.4 | 90.2 ±0.0 | 86.6 |
| ABC | ✓ | ✓ | 61.7 ±0.2 | **86.4** ±0.2 | **94.8** ±1.2 | **88.8** ±0.8 | **94.9** ±0.4 | **89.1** ±0.4 | **93.5** ±0.2 | **91.2** |
| **Final Epoch** | | | | | | | | | | |
| SHOT-O (Liang et al., 2020) | ✓ | ✗ | 18.1 ±0.2 | 57.7 ±1.9 | 56.7 ±1.1 | 59.2 ±1.5 | 75.2 ±3.7 | 57.6 ±3.3 | 69.1 ±0.7 | 62.6 |
| GLC (Qu et al., 2023) | ✓ | ✗ | **63.7** ±3.2 | 70.5 ±1.0 | 81.8 ±0.9 | 79.8 ±1.0 | 88.6 ±0.2 | 75.8 ±1.6 | 84.1 ±0.7 | 80.1 |
| GBNC-only | - | - | 34.6 ±0.2 | 81.0 ±0.0 | 89.8 ±2.6 | 85.1 ±4.4 | 88.8 ±2.8 | 80.8 ±2.0 | **94.0** ±0.0 | 86.6 |
| ABC | ✓ | ✓ | 59.4 ±0.3 | **84.2** ±0.9 | **93.6** ±2.7 | **87.0** ±1.9 | **93.6** ±1.2 | **85.3** ±0.3 | 88.6 ±2.1 | **88.7** |

# 4 Experiments

## 4.1 Experiment Setup

**Datasets:** We evaluate on three widely-used benchmark datasets: 1. *Office-31* (Saenko et al., 2010), comprises more than 4k images from 31 categories across three domains: *Amazon* (**A**), *DSLR* (**D**), and *Webcam* (**W**). 2. *Office-Home* (Venkateswara et al., 2017), includes about 15,500 images categorized into 65 classes across four domains: *Artistic* (**Ar**), *Clip Art* (**Cl**), *Product* (**Pr**), and *Real-World* (**Rw**); 3. VisDA-C (Peng et al., 2017) is a challenging synthetic-to-real benchmark with 12 object categories, featuring around 152k synthetic images in the source domain and 55k real-world images in the target domain. In the main experiments, we focus on the setting where neither $\mathcal{C}_s$ nor $\mathcal{C}_t$ is a subset of the other ($\mathcal{C}_s \not\subseteq \mathcal{C}_t$ and $\mathcal{C}_s \not\supseteq \mathcal{C}_t$), which is known as open-partial domain adaptation (OPDA) (Qu et al., 2023). We follow the class split settings used in prior OPDA works (Qu et al., 2023), using 10/10/11 shared/source-private/target-private classes for Office-31, 10/5/50 for Office-Home, and 6/3/3 for VisDA-C. We also discuss the case where $\mathcal{C}_s \subseteq \mathcal{C}_t$, known as open-set domain adaptation (OSDA) (Qu et al., 2023), in Appendix D.5.

**Evaluation Protocol:** We follow the standard evaluation protocol used in prior works (Chen et al., 2022; Liang et al., 2021; Qu et al., 2023), reporting the harmonic mean (H-score) between the average accuracy on known classes and the accuracy on unknown-class samples, along with the average per-class accuracy (mean accuracy), treating the unknown class as a single category. Since these widely used metrics treat all unknown ground-truth classes as a single category, we additionally report Normalized Mutual Information (NMI (Vinh et al., 2010); described in Appendix C) to evaluate fine-grained classification. NMI measures the alignment between predicted and ground-truth labels across individual classes, independent of class label permutations, thereby treating each unknown class separately. For all metrics, we report the average performance of our method and the reproduced source-free baselines across three independent random experiments, together with the standard deviation for each transfer task.

We compare our method against representative non-SFDA baselines in terms of H-score, with their results reprinted from previous work (Chen et al., 2022; Qu et al., 2023). Additionally, we evaluate against two recent SFDA methods: SHOT (Liang et al., 2020) and GLC (Qu et al., 2023), with results reproduced using their publicly available codebases. As both methods predict all unknown samples as a single "unknown" class and do not directly produce fine-grained classifications, we apply the *DPM*, using the same configuration as in our model, to the unknown target samples predicted by SHOT and GLC, and then combine the resulting cluster assignments with their known-class predictions to compute NMI against the ground-truth labels.

**Implementation Details:** For a fair comparison, we adopt the same backbone, ResNet-50 pretrained (He et al., 2016) on ImageNet (Deng et al., 2009), in line with existing benchmark methods SHOT (Liang et al., 2020) and GLC (Qu et al., 2023). We pretrain the source model using the code provided by GLC (Qu et al., 2023), following the source model pretraining procedure recommended in SHOT (Liang et al., 2020) to apply a 90/10 split for training and validation. Instead of a random split, we employ stratified sampling to maintain the class distribution across both training and validation sets, reducing the impact of class imbalance during source pretraining. Due to space constraints, implementation details and configuration settings of our method are provided in Appendix C.

## 4.2 Experiment Results

H-score, mean accuracy, and NMI results for all three datasets are reported in Tables 1a, 1b, A1, A2, 2a, and 2b, respectively. Due to the absence of standardized epoch selection criteria in SFDA methods, results are reported for both the **Best Epoch** (the highest H-score) and the **Final Epoch** (the last training epoch) for the two SFDA baselines and ours. Best Epoch metrics ensure comparability with prior literature, while Final Epoch metrics provide a practical, label-free evaluation that better reflects real-world performance. As shown in Tables 1a, 1b, A1, and A2, on both Office-31 and Office-Home, ABC outperforms GLC under the Final Epoch setting and consistently outperforms SHOT in terms of average H-score and mean accuracy. Notably, although GLC is one of the strongest baselines, ABC surpasses it by a substantial margin in the Final Epoch setting on both datasets, achieving at least a 10% improvement in average H-score, while also attaining higher mean accuracy under both settings. Furthermore, ABC delivers performance comparable to non-SFDA methods on Office-Home, and achieves higher average H-score on Office-31, regardless of whether results are taken from the best or final epoch. This demonstrates that ABC can achieve strong adaptation performance even without access to source data. On VisDA, while GLC slightly outperforms ABC, the gap is small, and ABC remains among the top-performing methods; both ABC and GLC significantly outperform SHOT, and surpass all compared non-SFDA methods.

From Tables 2a and 2b, which report NMI between true and predicted labels, including unknown classes, we observe that ABC achieves the highest average NMI on Office-31 (both settings) and on Office-Home under the Final Epoch, while closely matching GLC+DPM on Office-Home under the Best Epoch. On VisDA, ABC shows slightly lower average NMI compared to GLC+DPM. Note that the compared methods group all unknown classes into a single category and do not inherently support fine-grained (FG) classification by explicitly predicting labels for individual unknown classes. The NMI results of these methods are obtained by applying the DPM module, which is central to our approach, to their predicted unknown samples. Compared to appending a separate BNC step to existing methods, our approach integrates it into a unified framework, enabling joint optimization, minimizing error propagation between clustering and adapting the source model,

Table 2: NMI (%) on Office-Home, Office-31, and VisDA datasets under OPDA settings.

(a) Office-Home

| Methods | A2C | A2P | A2R | C2A | C2P | C2R | P2A | P2C | P2R | R2A | R2C | R2P | Avg |
|---|---|---|---|---|---|---|---|---|---|---|---|---|---|
| **Best Epoch** | | | | | | | | | | | | | |
| SHOT-O (Liang et al., 2020) + DPM | 48.6 ±0.5 | 66.5 ±0.1 | 64.3 ±0.5 | 57.6 ±0.5 | 63.3 ±0.2 | 63.1 ±0.5 | 57.4 ±0.7 | **48.0** ±0.1 | 64.0 ±0.9 | 58.0 ±0.6 | **48.8** ±0.5 | 67.4 ±0.5 | 58.9 |
| GLC (Qu et al., 2023) + DPM | **49.3** ±0.2 | **75.3** ±0.3 | 73.9 ±0.3 | 57.7 ±0.4 | **75.6** ±0.3 | **73.8** ±0.2 | **62.7** ±0.3 | 47.6 ±0.2 | **73.6** ±0.3 | **63.7** ±0.8 | 48.8 ±0.5 | 74.6 ±0.0 | **64.7** |
| ABC | 48.3 ±0.4 | 74.0 ±0.6 | **75.1** ±0.8 | **61.0** ±0.3 | 74.5 ±0.5 | 73.4 ±0.8 | 61.8 ±0.6 | 47.8 ±0.6 | 73.4 ±0.7 | 62.9 ±0.9 | 48.0 ±0.6 | **75.4** ±0.6 | 64.6 |
| **Final Epoch** | | | | | | | | | | | | | |
| SHOT-O (Liang et al., 2020) + DPM | 46.9 ±0.9 | 62.0 ±0.1 | 60.9 ±0.8 | 56.6 ±0.3 | 61.3 ±0.5 | 59.8 ±1.2 | 57.0 ±0.5 | 46.4 ±0.6 | 60.1 ±1.0 | 57.1 ±0.1 | 47.2 ±0.7 | 62.8 ±0.9 | 56.5 |
| GLC (Qu et al., 2023) + DPM | 40.4 ±0.3 | **75.5** ±0.4 | **73.7** ±0.3 | 49.0 ±0.3 | **75.6** ±0.3 | **73.7** ±0.3 | **63.6** ±0.2 | 35.4 ±0.4 | 73.1 ±1.1 | **64.5** ±0.2 | 40.3 ±0.5 | 74.2 ±0.0 | 61.6 |
| GBNC-only | **49.9** ±0.2 | 73.6 ±0.3 | 73.0 ±0.8 | 49.9 ±0.9 | 68.1 ±0.9 | 64.9 ±0.6 | 53.9 ±0.4 | 46.4 ±0.6 | 70.3 ±0.1 | 57.4 ±0.5 | **48.7** ±0.2 | 73.7 | 60.8 |
| ABC | 48.0 ±0.2 | 73.2 ±0.8 | 73.2 ±0.7 | **60.9** ±0.2 | 73.4 ±0.2 | 72.8 ±0.4 | 61.7 ±0.2 | **47.3** ±0.1 | **73.4** ±0.1 | 62.5 ±0.4 | 47.4 ±0.2 | **74.3** ±0.4 | **64.0** |

(b) Office-31 and VisDA

| Methods | VisDA | Office-31 | | | | | | |
|---|---|---|---|---|---|---|---|---|
| | S2R | A2D | A2W | D2A | D2W | W2A | W2D | Avg |
| **Best Epoch** | | | | | | | | |
| SHOT-O (Liang et al., 2020) + DPM | 39.4 ±0.4 | 78.0 ±0.5 | 77.6 ±0.4 | 68.3 ±1.2 | 82.1 ±0.2 | 67.8 ±0.7 | 83.0 ±0.2 | 76.1 |
| GLC (Qu et al., 2023) + DPM | **47.2** ±0.3 | 79.6 ±0.2 | 82.0 ±0.4 | 67.6 ±0.4 | 83.1 ±0.9 | 66.9 ±0.2 | 81.9 ±0.2 | 76.9 |
| ABC | 46.7 ±0.2 | **80.8** ±0.3 | **84.8** ±0.1 | **69.2** ±0.3 | **85.2** ±0.6 | **69.2** ±0.5 | 82.9 ±1.2 | **78.7** |
| **Final Epoch** | | | | | | | | |
| SHOT-O (Liang et al., 2020) + DPM | 43.5 ±0.6 | 79.5 ±0.2 | 79.1 ±0.2 | 70.0 ±0.7 | 82.4 ±0.3 | 68.6 ±0.6 | **83.6** ±0.3 | 77.2 |
| GLC (Qu et al., 2023) + DPM | 48.1 ±0.2 | **81.7** ±0.3 | 83.0 ±0.2 | 69.6 ±0.5 | 83.8 ±0.3 | 70.1 ±0.3 | 82.5 ±0.3 | 78.4 |
| GBNC-only | 32.4 ±0.1 | 78.8 ±0.2 | 84.2 ±0.7 | 69.1 ±0.1 | 83.5 ±0.4 | 69.9 ±0.7 | 83.2 ±0.0 | 78.1 |
| ABC | 46.9 ±0.3 | 81.0 ±0.2 | **85.1** ±0.3 | **70.4** ±0.6 | **84.9** ±0.4 | 70.5 ±0.4 | 81.5 ±0.5 | **78.9** |

and eliminating the need for post hoc processing. We also evaluate the GBNC component in isolation (GBNC-only), which performs competitively on Office-31 and Office-Home but significantly worse on VisDA, demonstrating both its effectiveness and the added value of the full learning framework. Furthermore, we evaluate ABC's performance in estimating the number of unknown clusters in Appendix D.3, focusing on major clusters containing more than 1% of the data. As shown in Figure A1, the estimated number closely matches the ground truth for Office-31 and VisDA, and is underestimated for Office-Home. Although estimating $K$ without prior knowledge is inherently challenging, ABC provides a reasonable approximation across datasets, demonstrating its ability to achieve fine-grained classification.

While our experimental validation primarily focuses on the OPDA setting, we also provide an additional evaluation under the Open-Set Domain Adaptation (OSDA) setting in the Appendix, where the source-private class set is empty (i.e., $\mathcal{C}_s \subseteq \mathcal{C}_t$). As shown in Appendix D.5, ABC also performs well in this setting, suggesting that it can generalize effectively to other scenarios involving unknown classes.

### 4.3 Experiment Analysis

To better understand the robustness and reliability of our method, we conduct comprehensive sensitivity and ablation analyses on key hyperparameters. For the sensitivity analyses, average results across tasks are reported in the main paper; per-task results are in Appendix D.6. All results are based on a single run. The results demonstrate that our method achieves robust performance across a wide range of hyperparameter values. In addition to the analyses presented in the main text, we include further studies in the Appendix on runtime and memory (Appendix D.2), clustering evaluation of the discovered unknown clusters samples (Appendix D.1), and t-SNE visualizations (Van der Maaten & Hinton, 2008) of discovered unknown clusters (Appendix D.4).

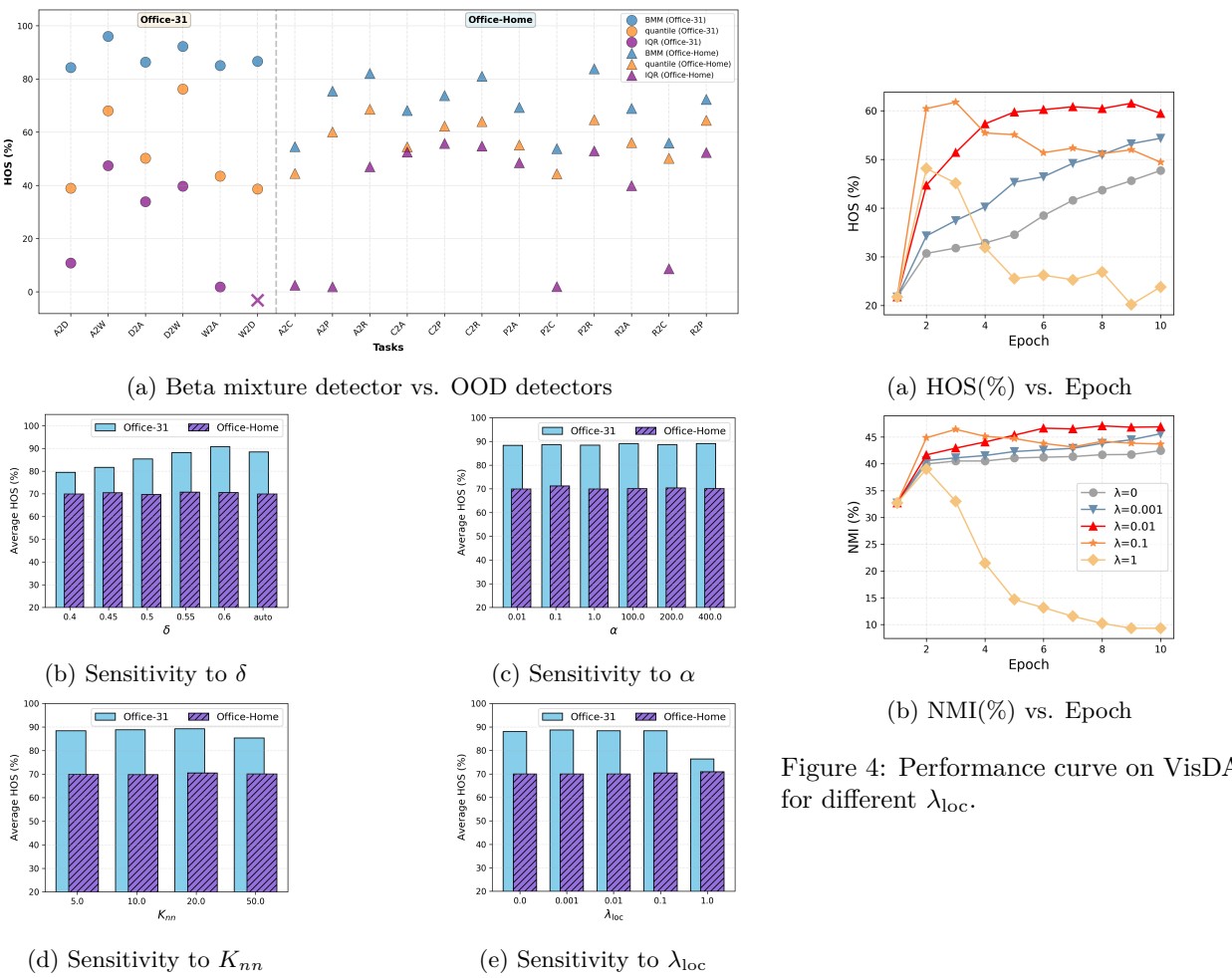

(a) Beta mixture detector vs. OOD detectors

(b) Sensitivity to $\delta$

(c) Sensitivity to $\alpha$

(d) Sensitivity to $K_{nn}$

(e) Sensitivity to $\lambda_{loc}$

(a) HOS(%) vs. Epoch

(b) NMI(%) vs. Epoch

Figure 4: Performance curve on VisDA for different $\lambda_{loc}$.

Figure 3: Hyperparameter sensitivity and ablation study (H-score) on Office-31 and Office-Home under OPDA.

### 4.3.1 High-Confidence Sample Selection

The anchor-based mechanism used in the first step to identify high-confidence common-class samples plays a critical role in the overall method. In this section, we present an ablation study to evaluate the impact of the Beta-mixture detector on high-confidence sample selection, along with a sensitivity analysis examining the effect of the decision-boundary threshold, $\delta$.

**Ablation Study of Beta-Mixture Detector:** In our method, we employ a Beta mixture model (BMM) to estimate the posterior probability $P(y \in \mathcal{C}_s \mid \kappa)$, representing the likelihood that a sample belongs to the common classes. To evaluate its role in high-confidence sample selection, we compare it with an alternative strategy based on out-of-distribution (OOD) detection for identifying unknown-class samples. Unlike the BMM, which explicitly models indicator values of known and unknown samples as two mixture components, OOD approaches treat unknown-class samples as outliers relative to the distribution of known-class samples. We consider two widely used statistical OOD methods based on $1 - \kappa$: a quantile-based approach and the interquartile range (IQR) method (Tukey et al., 1977). For both, we adopt automatic threshold selection. Specifically, for the quantile method, the quantile is searched over $\{0.05, 0.1, 0.15, 0.2, 0.25\}$; for the IQR method, the multiplier $k$ is varied over $\{0.5, 1.0, 1.5, 2.0, 3.0\}$ using a one-sided (lower-tail) criterion. In each case, the threshold that maximizes the Silhouette score is selected. We evaluate these detectors on Office-31 and Office-Home and compare them with BMM. When the OOD detectors classify too many samples as

known-class, yielding too few outliers, we adaptively reduce $K^*$ to ensure GBNC remains computationally feasible. We report the comparison between BMM and the OOD approaches at the final epoch. H-scores are shown in Figures 3a, with additional metrics in Figures A4. The results demonstrate that BMM consistently achieves superior performance in terms of H-scores, while the OOD approaches not only perform worse but also the IQR method fails on one task (marked by crosses). Regarding detection errors, two failure modes may occur when high-confidence labels are incorrect: (1) misassigned known-class samples, where confusion between similar source classes causes the GBNC step to propagate incorrect labels and bias class estimation; and (2) unknown samples treated as known, where excessive inclusion of unknown samples as anchors prevents discovery of some unknown classes. Empirically, the second case dominates: manual inspection shows the first case is rare (accounting for $\approx 0\%$ of incorrect labels in the majority of Office-31 tasks and $<5\%$ on average across Office-Home tasks). The poor performance of OOD alternatives therefore illustrates these failure scenarios, where inaccurate detectors misclassify many unknown samples as known. Overall, the results underscore the importance of the Beta mixture model, which explicitly models the indicator distributions of known and unknown samples rather than treating unknown samples merely as outliers.

$\delta$ **for High-Confidence Sample Selection:** To select high-confidence samples, we apply $P(y \in \mathcal{C}_s \mid \kappa) > \delta$. Setting $\delta$ too high results in too few selected samples, potentially omitting entire common classes, while a low $\delta$ may admit too many, increasing the risk of including misclassified unknown-class samples. The natural decision boundary, where $P(y \in \mathcal{C}_s \mid \kappa) > P(y \notin \mathcal{C}_s \mid \kappa)$, suggests a default threshold of $\delta = 0.5$, but it need not always be fixed at this value. For Office-31 and Office-Home, we tune $\delta$ per epoch within the range $[0.4, 0.6]$ (with a step size of 0.05), selecting the value that maximizes the Silhouette score (Rousseeuw, 1987) based on the embedded data and the estimated labels from the GBNC. To assess sensitivity, we also rerun our algorithm using fixed values $\delta \in \{0.4, 0.45, 0.5, 0.55, 0.6\}$ and compare the results to our tuning strategy (denoted as "auto"). As shown in Figures 3b, A8, A9, and A10, performance remains relatively stable around $\delta = 0.5$. The auto-tuning strategy yields similar results to fixed settings, further mitigating sensitivity to the choice of $\delta$ without requiring manual selection.

### 4.3.2 DPM Hyperparameter $\alpha$

The concentration parameter $\alpha$ plays a key role in the DPM model, controlling its ability to infer the number of clusters. Following a common practice, we set $\alpha = \frac{\alpha^*}{K^*}$ with $\alpha^* = 1$ as a noninformative prior, allowing the influence of the hyperparameters to diminish with increasing sample size. To assess sensitivity, we further report the H-score on Office-31 and Office-Home using $\alpha^* \in \{0.01, 0.1, 1, 100, 200, 400\}$ (i.e., $\alpha \in \{10^{-4}, 10^{-3}, 10^{-2}, 1, 2, 4\}$ for $K^* = 100$), as shown in Figures 3c, with additional metrics provided in Figures A8, A9, and A10. Results demonstrate robust performance, supporting the use of a non-informative prior. While $\alpha$ can also be estimated via empirical Bayes or assigned a hyperprior, where the hyperparameters are either inferred from the data or exert less influence higher in the model hierarchy, the performance stability across values suggests that our choice achieves reliable performance while providing both efficiency and simplicity, making the modeling process more controllable. In general, we prefer non-informative priors in our model to let the data drive the clustering process while keeping the implementation simple and efficient.

### 4.3.3 $K_{nn}$ for the Calculation of $\mathcal{L}_{\text{loc}}$

For Office-31 and Office-Home, we evaluate performance with different number of nearest neighbors $K_{nn} \in \{5, 10, 20, 50\}$, reporting H-scores in Figures 3d and additional metrics in Figures A8, A9, and A10. Our method demonstrates robustness to the choice of $K_{nn}$, with H-scores remaining stable across tested values. Given the size of these two datasets, this range of $K_{nn}$ is appropriate: too few neighbors (e.g., 1 or 2) can lead to unstable estimates, especially when those neighbors are misclassified, while too many may introduce noise by incorporating samples from different classes, as noted in similar settings (Yang et al., 2022). For larger datasets like VisDA, we adopt a higher $K_{nn}$ to account for its much larger target domain.

### 4.3.4 $\lambda_{\text{loc}}$ for Local Prediction Consistency Regularizer

We evaluate the impact of $\lambda_{\text{loc}} \in \{0, 0.001, 0.01, 0.1, 1\}$. H-scores on Office-31 and Office-Home are shown in Figures 3e, with NMI and mean accuracy provided in Figures A8, A9, and A10. Results show that performance

remains generally stable, indicating robustness to $\lambda_{\mathrm{loc}}$, though larger values may degrade performance on Office-31. For VisDA, we analyze performance across training epochs (Figure 4). $\lambda_{\mathrm{loc}}$ shows a stronger impact: small $\lambda_{\mathrm{loc}}$ values performs better compared to $\lambda_{\mathrm{loc}} = 0$, and it shows more stable, gradual convergence while larger values lead to faster convergence but compromise final performance. Across small $\lambda_{\mathrm{loc}}$ values, metrics steadily improve over time, indicating stable and effective learning. This suggests that $\mathcal{L}_{\mathrm{loc}}$ can be particularly helpful when baseline adaptation performance is weak, typically in challenging tasks, as in VisDA.

## 5   Conclusion

In this paper, we propose Adapt via Bayesian Nonparametric Clustering (ABC) for SFDA with unknown target classes, enabling both discovery and fine-grained classification of unknown classes. Through the development of a guided Bayesian nonparametric clustering algorithm, our approach first identifies high-confidence target samples from known classes and then uses them as anchors to effectively cluster known-class samples and discover previously unseen ones, without requiring prior knowledge of the number of unknown classes or specifying it in advance. A prototype-based regularizer and a local prediction-consistency term are incorporated to further refine the source model for adaptation to the target data. Experiments demonstrate that ABC achieves solid performance across various benchmarks and evaluation metrics. By enabling the identification and classification of unknown classes in new domains without access to source data, ABC has the potential to provide important practical values across a range of real-world applications.

**Acknowledgments**

This work was partially supported by a start-up fund from the College of Arts and Sciences at the University of Kentucky. Computational resources were also provided by the College of Arts and Sciences at the University of Kentucky.

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

# Appendix

| Appendix | Description |
|----------|-------------|
| Appendix A | More Details about Guided Nonparametric Bayesian Clustering |
| Appendix B | Approximate Posterior Inference via Variational Bayes |
| Appendix C | Additional Experiment Setup |
| Appendix D | Additional Results |

## A  More Details about Guided Nonparametric Bayesian Clustering

Treating the embedding data $\mathcal{Z}_t^U = \{z_1^U, \ldots, z_{n_U}^U\}$ as observed data, a Dirichlet Process Mixture (DPM) model for $\mathcal{Z}_t^U$ can be naturally represented within a Bayesian hierarchical framework (same as Equation 2),

$$z_i^U|\eta_i \sim p(z_i^U|\eta_i), \quad \eta_i|G \sim G, \quad G \sim DP(\alpha, G_0). \tag{A1}$$

The probability function $p(\cdot \mid \eta_i)$ is parameterized by $\eta_i$, which is drawn from a DP-distributed prior $G$. In our method, we adopt a Dirichlet Process Gaussian Mixture Model (DPGMM), where $p(\cdot \mid \eta_i)$ is a multivariate Gaussian, and $\eta_i$ consists of the mean and covariance parameters of the Gaussian. Equation A1 can be equivalently formulated as an infinite mixture model in Equation A2 by marginalizing out $G$.

$$z_i^U|y_i^U = k \sim p(z_i^U|\eta_k^\star), \quad y_i^U|\pi \sim Cat(\boldsymbol{\pi}), \quad \boldsymbol{\pi} = (\pi_1, \pi_2, \cdots)$$
$$\pi_k = v_k \prod_{j=1}^{k-1}(1 - v_j), \quad v_k \sim Beta(1, \alpha), \quad \eta_k^\star \sim G_0 \tag{A2}$$

where $\eta_k^\star$ represents the parameters of the distribution associated with the mixture component $k$ and $\boldsymbol{\pi}$ is a probability vector specifying the mixing distribution over clusters $k = 1, 2, \ldots$. The categorical distribution, denoted by $Cat(\cdot)$, assigns cluster memberships according to $y_i^U \mid \boldsymbol{\pi} \sim Cat(\boldsymbol{\pi})$, which is equivalent to the probability expression $P(y_i^U = k) = \pi_k$. Equation A2 adopts the stick-breaking representation, where the DP prior $G$ is explicitly expressed as an infinite mixture of point masses weighted by $\pi_k$ (Sethuraman, 1994; Blei & Jordan, 2006). In Equation A2, the mixing proportions $\pi_k$ can be viewed as being generated by sequentially "breaking" a unit-length stick into infinitely many pieces, with each piece determined by $v_k$ from the remaining stick.

For the embedding data from high-confidence samples, $\mathcal{Z}_t^H = \{z_1^H, \ldots, z_n^H\}$, we treat their labels $\hat{\mathcal{Y}}_t^H = \{\hat{y}_1^H, \ldots, \hat{y}_n^H\}$ obtained during the high-confidence sample selection step, as fixed, giving:

$$z_i^H \sim p(z_i^H|\eta_{\hat{y}_i^H}^\star). \tag{A3}$$

With the alignment mentioned in Section 3.2, we can combine Equation A3 with Equation A2 to obtain (same as Equation 3):

$$z_i^U|y_i^U = k \sim p(z_i^U|\eta_k^\star), \quad z_i^H \sim p(z_i^H|\eta_{\hat{y}_i^H}^\star)$$
$$y_i^U|\pi \sim Cat(\boldsymbol{\pi}), \quad \boldsymbol{\pi} = (\pi_1, \pi_2, \cdots, \pi_{|\widehat{\mathcal{C}}|}, \cdots)$$
$$\pi_k = v_k \prod_{j=1}^{k-1}(1 - v_j), \quad v_k \sim Beta(1, \alpha), \quad \eta_k^\star \sim G_0 \tag{A4}$$

where $p(\cdot|\eta_k^\star) = N(\cdot|\mu_k, \Lambda_k)$. We place a conjugate Normal-Wishart prior, $N(\mu_k|\zeta, \tau\Lambda_k) \times W(\Lambda_k|b, \Omega)$ on $(\mu_k, \Lambda_k)$, where $\tau$, $b$ and $\Omega$ are Bayesian hyperparameters.

# B  Approximate Posterior Inference via Variational Bayes

Our model is parameterized by four sets of parameters and latent variables: $\{v_k\}_{k=1}^{\infty}$, $\{\mu_k\}_{k=1}^{\infty}$, $\{\Lambda_k\}_{k=1}^{\infty}$, and $\{y_i^U\}_{i=1}^{n_U}$, where $n_U$ denotes the size of $\mathcal{Z}_t^U$. We employ Variational Bayes (VB) to approximate the posterior distribution. VB approximates the true posterior by minimizing the KL divergence between a variational distribution and the posterior. To facilitate this, we apply the mean filed approximation and assume a fully factorized variational distribution (Bishop & Nasrabadi, 2006),

$$q(\{v_k\}_{k=1}^{K^*-1}, \{\mu_k\}_{k=1}^{K^*}, \{\Lambda_k\}_{k=1}^{K^*}, , \{y_i^U\}_{i=1}^{n_U}) = \prod_{k=1}^{K^*-1} q(v_k) \prod_{k=1}^{K^*} q(\mu_k, \Lambda_k) \prod_{i=1}^{n_U} q(y_i^U), \tag{A5}$$

The stick-breaking process is truncated at a value $K^*$ with $q(v_{K^*} = 1) = 1$ for computation purpose, a construction known as the truncated stick-breaking representation. It should be noted that $K^*$ does not represent a pre-determined number of clusters, as in conventional clustering algorithms like K-Means. Instead, it serves as a practical upper bound on the number of clusters, and any sufficiently large value (e.g., 100) is generally expected to yield consistent results. We now present the model specifications.

$$z_i^U \mid y_i^U, \boldsymbol{\mu}, \boldsymbol{\Lambda} \sim \mathcal{N}(\mu_{y_i^U}, \Lambda_{y_i^U})$$
$$z_i^H \mid \boldsymbol{\mu}, \boldsymbol{\Lambda} \sim \mathcal{N}(\mu_{\hat{y}_i^H}, \Lambda_{\hat{y}_i^H}) \quad (\hat{y}_i^H \text{ considered as fixed})$$
$$y_i^U \mid \boldsymbol{\pi} \sim \mathrm{Cat}(\boldsymbol{\pi})$$
$$\mu_k \mid \Lambda_k \sim \mathcal{N}(\zeta, \tau\Lambda_k)$$
$$\Lambda_k \sim \mathcal{W}(b, \Omega)$$
$$\pi_k = v_k \prod_{j=1}^{k-1}(1 - v_j)$$
$$v_k \sim \mathrm{Beta}(1, \alpha) \tag{A6}$$

Given Equation A6, the joint distribution of all random variables, expressed in their vectorized forms, can be factorized as:

$$p(\mathbf{Z}^U, \mathbf{Z}^H, \boldsymbol{Y}^U, \boldsymbol{v}, \boldsymbol{\mu}, \boldsymbol{\Lambda}) = p(\mathbf{Z}^U \mid \boldsymbol{Y}^U, \boldsymbol{\mu}, \boldsymbol{\Lambda}) p(\mathbf{Z}^H \mid \boldsymbol{\mu}, \boldsymbol{\Lambda}) p(\boldsymbol{Y}^U \mid \boldsymbol{v}) p(\boldsymbol{v}) p(\boldsymbol{\mu} \mid \boldsymbol{\Lambda}) p(\boldsymbol{\Lambda}) \tag{A7}$$

The full conditional distributions in our model belong to the exponential family. Accordingly, we select variational distributions also from the exponential family: $q(v_k) = Beta(v_k | \gamma_{k1}, \gamma_{k2})$, $q(\mu_k, \Lambda_k) = N(\mu_k \mid m_k, \tau_k\Lambda_k)W(\Lambda_k \mid c_k, D_k)$, $q(y_i^U) = Cat(\phi_i)$, and apply mean-field variational inference for exponential families (Blei & Jordan, 2006). Variational inference typically proceeds via a coordinate ascent algorithm, updating the parameters of each variational distribution sequentially. Under the mean-field framework, the variational parameters for each distribution are updated by computing the expected natural parameters of the corresponding full conditional, evaluated with respect to the current variational distributions. For completeness, we provide below the detailed update steps for our model.

Variational distributions:

$$q(\mu_k, \Lambda_k) = N(\mu_k \mid m_k, \tau_k\Lambda_k)W(\Lambda_k \mid c_k, D_k)$$
$$q(v_k) = Beta(v_k \mid \gamma_{k1}, \gamma_{k2})$$
$$q(y_i^U = k) = \phi_{ik}^{I(y_i^U = k)} \tag{A8}$$

Update variational parameters $(\tau_k, m_k, c_k, D_k)$:

To simplify the notations, we let $n_k = \sum_{i=1}^{n_U} I(y_i^U = k)$, $\hat{n}_k = E_q[n_k] = \sum_{i=1}^{n_U} \phi_{ik}$, and $\hat{n}_k^H = \sum_{i=1}^{n_H} I(\hat{y}_i^H = k)$. Also, we set

$$\bar{z}_k = \frac{1}{n_k + \hat{n}_k^H}[\sum_{i=1}^{n_U} I(y_i^U = k)z_i^U + \sum_{i=1}^{n_H} I(\hat{y}_i^H = k)z_i^H] \tag{A9}$$

$$S_k = \frac{1}{n_k + \hat{n}_k^H}[\sum_{i=1}^{n_U} I(y_i^U = k)\left(z_i^U - \bar{z}_k\right)\left(z_i^U - \bar{z}_k\right)^{\mathrm{T}} + \sum_{i=1}^{n_H} I(\hat{y}_i^H = k)\left(z_i^H - \bar{z}_k\right)\left(z_i^H - \bar{z}_k\right)^{\mathrm{T}}] \tag{A10}$$

For the likelihood of the parameters associated with cluster $k$, given the embedding data $z_i^U$ and $z_i^H$, we have

$$
\begin{aligned}
&\prod_{i=1}^{n_U} N(z_i^U \mid \mu_k, \Lambda_k)^{I(y_i^U=k)} \prod_{i=1}^{n_H} N(z_i^H \mid \mu_k, \Lambda_k)^{I(\hat{y}_i^H=k)} \\
&\propto \exp\left(-\frac{1}{2}\sum_{i=1}^{n_U} I(y_i^U = k)\{(z_i^U - \mu_k)^\top \Lambda_k(z_i^U - \mu_k) - \log|\Lambda_k|\}\right) \\
&\times \exp\left(-\frac{1}{2}\sum_{i=1}^{n_H} I(\hat{y}_i^H = k)\{(z_i^H - \mu_k)^\top \Lambda_k(z_i^H - \mu_k) - \log|\Lambda_k|\}\right) \\
&\propto \exp\left(-\frac{n_k + \hat{n}_k^H}{2}\operatorname{tr}(S_k\Lambda_k) - \frac{n_k + \hat{n}_k^H}{2}(\bar{z}_k - \mu_k)^\top\Lambda_k(\bar{z}_k - \mu_k) + \frac{n_k + \hat{n}_k^H}{2}\log|\Lambda_k|\right)
\end{aligned}
\tag{A11}
$$

Let $d$ denote the dimensionality of the embedding data. We can easily show that the full conditional distribution of each $\mu_k$, $\Lambda_k$.

$$
\begin{aligned}
p(\mu_k, \Lambda_k \mid \cdot) &\propto \prod_{i=1}^{n_U} N(z_i^U \mid \mu_k, \Lambda_k)^{I(y_i^U=k)} \prod_{i=1}^{n_H} N(z_i^H \mid \mu_k, \Lambda_k)^{I(\hat{y}_i^H=k)} N(\mu_k \mid \zeta, \tau\Lambda_k) W(\Lambda_k \mid b, \Omega) \\
&\propto \exp\left(-\frac{n_k + \hat{n}_k^H}{2}\operatorname{tr}(S_k\Lambda_k) - \frac{n_k + \hat{n}_k^H}{2}(\bar{z}_k - \mu_k)^\top\Lambda_k(\bar{z}_k - \mu_k) + \frac{n_k + \hat{n}_k^H}{2}\log|\Lambda_k|\right) \\
&\times \exp\left(-\frac{1}{2}\tau(\mu_k - \zeta)^\top\Lambda_k(\mu_k - \zeta) + \frac{1}{2}\log|\Lambda_k|\right) \\
&\times \exp\left(-\frac{1}{2}\operatorname{tr}(\Omega^{-1}\Lambda_k) + \frac{b-d-1}{2}\log|\Lambda_k|\right) \\
&\propto \exp\left(-\frac{1}{2}[(n_k + \hat{n}_k^H)(\bar{z}_k - \mu_k)^\top\Lambda_k(\bar{z}_k - \mu_k) + \tau(\mu_k - \zeta)^\top\Lambda_k(\mu_k - \zeta)]\right. \\
&\quad \left. -\frac{1}{2}\operatorname{tr}(((n_k + \hat{n}_k^H)S_k + \Omega^{-1})\Lambda_k) + \frac{1}{2}(n_k + \hat{n}_k^H + b - d)\log|\Lambda_k|\right) \\
&\propto \exp\left(-\frac{1}{2}[(\tau + n_k + \hat{n}_k^H)(\mu_k - \frac{\tau\zeta + (n_k + \hat{n}_k^H)\bar{z}_k}{\tau + n_k + \hat{n}_k^H})^\top\Lambda_k(\mu_k - \frac{\tau\zeta + (n_k + \hat{n}_k^H)\bar{z}_k}{\tau + n_k + \hat{n}_k^H})\right. \\
&\quad \left. +\frac{(n_k + \hat{n}_k^H)\tau}{\tau + n_k + \hat{n}_k^H}(\bar{z}_k - \zeta)^\top\Lambda_k(\bar{z}_k - \zeta)]\right. \\
&\quad \left. -\frac{1}{2}\operatorname{tr}((\Omega^{-1} + (n_k + \hat{n}_k^H)S_k)\Lambda_k) + \frac{1}{2}(n_k + \hat{n}_k^H + b - d)\log|\Lambda_k|\right) \\
&\propto \exp\left(-\frac{1}{2}(\tau + n_k + \hat{n}_k^H)(\mu_k - \frac{\tau\zeta + (n_k + \hat{n}_k^H)\bar{z}_k}{\tau + n_k + \hat{n}_k^H})^\top\Lambda_k(\mu_k - \frac{\tau\zeta + (n_k + \hat{n}_k^H)\bar{z}_k}{\tau + n_k + \hat{n}_k^H})\right. \\
&\quad \left. -\frac{1}{2}\operatorname{tr}((\Omega^{-1} + (n_k + \hat{n}_k^H)S_k + \frac{(n_k + \hat{n}_k^H)\tau}{\tau + n_k + \hat{n}_k^H}(\bar{z}_k - \zeta)(\bar{z}_k - \zeta)^\top)\Lambda_k)\right. \\
&\quad \left. +\frac{1}{2}(n_k + \hat{n}_k^H + b - d)\log|\Lambda_k|\right)
\end{aligned}
\tag{A12}
$$

Thus the natural parameters are

$$\eta_1 = \text{vec}(\Omega^{-1} + (n_k + \hat{n}_k^H)S_k + \frac{(n_k + \hat{n}_k^H)\tau}{\tau + n_k + \hat{n}_k^H}(\bar{z}_k - \zeta)(\bar{z}_k - \zeta)^\top + \tag{A13}$$

$$\frac{(\tau\zeta + (n_k + \hat{n}_k^H)\bar{z}_k)(\tau\zeta + (n_k + \hat{n}_k^H)\bar{z}_k)^\top}{\tau + n_k + \hat{n}_k^H}) \tag{A14}$$

$$\eta_2 = \tau\zeta + (n_k + \hat{n}_k^H)\bar{z}_k = \sum_i^{n_U} I(y_i^U = k)z_i^U + \sum_i^{n_H} I(\hat{y}_i^H = k)z_i^H + \tau\zeta, \tag{A15}$$

$$\eta_3 = n_k + \hat{n}_k^H + \tau, \tag{A16}$$

$$\eta_4 = n_k + \hat{n}_k^H + b - d \tag{A17}$$

Solve $\tau_k$

$$\tau_k = (\tau + \hat{n}_k + \hat{n}_k^H)$$
$$\Rightarrow \tau_k = \tau + \hat{n}_k + \hat{n}_k^H \tag{A18}$$

Solve $m_k$

$$\tau_k m_k = E_q[\sum_i^{n_U} I(y_i^U = k)z_i^U + \sum_i^{n_H} I(\hat{y}_i^H = k)z_i^H + \tau\zeta]$$

$$\tau_k m_k = \tau\zeta + \sum_{i=1}^{n_U} \phi_{ik} z_i^U + \sum_{i=1}^{n_H} z_i^H I(\hat{y}_i^H = k)$$

$$\Rightarrow m_k = \frac{\tau\zeta + \sum_{i=1}^{n_U} \phi_{ik} z_i^U + \sum_{i=1}^{n_H} z_i^H I(\hat{y}_i^H = k)}{\tau + \hat{n}_k + \hat{n}_k^H} \tag{A19}$$

Solve $c_k$

$$c_k - d = E_q[n_k + \hat{n}_k^H + b - d]$$
$$c_k - d = b - d + \hat{n}_k + \hat{n}_k^H$$
$$\Rightarrow c_k = \hat{n}_k + \hat{n}_k^H + b \tag{A20}$$

Solve $D_k$

$$D_k^{-1} + \tau_k m_k m_k^\top = E_q\Big[\Omega^{-1} + (n_k + \hat{n}_k^H)S_k + \frac{(n_k + \hat{n}_k^H)\tau}{\tau + n_k + \hat{n}_k^H}(\bar{z}_k - \zeta)(\bar{z}_k - \zeta)^\top \tag{A21}$$

$$+ \frac{(\tau\zeta + (n_k + \hat{n}_k^H)\bar{z}_k)(\tau\zeta + (n_k + \hat{n}_k^H)\bar{z}_k)^\top}{\tau + n_k + \hat{n}_k^H}\Big]$$

$$D_k^{-1} + \tau_k m_k m_k^\top = E_q\Big[\Omega^{-1} + \sum_{i=1}^{n_U} I(y_i^U = k)\left(z_i^U - \bar{z}_k\right)\left(z_i^U - \bar{z}_k\right)^{\mathrm{T}} \tag{A22}$$

$$+ \sum_{i=1}^{n_H} I(\hat{y}_i^H = k)\left(z_i^H - \bar{z}_k\right)\left(z_i^H - \bar{z}_k\right)^{\mathrm{T}} + (n_k + \hat{n}_k^H)\bar{z}_k\bar{z}_k^\top + \tau\zeta\zeta^\top\Big]$$

$$D_k^{-1} + \tau_k m_k m_k^\top = E_q\Big[\Omega^{-1} + (\sum_{i=1}^{n_U} I(y_i^U = k)z_i^U(z_i^U)^\top + \sum_{i=1}^{n_H} I(\hat{y}_i^H = k)z_i^H(z_i^H)^\top)$$

$$- 2\bar{z}_k(\sum_{i=1}^{n_U} I(y_i^U = k)(z_i^U)^\top + \sum_{i=1}^{n_H} I(\hat{y}_i^H = k)(z_i^H)^\top) + 2(n_k + \hat{n}_k^H)\bar{z}_k\bar{z}_k^\top + \tau\zeta\zeta^\top\Big]$$

$$D_k^{-1} + \tau_k m_k m_k^\top = E_q\Big[\Omega^{-1} + (\sum_{i=1}^{n_U} I(y_i^U = k)z_i^U(z_i^U)^\top + \sum_{i=1}^{n_H} I(\hat{y}_i^H = k)z_i^H(z_i^H)^\top + \tau\zeta\zeta^\top\Big]$$

$$D_k^{-1} = \Omega^{-1} + \sum_{i=1}^{n_U} \phi_{ik}z_i^U(z_i^U)^\top + \sum_{i=1}^{n_H} I(\hat{y}_i^H = k)z_i^H(z_i^H)^\top + \tau\zeta\zeta^\top - \tau_k m_k m_k^\top \tag{A23}$$

If we further set:

$$\bar{z}_k^* = \frac{1}{\hat{n}_k + \hat{n}_k^H}\Big[\sum_{i=1}^{n_U} \phi_{ik}z_i^U + \sum_{i=1}^{n_H} I(\hat{y}_i^H = k)z_i^H\Big] \tag{A24}$$

$$S_k^* = \frac{1}{\hat{n}_k + \hat{n}_k^H}\Big[\sum_{i=1}^{n_U} \phi_{ik}\left(z_i^U - \bar{z}_k^*\right)\left(z_i^U - \bar{z}_k^*\right)^{\mathrm{T}} + \sum_{i=1}^{n_H} I(\hat{y}_i^H = k)\left(z_i^H - \bar{z}_k^*\right)\left(z_i^H - \bar{z}_k^*\right)^{\mathrm{T}}\Big] \tag{A25}$$

We can rearrange Equation (A23) to have:

$$D_k^{-1} = \Omega^{-1} + (\hat{n}_k + \hat{n}_k^H)S_k^* + \frac{(\hat{n}_k + \hat{n}_k^H)\tau}{\tau + \hat{n}_k + \hat{n}_k^H}(\bar{z}_k^* - \zeta)(\bar{z}_k^* - \zeta)^\top \tag{A26}$$

Update variational parameters $(\gamma_{k1}, \gamma_{k2})$:

Given $\hat{n}_k = E_q[n_k] = \sum_{i=1}^{n_U} \phi_{ik}$.

$$\gamma_{k1} = E_q[n_k + 1] = \hat{n}_k + 1 \tag{A27}$$

$$\gamma_{k2} = E_q[\sum_{j>k} n_j + \alpha] = \sum_{j>k} \hat{n}_j + \alpha \tag{A28}$$

Update variational parameters $\phi_i = (\phi_{i1}, \phi_{i2}, \dots)$:

$$\log(\phi_{ik}) \propto E_q[\log \pi_k + \log N(z_i^U|\mu_k, \Lambda_k)]$$

$$\propto E_q\Big[\log \upsilon_k + \sum_{j=1}^{k-1} \log(1 - \upsilon_j) + \frac{1}{2}\log|\Lambda_k| - \frac{1}{2}(z_i^U - \mu_k)^T \Lambda_k (z_i^U - \mu_k)\Big]$$

$$\propto \psi(\gamma_{k1}) - \psi(\gamma_{k1} + \gamma_{k2}) + \sum_{j=1}^{k-1}[\psi(\gamma_{j2}) - \psi(\gamma_{j1} + \gamma_{j2})]$$

$$+ \frac{1}{2}\Big[\psi_d\Big(\frac{c_k}{2}\Big) + \log|D_k|\Big] - \frac{1}{2}[d\tau_k^{-1} + c_k(z_i^U - m_k)^\top D_k(z_i^U - m_k)] \tag{A29}$$

where $\psi(\cdot)$ and $\psi_d(\cdot)$ represent the digamma and multivariate digamma functions, respectively.

Given these update steps for the variational parameters, we employ a coordinate descent algorithm to iteratively refine them until a convergence criterion is met, such as the convergence of the Evidence Lower Bound (ELBO), which is typically used as the objective in standard Dirichlet Process Mixture (DPM) models (Blei & Jordan, 2006). For initialization, we assign clusters using a variant of the KMeans algorithm, adapted to our setting (standard KMeans is commonly used in the variational Bayes (VB) procedure for DPMs). In this variant, we incorporate the labels of high-confidence samples during updates of the source-class cluster centers and modify the KMeans++ initialization (Arthur & Vassilvitskii, 2006) to consider class-center information from these samples at the start of KMeans. Finally, using the estimated values of $\phi_{ik}$, the predicted labels for the uncertain samples, $\hat{\mathcal{Y}}_t^U = \{\hat{y}_1^U, \ldots, \hat{y}_{n_U}^U\}$, are determined by $\hat{y}_i^U = \arg\max_k \phi_{ik}$.

In our method section, we assume all estimated labels $\hat{y}_i^H$ take values from the index set $\{1, 2, \ldots, |\widehat{\mathcal{C}}|\}$ (for clarity, we use one-based indices in the method description, while in practice zero-based indices are used). This allows unknown clusters to be represented by indices larger than $|\widehat{\mathcal{C}}|$ ($|\widehat{\mathcal{C}}| - 1$ in zero-based indexing), enabling prototype-based regularization to use one-hot encodings of cluster labels. Inference is then performed by assigning samples to the nearest prototypes in $\mathbf{c} = [c_1, c_2, \ldots, c_K]$ with consecutive indices. In our experiments, all datasets use zero-based class indices, and transfer tasks are designed so that unknown classes are assigned indices greater than or equal to the number of source classes, $|\mathcal{C}_s|$ (following the conventions of prior work). For example, in Office-31 with a 10/10/11 split (shared/source-private/target-private classes), shared classes get indices $0 \sim 9$, source-private classes get indices $10 \sim 19$, and unknown target classes get indices $\geq 20$. After the GBNC step, we add $|\mathcal{C}_s|$ only to the labels of samples whose class is not among those represented by the high-confidence samples $\hat{\mathcal{Y}}_t^H$, ensuring clear separation between known and unknown classes for subsequent implementation. We then reindex all estimated labels to consecutive zero-based indices $\{0, 1, 2, \ldots\}$ using a monotonic mapping that preserves ordering. This produces both the reindexed label array and a reverse mapping from new indices to original labels, which we use during inference to identify unknown-class samples and recover original cluster indices for known classes.

## C  Additional Experiment Setup

**Normalized Mutual Information (NMI) (Vinh et al., 2010).** NMI is a standard metric for evaluating the agreement between two label assignments, invariant to permutations of the class labels. Given label assignments $\mathcal{Y}_a$ and $\mathcal{Y}_b$, NMI can be computed by scaling the mutual information between $\mathcal{Y}_a$ and $\mathcal{Y}_b$ by the arithmetic mean of their respective entropies:

$$NMI(\mathcal{Y}_a; \mathcal{Y}_b) = \frac{2 \times I(\mathcal{Y}_a; \mathcal{Y}_b)}{H(\mathcal{Y}_a) + H(\mathcal{Y}_b)}, \tag{A30}$$

where $I(\cdot; \cdot)$ represents the mutual information between two random variables and $H(\cdot)$ represents the entropy of a random variable. NMI takes values between 0, suggesting no mutual information, and 1, indicating perfect alignment, with higher values reflecting better correspondence between the two label assignments. In our evaluation, we compute $NMI(\mathcal{Y}; \hat{\mathcal{Y}})$, comparing ground-truth labels $\mathcal{Y}$ with predicted cluster labels $\hat{\mathcal{Y}}$ across all samples. To evaluate NMI for the compared methods in our experiments, which requires estimated

labels for unknown-class samples, we apply the *DPM* component, using the same model configurations as in our approach, to the unknown target samples predicted by SHOT and GLC, then combine the resulting cluster assignments with their known-class predictions to compute NMI against the ground-truth labels.

**Additional Settings:** During target model adaptation, we use the SGD optimizer with a momentum of 0.9 and weight decay of 0.001, following the conventional inverse learning rate decay schedule defined as $\eta_l = \frac{\eta_0}{(1+10\cdot\min(1,\frac{l}{L}))^{0.75}}$, where $\eta_0$ is the initial learning rate. $\eta_0$ is set to $10^{-2}$ for the classifier on Office-31, $10^{-4}$ for Office-Home, and $10^{-3}$ for VisDA, with a scaling factor of 0.1 applied to the extractor and bottleneck layers. The batch size is fixed at 64 for all benchmark datasets. The weight for $\lambda_{\text{loc}}$ is set to 0.01. The number of nearest neighbors, $K_{\text{nn}}$, is set to 5 for both Office-31 and Office-Home, and 100 for VisDA to account for its considerably larger data scale. We run Algorithm 1 for 10 epochs across all three datasets. For high-confidence sample selection, we use the default threshold value $\delta = 0.5$ for the VisDA dataset. For the other two smaller datasets, we tune $\delta$ over the range $[0.4, 0.6]$ in increments of 0.05, selecting the value that maximizes the Silhouette score (Rousseeuw, 1987) computed on the embedded data and the estimated labels produced by the GBNC. This choice is supported by the sensitivity analysis in Section 4.3. The hyperparameters in the DP base distribution $G_0$ are set to $\zeta = \bar{z}$, which is the mean of embedding data, $\tau = 1$, $b = d$, and $\Omega^{-1} = 0.001 \times d \times I_d$, where $d$ is the dimension of the embedding, which is 256 in our model. $K^*$ is chosen with 100 for all the datasets. The VB algorithm, as well as KMeans and KMeans++ initialization, for our GBNC model is implemented by adapting the DPM implementation provided in the *scikit-learn* library (Pedregosa et al., 2011). All experiments are conducted on a server with AMD EPYC 7413 24-Core processors and eight NVIDIA RTX A5500 GPUs, using PyTorch (Paszke et al., 2019).

# D    Additional Results

This section presents additional results, including tables, figures, and sensitivity analyses.

## D.1    Additional Results for OPDA

We report the mean accuracy of all methods under the OPDA settings, as shown in Table A1 and Table A2 for the Office-Home, Office-31, and VisDA datasets. These results are consistent with the trends observed in H-scores and NMI metrics (Tables 1a, 2a, 1b, 2b). Our method achieves the highest average mean accuracy on Office-Home and Office-31, and performs comparably to GLC, the best-performing method, on VisDA.

Table A1: Mean Accuracy (%) on the Office-Home dataset under OPDA settings.

| Methods | SF | FG | A2C | A2P | A2R | C2A | C2P | C2R | P2A | P2C | P2R | R2A | R2C | R2P | Avg |
|---|---|---|---|---|---|---|---|---|---|---|---|---|---|---|---|
| **Best Epoch** | | | | | | | | | | | | | | | |
| SHOT-O (Liang et al., 2020) | ✓ | ✗ | 36.9 | 55.9 | 58.1 | 52.0 | 55.2 | 62.0 | 50.7 | 41.0 | 62.7 | 53.1 | 36.9 | 56.5 | 51.8 |
| | | | ±1.8 | ±0.5 | ±3.0 | ±3.8 | ±2.1 | ±2.7 | ±0.8 | ±1.2 | ±0.4 | ±0.6 | ±0.3 | ±0.6 | |
| GLC (Qu et al., 2023) | ✓ | ✗ | 44.5 | 74.4 | 89.1 | 57.8 | 68.5 | **89.2** | 70.3 | 47.8 | **87.9** | 74.5 | 47.6 | 79.7 | 69.3 |
| | | | ±0.4 | ±1.7 | ±1.1 | ±0.2 | ±4.9 | ±0.5 | ±0.9 | ±1.0 | ±0.2 | ±0.7 | ±0.5 | ±0.0 | |
| ABC | ✓ | ✓ | **50.6** | **82.8** | **90.1** | **73.9** | **82.2** | 88.6 | **72.0** | **48.4** | 86.9 | **79.0** | **50.5** | **81.8** | **73.9** |
| | | | ±1.7 | ±2.0 | ±0.2 | ±1.8 | ±1.0 | ±0.3 | ±0.2 | ±1.9 | ±0.6 | ±0.8 | ±1.1 | ±1.1 | |
| **Final Epoch** | | | | | | | | | | | | | | | |
| SHOT-O (Liang et al., 2020) | ✓ | ✗ | 41.0 | 56.0 | 58.1 | 54.7 | 54.4 | 62.0 | 53.9 | 42.6 | 60.7 | 55.3 | 42.9 | 57.7 | 53.3 |
| | | | ±0.8 | ±0.8 | ±3.9 | ±0.4 | ±1.2 | ±0.2 | ±1.5 | ±0.3 | ±0.6 | ±0.6 | ±1.5 | ±0.9 | |
| GLC (Qu et al., 2023) | ✓ | ✗ | 47.3 | 74.5 | **90.9** | 59.3 | 68.5 | **89.9** | 70.1 | 30.8 | **88.2** | 77.3 | 38.3 | 81.9 | 68.1 |
| | | | ±0.2 | ±1.6 | ±0.3 | ±1.6 | ±5.0 | ±0.2 | ±0.3 | ±1.5 | ±0.2 | ±0.9 | ±2.5 | ±0.0 | |
| GBNC-only | - | - | 44.6 | 77.4 | 90.1 | 65.9 | 66.7 | 82.7 | 66.3 | 43.8 | 86.4 | 74.6 | 46.7 | **82.4** | 69.0 |
| | | | ±0.3 | ±2.9 | ±0.5 | ±1.7 | ±3.3 | ±0.6 | ±0.6 | ±2.4 | ±0.8 | ±0.6 | ±1.1 | ±1.8 | |
| ABC | ✓ | ✓ | **48.3** | **77.7** | 88.5 | **73.5** | **71.6** | 86.9 | **72.7** | **46.9** | 87.7 | **77.9** | **49.2** | 74.0 | **71.2** |
| | | | ±0.1 | ±4.0 | ±2.3 | ±1.8 | ±3.7 | ±0.3 | ±0.4 | ±0.6 | ±1.2 | ±1.4 | ±2.7 | ±2.4 | |

To further assess clustering quality on the discovered unknown classes, we additionally report NMI (Vinh et al., 2010) and Purity (Manning, 2008) computed exclusively on the unknown-class samples identified by ABC, GLC, and SHOT in Figures A3, A4, A5, and A6. Since GLC and SHOT do not directly produce individual unknown-class labels, we follow the same procedure used for computing NMI on the full dataset: we use the cluster assignments obtained by applying DPM to the unknown samples detected by these methods, and then compute NMI and Purity based on these assignments. This more fine-grained analysis complements

Table A2: Mean Accuracy (%) on the Office-31 and VisDA datasets under OPDA settings.

| Methods | SF | FG | VisDA | Office-31 | | | | | | |
|---|---|---|---|---|---|---|---|---|---|---|
| | | | S2R | A2D | A2W | D2A | D2W | W2A | W2D | Avg |
| **Best Epoch** | | | | | | | | | | |
| SHOT-O (Liang et al., 2020) | ✓ | ✗ | 29.5 ±1.0 | 58.5 ±2.4 | 55.6 ±0.8 | 56.6 ±1.7 | 76.9 ±3.0 | 55.5 ±2.3 | 65.6 ±0.7 | 61.5 |
| GLC (Qu et al., 2023) | ✓ | ✗ | **70.6** ±1.9 | 81.1 ±1.5 | 84.8 ±1.3 | 88.4 ±1.5 | **92.8** ±1.2 | 87.1 ±1.3 | **95.6** ±0.0 | 88.3 |
| ABC | ✓ | ✓ | 61.5 ±1.2 | **90.0** ±0.4 | **92.8** ±1.5 | 89.3 ±0.6 | 91.3 ±0.7 | **89.6** ±0.2 | 94.0 ±1.6 | **91.2** |
| **Final Epoch** | | | | | | | | | | |
| SHOT-O (Liang et al., 2020) | ✓ | ✗ | 34.3 ±2.4 | 59.1 ±0.6 | 62.1 ±2.0 | 55.2 ±0.3 | 75.5 ±3.7 | 55.1 ±3.5 | 65.7 ±0.1 | 62.1 |
| GLC (Qu et al., 2023) | ✓ | ✗ | **73.2** ±0.9 | 80.4 ±1.0 | 88.2 ±0.7 | **90.7** ±0.2 | **93.7** ±0.1 | **90.8** ±0.3 | 92.5 ±0.3 | 89.4 |
| GBNC-only | - | - | 29.5 ±0.2 | 80.4 ±0.0 | 89.1 ±2.7 | 84.1 ±3.6 | 83.6 ±5.5 | 80.1 ±0.4 | 94.3 ±0.0 | 85.3 |
| ABC | ✓ | ✓ | 63.4 ±0.3 | **88.7** ±1.1 | **92.3** ±2.0 | 89.9 ±0.3 | 89.5 ±1.8 | 89.8 ±0.4 | **94.4** ±1.2 | **90.8** |

the standard evaluation metrics and demonstrates that ABC, as well as the DPM module when applied to different baseline methods, achieves reasonable performance in distinguishing individual unknown classes.

Overall, ABC achieves the highest average NMI in most settings: Final Epoch on Office-31, Best Epoch on Office-Home, and both settings on VisDA. Although this evaluation is conducted on the discovered unknown-class samples of each method, the results suggest that a unified Bayesian nonparametric framework that jointly detects known classes and separates unknown classes can yield better separation than performing a separate BNC step on the unknown samples produced by existing source-free universal domain adaptation methods. On Office-31 and Office-Home, ABC attains an average NMI of approximately 65% on the unknown subsets, indicating good agreement between ground-truth and estimated labels. On VisDA, the lower NMI highlights the greater difficulty of this dataset, which is consistent with its lower overall classification performance reported in the main results. For Purity, we observe a similar trend on Office-Home, where ABC achieves moderate agreement with an average value around 60%. On VisDA and Office-31, ABC attains higher average Purity scores (approximately 67% and 87%, respectively) than NMI. Since Purity emphasizes cluster homogeneity, the gap between NMI and Purity suggests that clusters are relatively homogeneous but more numerous than the true classes, meaning some true classes are split into multiple smaller clusters. A similar discrepancy between Purity and NMI is also observed when applying the BNC module, DPM, to SHOT and GLC. This behavior is consistent with the well-known tendency of Bayesian nonparametric clustering (BNC) methods to produce numerous small clusters that are often difficult to interpret (Müller et al., 2013; Miller & Harrison, 2013). Because ABC relies on a BNC module to cluster unknown samples, this over-clustering phenomenon also appears in ABC's results. In the BNC literature, it is common to apply post-processing to filter out tiny clusters and focus on larger, meaningful clusters for downstream analysis (Wang et al., 2021; Ni et al., 2020a; Guha et al., 2021). Additionally, several methods handle small clusters through cluster merging as a post-processing step; for example, SIGN (Ni et al., 2020c) merges clusters using a "clustering the clusters" strategy. In our downstream analyses (Appendices D.3 and D.4), we do not integrate this merging step, and thus our post-processing focuses solely on filtering. Incorporating merging strategies will be considered as future work.

## D.2 Runtime and Memory Analysis

The computational cost of ABC is primarily dominated by the VB procedure in the GBNC step. Assuming the truncation level $K^*$ is chosen as a multiple of the expected number of clusters $K$, each VB iteration has time complexity $\mathcal{O}(NKd^2)$, where $N$ denotes the number of samples and $d$ is the embedding dimension. In contrast, popular clustering-based domain adaptation methods (Qu et al., 2023; Li et al., 2021) typically rely on KMeans or similar centroid-based algorithms to detect unknown-class samples, with per-iteration complexity $\mathcal{O}(NKd)$. Thus, both approaches scale linearly with respect to $N$; however, VB exhibits quadratic complexity in $d$, whereas KMeans scales linearly in $d$. This difference explains why our method is generally

Table A3: NMI score (%) for discovered unknown-class samples on the Office-Home dataset under OPDA settings.

| Methods | A2C | A2P | A2R | C2A | C2P | C2R | P2A | P2C | P2R | R2A | R2C | R2P | Avg |
|---|---|---|---|---|---|---|---|---|---|---|---|---|---|
| **Best Epoch** | | | | | | | | | | | | | |
| SHOT-O (Liang et al., 2020)+ DPM | 46.6 ±0.4 | 68.8 ±0.5 | 62.8 ±0.8 | 54.0 ±0.8 | 64.6 ±0.3 | 62.0 ±1.0 | 52.9 ±1.0 | 47.2 ±0.7 | 63.7 ±1.2 | 55.4 ±0.6 | 47.2 ±0.3 | 69.7 ±0.2 | 57.9 |
| GLC (Qu et al., 2023)+ DPM | 47.7 ±0.4 | 71.9 ±0.4 | 69.1 ±0.6 | 56.6 ±0.3 | 72.3 ±0.3 | 69.5 ±0.4 | 57.4 ±0.6 | 47.5 ±0.4 | 69.0 ±0.3 | 58.1 ±1.0 | **48.7** ±0.9 | 72.0 ±0.0 | 61.7 |
| ABC | **49.6** ±0.8 | **75.6** ±0.9 | **72.1** ±1.0 | **61.9** ±1.7 | **75.2** ±1.0 | **71.2** ±0.7 | **63.6** ±0.6 | **49.6** ±0.6 | **72.4** ±1.0 | **65.1** ±0.9 | 48.0 ±0.8 | **76.8** ±0.8 | **65.1** |
| **Final Epoch** | | | | | | | | | | | | | |
| SHOT-O (Liang et al., 2020)+ DPM | 53.3 ±1.4 | 62.6 ±0.4 | 58.3 ±1.5 | 54.0 ±0.6 | 64.1 ±1.1 | 58.8 ±1.4 | 53.5 ±1.3 | 52.8 ±0.6 | 57.0 ±0.8 | 54.6 ±1.1 | 53.6 ±0.5 | 63.8 ±1.3 | 57.2 |
| GLC (Qu et al., 2023)+DPM | **72.5** ±0.7 | 72.3 ±0.7 | 69.0 ±0.4 | **77.4** ±2.0 | 72.3 ±0.3 | 69.5 ±0.4 | 58.9 ±0.5 | **66.8** ±2.0 | 68.6 ±1.5 | 59.3 ±0.3 | **65.6** ±0.6 | 71.6 ±0.0 | **68.7** |
| ABC | 49.9 ±0.3 | **75.4** ±1.1 | **71.2** ±0.9 | 61.2 ±0.7 | **74.1** ±0.6 | **70.1** ±0.9 | **64.3** ±0.8 | 49.1 ±0.2 | **72.3** ±0.5 | **63.3** ±1.2 | 48.3 ±0.8 | **75.1** ±0.5 | 64.5 |

Table A4: NMI score (%) for discovered unknown-class samples on the Office-31 and VisDA datasets under OPDA settings.

| Methods | VisDA | Office-31 | | | | | | |
|---|---|---|---|---|---|---|---|---|
| | S2R | A2D | A2W | D2A | D2W | W2A | W2D | Avg |
| **Best Epoch** | | | | | | | | |
| SHOT-O (Liang et al., 2020)+ DPM | 26.9 ±1.0 | 72.3 ±1.3 | **75.2** ±0.3 | 50.0 ±1.8 | 71.8 ±0.8 | 48.7 ±1.7 | 70.3 ±0.3 | 64.7 |
| GLC (Qu et al., 2023)+DPM | 26.1 ±0.3 | **74.4** ±0.7 | 74.0 ±0.7 | 49.9 ±0.9 | 72.1 ±0.3 | 50.3 ±1.3 | 70.2 ±0.2 | **65.2** |
| ABC | **34.8** ±0.2 | 69.0 ±0.8 | 74.2 ±0.3 | **50.9** ±0.2 | **74.7** ±0.7 | **51.0** ±0.8 | 70.6 ±3.5 | 65.1 |
| **Final Epoch** | | | | | | | | |
| SHOT-O (Liang et al., 2020)+ DPM | 25.3 ±0.2 | 68.4 ±1.2 | 70.3 ±1.1 | 47.6 ±0.6 | 70.3 ±0.6 | **50.6** ±2.5 | 67.1 ±0.7 | 62.4 |
| GLC (Qu et al., 2023)+DPM | 24.4 ±1.4 | **71.4** ±0.5 | 72.8 ±0.4 | 48.4 ±0.6 | 71.4 ±0.5 | 49.4 ±1.0 | **70.2** ±0.1 | 64.0 |
| ABC | **34.5** ±0.5 | 71.2 ±1.2 | **74.7** ±0.9 | **53.1** ±1.8 | **75.3** ±0.9 | 49.1 ±0.7 | 66.5 ±1.3 | **65.0** |

Table A5: Purity score (%) for discovered unknown-class samples on the Office-Home dataset under OPDA settings.

| Methods | A2C | A2P | A2R | C2A | C2P | C2R | P2A | P2C | P2R | R2A | R2C | R2P | Avg |
|---|---|---|---|---|---|---|---|---|---|---|---|---|---|
| **Best Epoch** | | | | | | | | | | | | | |
| SHOT-O (Liang et al., 2020)+ DPM | 36.4 ±0.1 | 69.1 ±1.4 | 61.1 ±1.8 | 43.2 ±1.4 | 61.2 ±0.4 | 58.2 ±1.7 | 42.1 ±2.2 | 36.9 ±0.8 | 61.4 ±2.3 | 45.6 ±1.0 | 36.4 ±0.3 | 69.5 ±0.5 | 51.8 |
| GLC (Qu et al., 2023)+DPM | 38.0 ±1.3 | 71.4 ±0.6 | 68.7 ±0.7 | 46.6 ±0.7 | 71.1 ±0.3 | 68.6 ±0.3 | 50.5 ±1.1 | 36.3 ±0.3 | 67.9 ±0.6 | 51.3 ±1.0 | 36.7 ±1.0 | 71.0 ±0.0 | 56.5 |
| ABC | **39.7** ±1.0 | **73.9** ±1.1 | **73.0** ±1.7 | **56.1** ±2.0 | **73.3** ±1.6 | **69.7** ±1.7 | **58.2** ±1.1 | **40.8** ±1.6 | **71.9** ±1.2 | **56.7** ±0.7 | **38.3** ±1.0 | **75.5** ±1.7 | **60.6** |
| **Final Epoch** | | | | | | | | | | | | | |
| SHOT-O (Liang et al., 2020)+ DPM | 40.8 ±1.4 | 60.2 ±1.9 | 51.3 ±3.1 | 41.3 ±0.4 | 56.0 ±1.6 | 52.6 ±3.9 | 41.1 ±1.4 | 41.1 ±0.8 | 51.8 ±1.9 | 40.8 ±1.7 | 40.5 ±0.6 | 61.2 ±1.7 | 48.2 |
| GLC (Qu et al., 2023)+DPM | **62.2** ±2.1 | 72.0 ±0.4 | 68.6 ±0.9 | **66.9** ±2.8 | 71.2 ±0.3 | **69.2** ±0.2 | 52.5 ±0.7 | **47.5** ±3.2 | 67.6 ±2.4 | 52.8 ±1.1 | **50.5** ±0.9 | 70.7 ±0.0 | **62.6** |
| ABC | 41.2 ±0.9 | **73.4** ±1.4 | **71.9** ±1.3 | 55.4 ±0.8 | **71.8** ±1.5 | 68.4 ±0.4 | **58.6** ±0.3 | 39.9 ±0.6 | **70.4** ±0.9 | **56.0** ±1.6 | 39.4 ±2.0 | **73.8** ±1.2 | 60.0 |

Table A6: Purity score (%) for discovered unknown-class samples on the Office-31 and VisDA datasets under OPDA settings.

| Methods | VisDA | Office-31 | | | | | | |
|---|---|---|---|---|---|---|---|---|
| | S2R | A2D | A2W | D2A | D2W | W2A | W2D | Avg |
| **Best Epoch** | | | | | | | | |
| SHOT-O (Liang et al., 2020)+ DPM | 57.9 ±0.9 | **99.8** ±0.4 | **98.7** ±0.4 | 66.3 ±1.4 | **98.7** ±0.4 | 65.6 ±3.2 | **100.0** ±0.0 | **88.2** |
| GLC (Qu et al., 2023)+DPM | 63.1 ±0.7 | 99.1 ±0.7 | 98.1 ±0.4 | 66.1 ±0.6 | **98.7** ±0.9 | 65.1 ±1.4 | **100.0** ±0.0 | 87.9 |
| ABC | **67.4** ±0.5 | 96.6 ±1.3 | 98.6 ±0.3 | **67.2** ±0.3 | 97.0 ±1.0 | **67.2** ±1.5 | 95.8 ±0.8 | 87.1 |
| **Final Epoch** | | | | | | | | |
| SHOT-O (Liang et al., 2020)+ DPM | 52.8 ±0.2 | **100.0** ±0.0 | 99.6 ±0.6 | 65.2 ±0.6 | 98.9 ±0.7 | **69.6** ±3.6 | **100.0** ±0.0 | **88.9** |
| GLC (Qu et al., 2023)+DPM | 62.6 ±1.5 | **100.0** ±0.0 | 98.2 ±0.2 | 66.9 ±0.7 | **99.5** ±0.5 | 67.6 ±2.9 | **100.0** ±0.0 | 88.7 |
| ABC | **66.8** ±0.4 | 98.5 ±0.8 | 99.0 ±0.2 | **70.1** ±2.3 | 97.8 ±1.1 | 66.1 ±0.6 | 96.2 ±0.4 | 87.9 |

slower than KMeans-based approaches in the runtime comparisons (Tables A7a and A7b). Nevertheless, the linear scalability with $N$ ensures that our method remains practical for large-scale datasets. For example, on the VisDA dataset with tens of thousands of target-domain images, much larger than the other two datasets, our method achieves a reasonable runtime, approximately twice that of GLC/SHOT in average time per epoch, and requires about 2.5 hours of total training time compared to 2 hours for GLC/SHOT, demonstrating good scalability in practice.

Furthermore, as discussed in the introduction, the true number of unknown clusters is not known *a priori*. Unlike VB, which automatically infers the effective number of clusters given only an upper bound $K^*$, KMeans requires a fixed pre-specified value of $K$. In practice, this often necessitates repeated clustering over multiple candidate values of $K$, resulting in additional computational overhead. If clustering is performed over $M$ candidate values, the total complexity becomes $\mathcal{O}(NKdM)$. When $M$ scales proportionally with $K$ (e.g., searching over a range such as $[K/2, 2K]$ with unit step size), the overall complexity increases to $\mathcal{O}(NK^2d)$, leading to quadratic dependence on $K$. Even with more efficient search strategies, the cumulative cost of repeated clustering can be substantial. In this sense, the proposed method may exhibit increasing computational advantages as $K$ grows or when finer granularity in estimating $K$ is required. Furthermore, the quadratic dependence on $d$ in VB arises from modeling the full precision matrix, which captures covariance structure in the embedding space, information that KMeans does not incorporate. This richer representation can improve cluster quality in practice. If computational efficiency is prioritized, the precision matrix in our method can be easily constrained to be diagonal, reducing the complexity to $\mathcal{O}(NKd)$ while retaining automatic cluster number estimation and eliminating the need for repeated clustering.

Next, we evaluate both the training time and memory consumption of ABC and compare them with two baseline SFDA methods, SHOT and GLC, which are also included in our performance comparison. Tables A7a and A7b reports both total training time (in minutes) and average time per epoch (in seconds). Despite modeling the covariance structure across features using a Dirichlet Process Gaussian Mixture Model, ABC remains computationally efficient. On Office-31 and Office-Home, entire training completes in under 10 and approximately 20 minutes on average, respectively, comparable to SHOT and GLC, a clustering-based approach using K-means. On the larger-scale VisDA-C dataset, ABC requires around 2.5 hours compared to GLC's 2 hour, demonstrating reasonable scalability despite the added modeling complexity. While ABC is slightly slower than GLC, both methods exhibit similar orders of magnitude for total and per-epoch training time, indicating that ABC achieves a favorable balance between computational cost and modeling capability. Table A7c reports peak memory usage during training. ABC consumes roughly twice the memory of SHOT and GLC, likely due to calculations involving the precision matrix. However, memory usage remains comparable to baseline methods and scales similarly across datasets of different sizes, suggesting that ABC also exhibits good scalability in terms of memory. It is worth noting that GLC accelerates several of its KMeans steps using the FAISS library(Johnson et al., 2019), a highly optimized implementation with GPU acceleration and vectorized operations. In contrast, the iterative updates of our VB procedure are

Table A7: Runtime and memory comparison under OPDA settings.

(a) Runtime comparison: Office-Home

| Methods | A2C | A2P | A2R | C2A | C2P | C2R | P2A | P2C | P2R | R2A | R2C | R2P | Avg |
|---|---|---|---|---|---|---|---|---|---|---|---|---|---|
| **Total Training Time (min)** | | | | | | | | | | | | | |
| SHOT-O (Liang et al., 2020) | ~14.5 | ~15.5 | ~51.3 | ~12.8 | ~15.6 | ~52.0 | ~13.2 | ~15.6 | ~52.1 | ~12.9 | ~15.5 | ~13.4 | ~23.7 |
| GLC (Qu et al., 2023) | ~25.5 | ~27.4 | ~25.1 | ~11.1 | ~20.5 | ~21.7 | ~12.3 | ~18.1 | ~21.7 | ~12.3 | ~18.0 | ~20.4 | ~19.5 |
| ABC | ~17.2 | ~17.1 | ~35.5 | ~12.5 | ~17.0 | ~35.5 | ~12.6 | ~17.1 | ~35.8 | ~12.5 | ~17.3 | ~17.2 | ~20.6 |
| **Average Time per Epoch (s)** | | | | | | | | | | | | | |
| SHOT-O (Liang et al., 2020) | ~58 | ~62 | ~205 | ~51 | ~62 | ~208 | ~53 | ~62 | ~208 | ~52 | ~62 | ~54 | ~95 |
| GLC (Qu et al., 2023) | ~31 | ~33 | ~30 | ~13 | ~25 | ~26 | ~15 | ~22 | ~26 | ~15 | ~22 | ~24 | ~23 |
| ABC | ~103 | ~103 | ~213 | ~75 | ~102 | ~213 | ~75 | ~103 | ~214 | ~75 | ~104 | ~103 | ~123 |

(b) Runtime comparison: Office-31 and VisDA

| Methods | VisDA | Office-31 | | | | | | |
|---|---|---|---|---|---|---|---|---|
| | S2R | A2D | A2W | D2A | D2W | W2A | W2D | Avg |
| **Total Training Time (min)** | | | | | | | | |
| SHOT-O (Liang et al., 2020) | ~121.3 | ~4.2 | ~3.3 | ~7.9 | ~3.5 | ~7.8 | ~4.2 | ~5.2 |
| GLC (Qu et al., 2023) | ~117.6 | ~4.5 | ~5.8 | ~15.2 | ~5.8 | ~14.4 | ~4.4 | ~8.4 |
| ABC | ~146.0 | ~3.7 | ~3.8 | ~8.3 | ~3.9 | ~8.4 | ~3.7 | ~5.3 |
| **Average Time per Epoch (s)** | | | | | | | | |
| SHOT-O (Liang et al., 2020) | ~485 | ~17 | ~13 | ~32 | ~14 | ~31 | ~17 | ~21 |
| GLC (Qu et al., 2023) | ~353 | ~5 | ~7 | ~18 | ~7 | ~17 | ~5 | ~10 |
| ABC | ~875 | ~22 | ~23 | ~50 | ~23 | ~50 | ~22 | ~32 |

(c) Memory comparison

| Methods | Office-31 | Office-Home | VisDA |
|---|---|---|---|
| **Peak Memory (GB)** | | | |
| SHOT-O (Liang et al., 2020) | ~1.7 | ~1.8 | ~2.5 |
| GLC (Qu et al., 2023) | ~1.9 | ~2.7 | ~2.1 |
| ABC | ~4.2 | ~4.3 | ~4.9 |

implemented in standard Python without specialized acceleration. Therefore, ABC's runtime could be further reduced through optimized implementations leveraging high-performance computing techniques such as GPU parallelization or optimized linear algebra libraries.

### D.3 Finding the Number of Estimated Clusters

As part of our downstream analysis, we report the estimated number $K_{unk}$ of unknown clusters and compare it with the true value. Figure A1 shows this comparison, with results reported separately for the best epoch and the final epoch. As discussed in Appendix D.1, BNC models tend to generate many tiny clusters, and thus post-processing steps to filter out such clusters are commonly adopted. Following the same practice as in (Wang et al., 2021), in Figure A1, we focus only on the major clusters with $> 1\%$ of the total images when counting $K_{unk}$. As shown in Figure A1, for Office-31 and VisDA, the estimated number of unknown clusters closely matches the ground truth, indicating that ABC reliably estimates $K_{unk}$ across multiple transfer tasks on Office-31 and on the VisDA transfer task. In contrast, the number of clusters is underestimated for Office-Home. While estimating $K_{unk}$ without prior knowledge is inherently challenging, ABC produces accurate estimates in some tasks and generally provides a reasonable approximation across datasets.

### D.4 Visualization of Discovered Unknown Clusters

In this section, we present t-SNE visualizations of discovered unknown clusters based on their learned embeddings. The t-SNE algorithm produces a low-dimensional representation that preserves local structure by maintaining the relative proximity of similar points from the original high-dimensional embedding space (Van der Maaten & Hinton, 2008). Specifically, we apply t-SNE to obtain a nonlinear two-dimensional mapping for visualizing the sample distribution of the major unknown clusters described in Appendix D.3. The resulting visualizations are shown in Figures A2 and A3, corresponding to the Best Epoch and Final Epoch, respectively. In each figure, individual points are colored according to their estimated cluster labels. Across most settings, the visualizations exhibit well-separated groups that align closely with the estimated labels, indicating that ABC effectively partitions samples from discovered unknown classes and supports fine-grained discrimination.

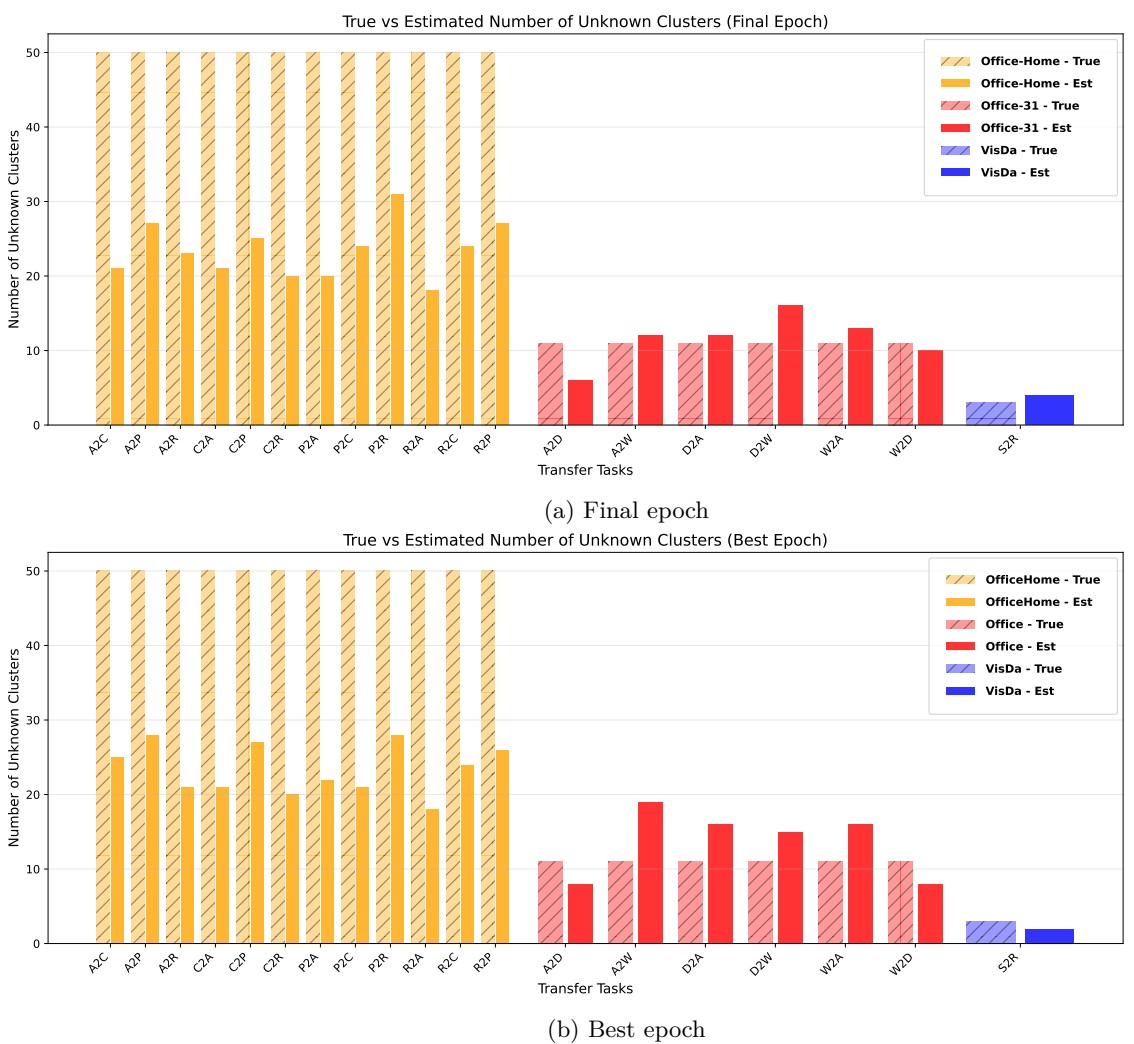

(a) Final epoch

(b) Best epoch

Figure A1: Evaluation of Unknown Cluster Number Estimation. The hatched bars represent the true number of unknown clusters, while the solid bars represent the estimated number.

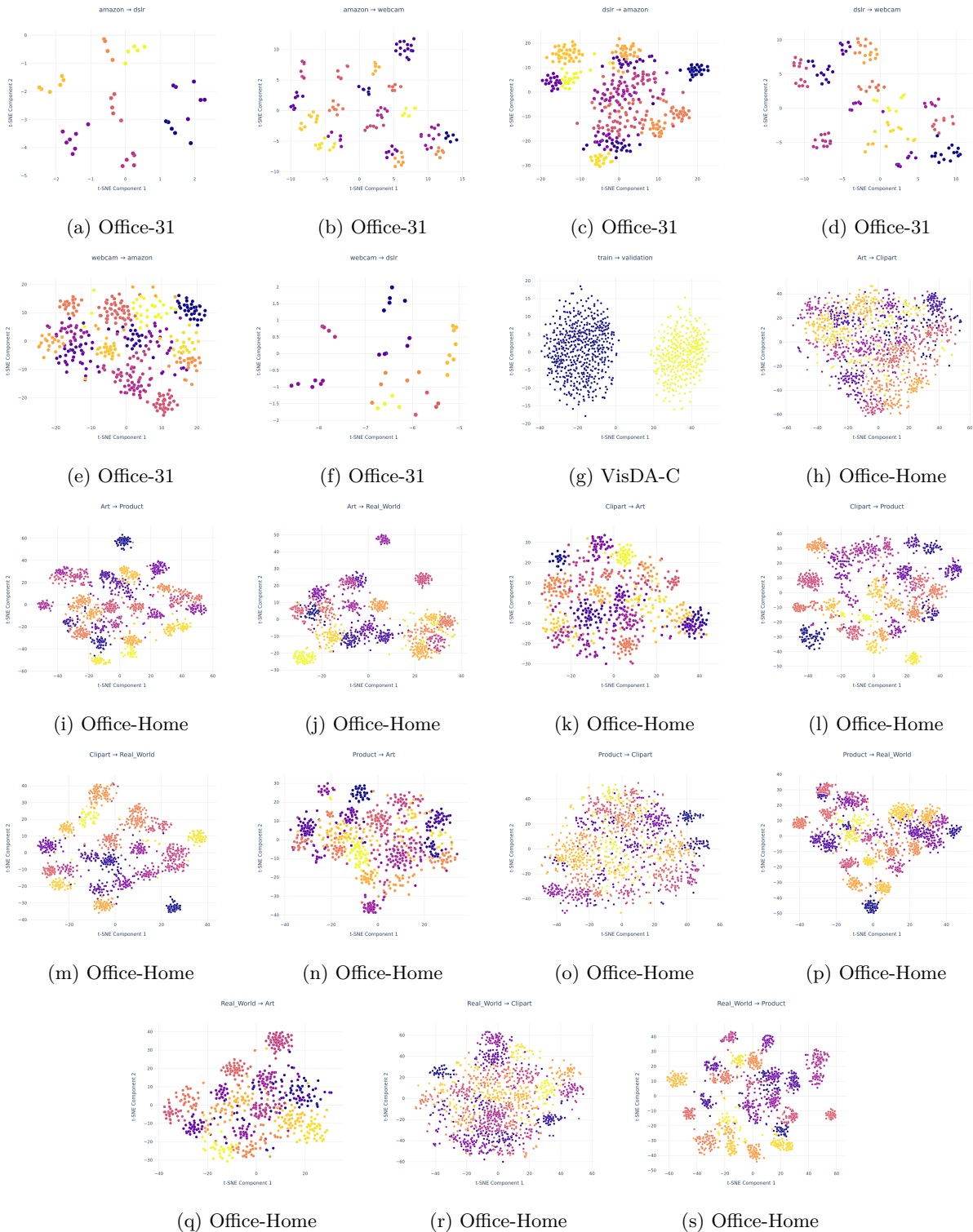

Figure A2: t-SNE visualization of discovered unknown classes (Best Epoch). Each point is colored according to its predicted cluster label.

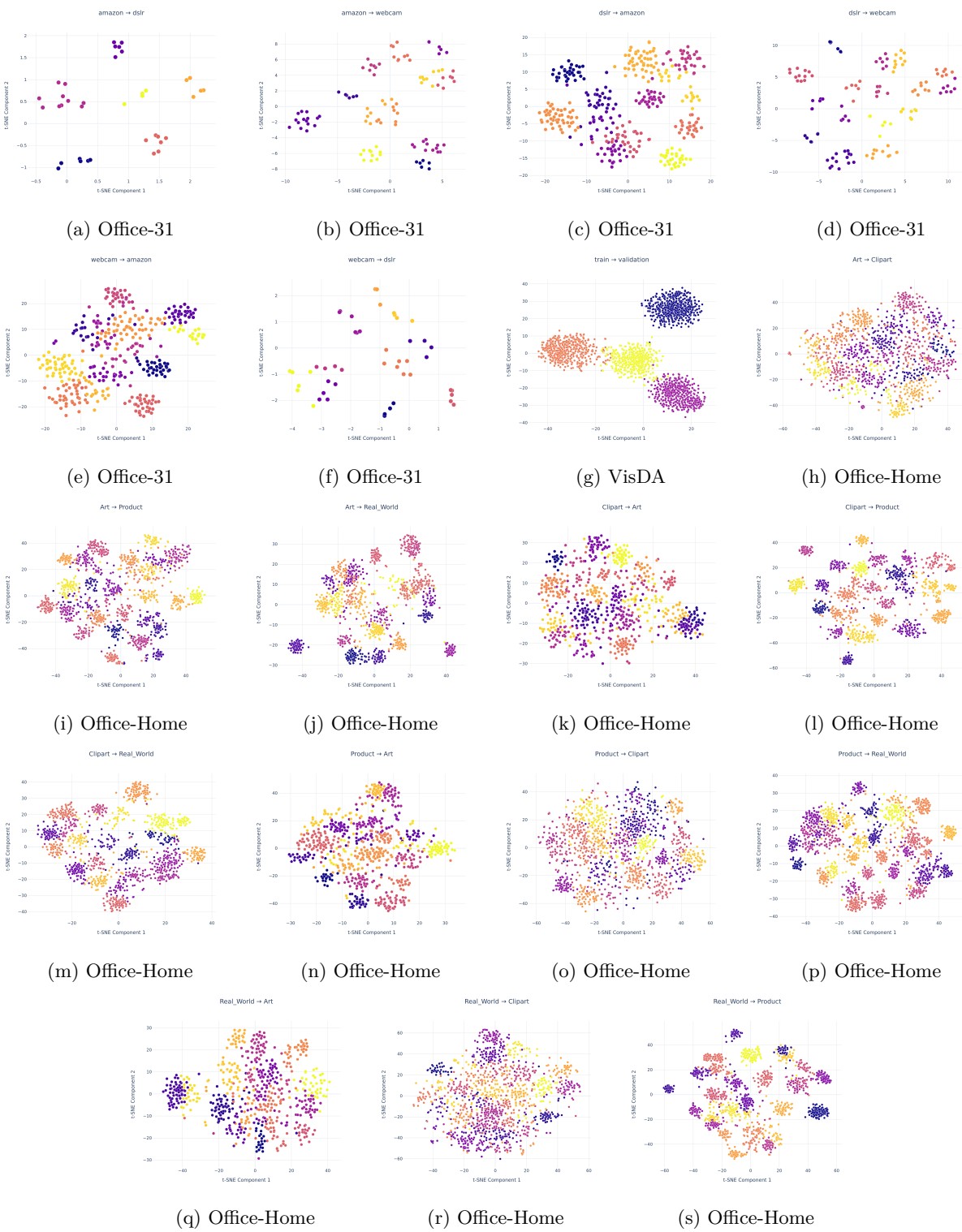

Figure A3: t-SNE visualization of discovered unknown classes (Final Epoch). Each point is colored according to its predicted cluster label.

## D.5 Results for OSDA

In this section, we present results under the OSDA setting for the Office-31 dataset, reporting H-score, mean accuracy, and NMI in Tables A8, A9, and A10, respectively. We follow the standard experimental split used in prior work (Qu et al., 2023), with 10 shared, 0 source-private, and 11 target-private classes. The results are consistent with our observations from the OPDA settings, with our method achieving solid performance across metrics. Notably, SHOT shows significantly improved performance under the OSDA setting, achieving the highest average mean accuracy.

Table A8: H-score (%) on the Office-31 dataset under OSDA settings.

| Methods | SF | FG | Office-31 | | | | | | |
|---|---|---|---|---|---|---|---|---|---|
| | | | A2D | A2W | D2A | D2W | W2A | W2D | Avg |
| OSBP (Saito et al., 2018) | ✗ | ✗ | 82.4 | 82.7 | 75.1 | 97.2 | 73.7 | 91.1 | 83.7 |
| ROS (Bucci et al., 2020) | ✗ | ✗ | 82.4 | 82.1 | 77.9 | 96.0 | 77.2 | **99.7** | 85.9 |
| CMU (Fu et al., 2020) | ✗ | ✗ | 52.6 | 55.7 | 76.5 | 75.9 | 65.8 | 64.7 | 65.2 |
| DANCE (Saito et al., 2020) | ✗ | ✗ | 84.9 | 78.8 | 79.1 | 78.8 | 68.3 | 88.9 | 79.8 |
| DCC (Li et al., 2021) | ✗ | ✗ | 58.3 | 54.8 | 67.2 | 89.4 | 85.3 | 80.9 | 72.6 |
| OVANet (Saito & Saenko, 2021) | ✗ | ✗ | **90.5** | **88.3** | **86.7** | **98.2** | **88.3** | 98.4 | **91.7** |
| GATE (Chen et al., 2022) | ✗ | ✗ | 88.4 | 86.5 | 84.2 | 95.0 | 86.1 | 96.7 | 89.5 |
| **Best Epoch (Best H-score)** | | | | | | | | | |
| SHOT-O (Liang et al., 2020) | ✓ | ✗ | 89.1 ±0.6 | 86.4 ±1.5 | 82.7 ±1.1 | 89.2 ±1.2 | 74.3 ±4.5 | 86.0 ±0.6 | 84.6 |
| GLC (Qu et al., 2023) | ✓ | ✗ | 83.2 ±0.3 | 76.7 ±0.3 | **92.5** ±0.4 | **95.3** ±0.4 | **92.9** ±0.1 | 86.0 ±0.3 | 87.8 |
| ABC | ✓ | ✓ | **90.1** ±0.8 | **90.5** ±0.6 | 91.4 ±1.1 | 91.0 ±1.1 | 91.2 ±1.7 | **95.4** ±1.3 | **91.6** |
| **Final Epoch** | | | | | | | | | |
| SHOT-O (Liang et al., 2020) | ✓ | ✗ | 85.4 ±1.0 | 82.0 ±3.1 | 68.1 ±3.0 | 78.4 ±1.6 | 66.8 ±3.0 | 85.7 ±0.5 | 77.7 |
| GLC (Qu et al., 2023) | ✓ | ✗ | 82.2 ±0.3 | 75.4 ±0.5 | **92.4** ±0.4 | **95.3** ±0.4 | **92.5** ±0.2 | 84.1 ±0.2 | 87.0 |
| GBNC-only | - | - | **89.3** ±0.0 | 87.9 ±1.9 | 87.7 ±5.0 | 89.3 ±1.6 | 83.9 ±4.9 | **94.1** ±0.7 | **88.7** |
| ABC | ✓ | ✓ | 87.0 ±1.2 | **88.9** ±0.9 | 88.4 ±3.0 | 87.3 ±0.9 | 89.5 ±0.6 | 91.2 ±1.0 | **88.7** |

Table A9: Mean Accuracy (%) on the Office-31 dataset under OSDA settings.

| Methods | SF | FG | Office-31 | | | | | | |
|---|---|---|---|---|---|---|---|---|---|
| | | | A2D | A2W | D2A | D2W | W2A | W2D | Avg |
| **Best Epoch** | | | | | | | | | |
| SHOT-O (Liang et al., 2020) | ✓ | ✗ | 89.9 ±0.7 | **90.9** ±1.9 | 87.1 ±3.2 | 91.7 ±1.6 | 74.6 ±5.0 | 85.6 ±0.8 | 86.6 |
| GLC (Qu et al., 2023) | ✓ | ✗ | 76.9 ±0.4 | 67.7 ±0.3 | **92.0** ±0.3 | **94.2** ±0.2 | **91.3** ±0.2 | 77.7 ±0.4 | 83.3 |
| ABC | ✓ | ✓ | **91.2** ±0.3 | 89.4 ±1.6 | 88.9 ±1.2 | 85.1 ±1.8 | 90.4 ±0.4 | **92.7** ±1.8 | **89.6** |
| **Final Epoch** | | | | | | | | | |
| SHOT-O (Liang et al., 2020) | ✓ | ✗ | 92.7 ±1.1 | **94.3** ±0.5 | 90.2 ±0.4 | **94.2** ±0.2 | 77.2 ±4.0 | 86.7 ±0.2 | **89.2** |
| GLC (Qu et al., 2023) | ✓ | ✗ | 74.8 ±0.6 | 65.1 ±0.6 | **92.0** ±0.2 | **94.2** ±0.2 | **91.5** ±0.2 | 75.0 ±0.2 | 82.1 |
| GBNC-only | - | - | **93.1** ±0.0 | 83.9 ±3.7 | 82.2 ±7.8 | 83.5 ±2.6 | 80.4 ±10.4 | 92.2 ±1.2 | 85.9 |
| ABC | ✓ | ✓ | 87.3 ±0.3 | 90.0 ±1.9 | 87.0 ±5.1 | 81.9 ±1.3 | 89.7 ±0.8 | **92.8** ±1.5 | 88.1 |

Table A10: NMI (%) on the Office-31 dataset under OSDA settings.

| Methods | Office-31 | | | | | | |
|---|---|---|---|---|---|---|---|
| | A2D | A2W | D2A | D2W | W2A | W2D | Avg |
| **Best Epoch** | | | | | | | |
| SHOT-O (Liang et al., 2020) | **80.5** | 81.1 | **70.5** | 81.0 | 68.8 | 80.0 | 77.0 |
| + DPM | ±0.2 | ±0.8 | ±0.9 | ±0.2 | ±0.6 | ±0.3 | |
| GLC (Qu et al., 2023) + DPM | 79.2 | 78.6 | 69.0 | 83.3 | 69.0 | 80.6 | 76.6 |
| | ±0.5 | ±0.9 | ±0.2 | ±0.1 | ±0.3 | ±0.1 | |
| ABC | 80.4 | **83.7** | 70.1 | **84.3** | **70.5** | **82.4** | **78.6** |
| | ±0.5 | ±0.4 | ±0.3 | ±0.3 | ±1.3 | ±0.5 | |
| **Final Epoch** | | | | | | | |
| SHOT-O (Liang et al., 2020) | **80.7** | 81.4 | **69.9** | 79.8 | 68.2 | 79.9 | 76.7 |
| + DPM | ±0.7 | ±0.7 | ±0.3 | ±0.6 | ±0.6 | ±0.1 | |
| GLC (Qu et al., 2023) + DPM | 79.4 | 79.1 | 69.0 | 83.3 | 68.8 | 80.3 | 76.7 |
| | ±0.4 | ±0.3 | ±0.4 | ±0.1 | ±0.3 | ±0.3 | |
| GBNC-only | 79.8 | 82.5 | 69.6 | **83.6** | **69.6** | **81.8** | 77.8 |
| | ±0.0 | ±0.6 | ±0.6 | ±0.4 | ±0.9 | ±0.1 | |
| ABC | 80.1 | **83.4** | 69.7 | 83.3 | 69.5 | **81.8** | **78.0** |
| | ±0.4 | ±0.5 | ±0.2 | ±0.8 | ±0.4 | ±0.7 | |

## D.6 Additional Ablation Study and Sensitivity Analysis Results

In this section, we present supplementary figures for the ablation studies and sensitivity analyses discussed in the main text. We first show a comparison between the Beta mixture detector and the OOD detectors in terms of NMI and mean accuracy for our ablation study (Figure A4). The results based on NMI lead to conclusions consistent with those obtained using HOS in the main text, whereas the mean accuracy results reveal a slight difference: on Office-Home, the OOD detectors achieve marginally better performance across transfer tasks. This difference is expected, as the OOD detectors tend to retain more samples as known classes, while mean accuracy places less emphasis on unknown-class performance by treating the aggregated unknown class as a single category, equivalent to each individual known class.

We further provide supplementary figures and extended discussion on the sensitivity analysis of key hyperparameters in our model under the OPDA setting, including $\lambda_{\text{loc}}$, $\alpha$, $K_{nn}$, and $\delta$. These results are visualized across multiple figures (Figures A5–A10) and cover metrics such as H-score, NMI, and mean accuracy at the Final Epoch on the Office-31 and Office-Home datasets.

To provide a comprehensive view of hyperparameter effects, we present two complementary visualization formats. Specifically, line plots (Figures A5–A7) illustrate per-task performance variations as each hyperparameter changes, highlighting the response patterns across different domain adaptation scenarios. In contrast, bar plots (Figures A8– A10) summarize the average performance across all tasks, offering a high-level overview of model robustness.

Overall, the results show that the performance of our model remains stable across a broad range of hyperparameter values, with only moderate sensitivity observed in specific cases. For instance, the performance (measured by NMI and H-score) remains stable when varying $\alpha$, while it is slightly more sensitive to the choice of $K_{nn}$. The results validate our hyperparameter choices and indicate that ABC performs well without extensive tuning, making it practical for real-world deployment.

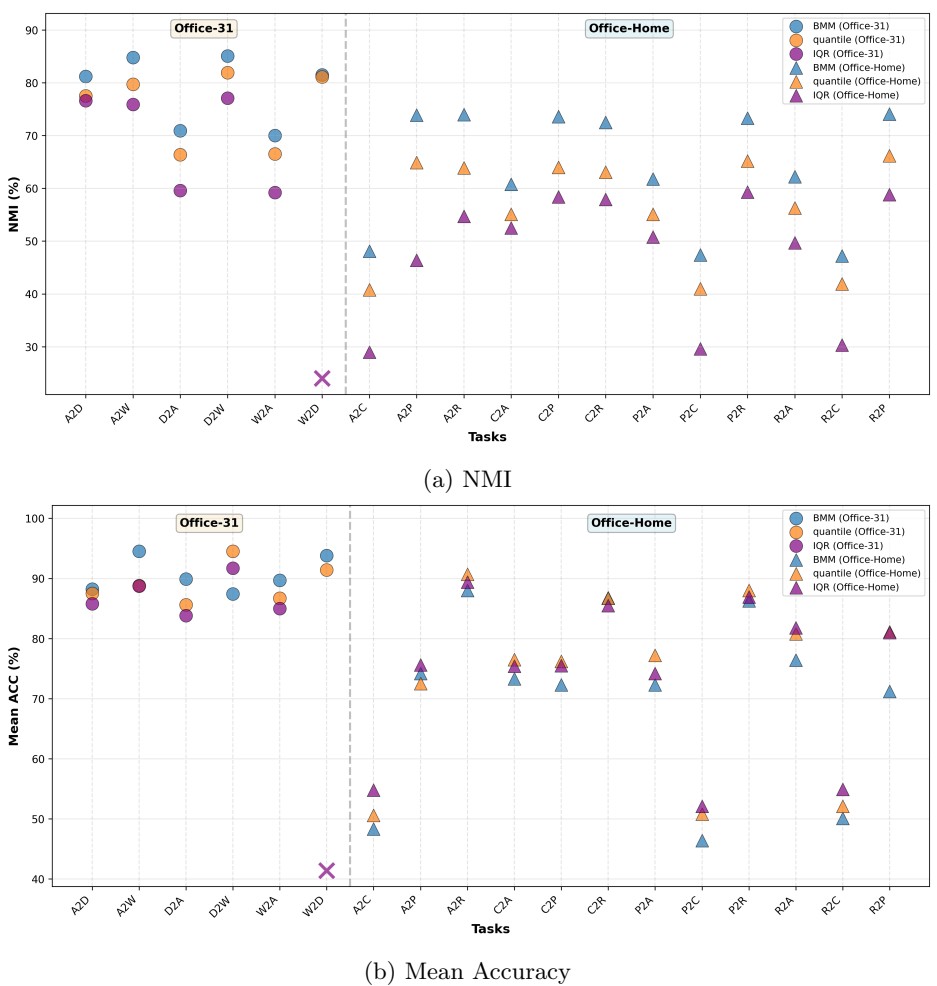

(a) NMI

(b) Mean Accuracy

Figure A4: Beta mixture detector vs. OOD detectors

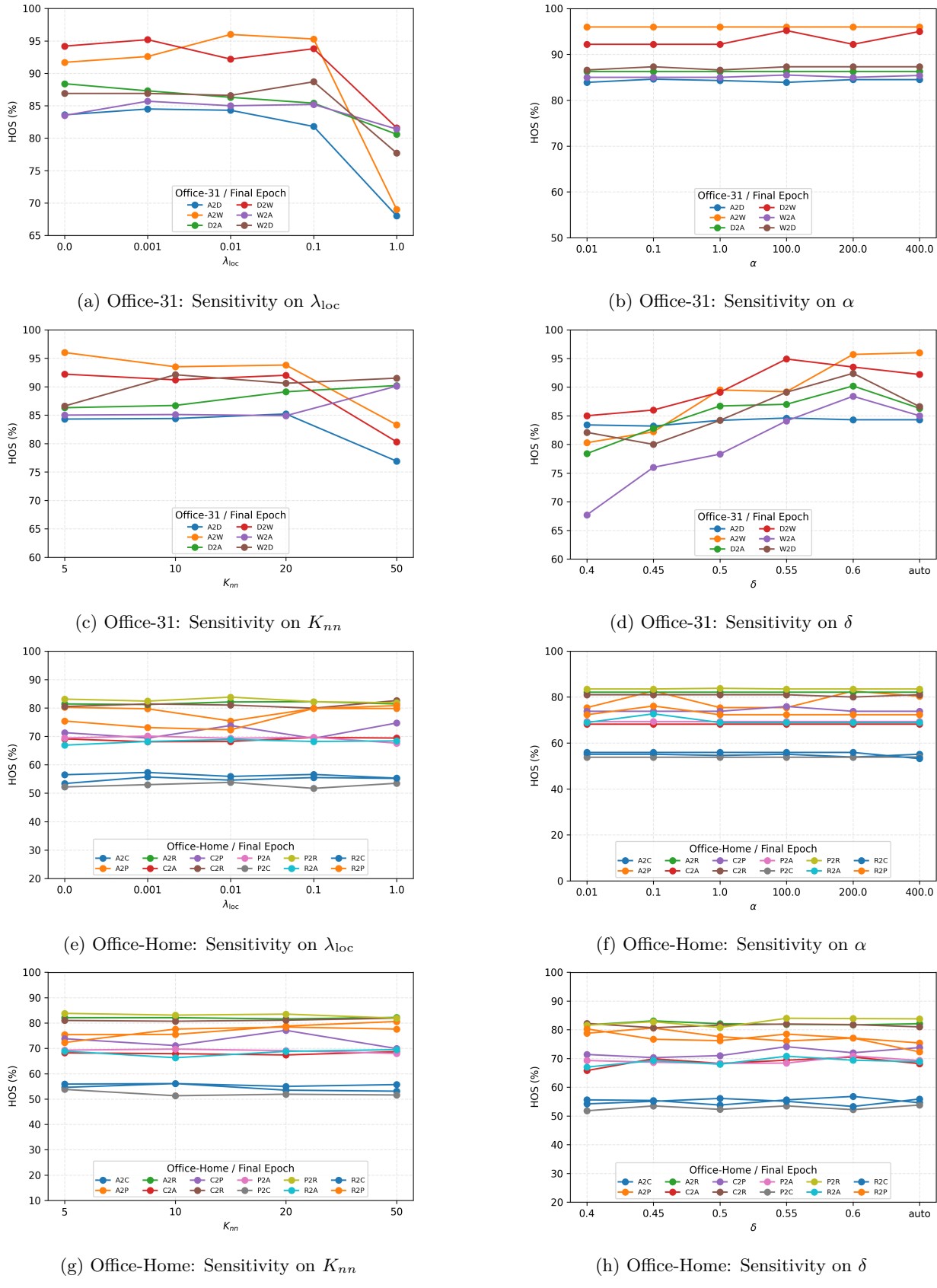

(a) Office-31: Sensitivity on $\lambda_{\text{loc}}$      (b) Office-31: Sensitivity on $\alpha$

(c) Office-31: Sensitivity on $K_{nn}$      (d) Office-31: Sensitivity on $\delta$

(e) Office-Home: Sensitivity on $\lambda_{\text{loc}}$      (f) Office-Home: Sensitivity on $\alpha$

(g) Office-Home: Sensitivity on $K_{nn}$      (h) Office-Home: Sensitivity on $\delta$

Figure A5: Hyperparameter sensitivity analysis per task (H-score) of $\lambda_{\text{loc}}$, $\alpha$, $K_{nn}$, and $\delta$ on Office-31 and Office-Home under the OPDA setting.

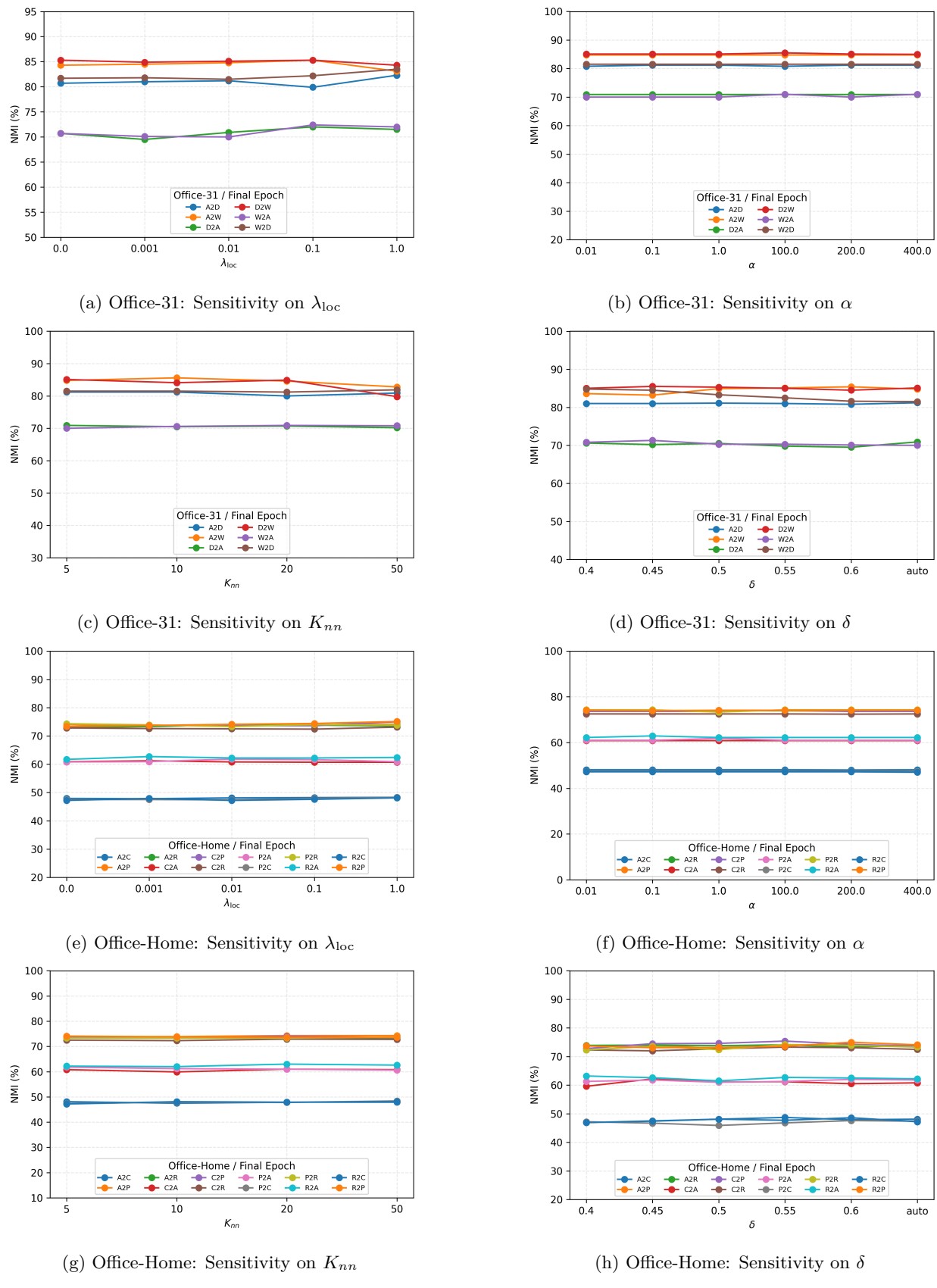

Figure A6: Hyperparameter sensitivity analysis per task (NMI) of $\lambda_{\text{loc}}$, $\alpha$, $K_{nn}$, and $\delta$ on Office-31 and Office-Home under the OPDA setting.

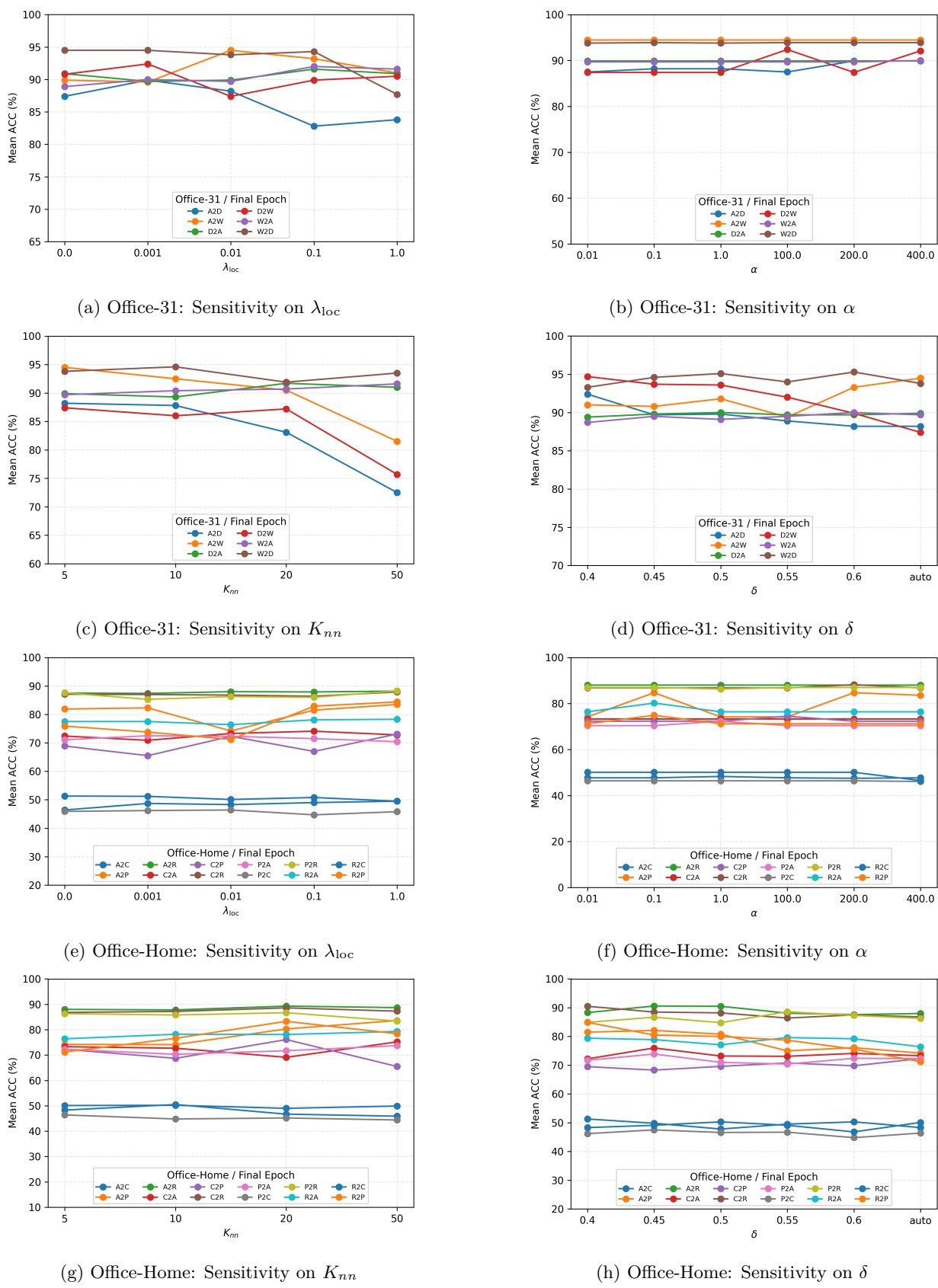

Figure A7: Hyperparameter sensitivity analysis per task (Mean Accuracy) of $\lambda_{\mathrm{loc}}$, $\alpha$, $K_{nn}$, and $\delta$ on Office-31 and Office-Home under the OPDA setting.

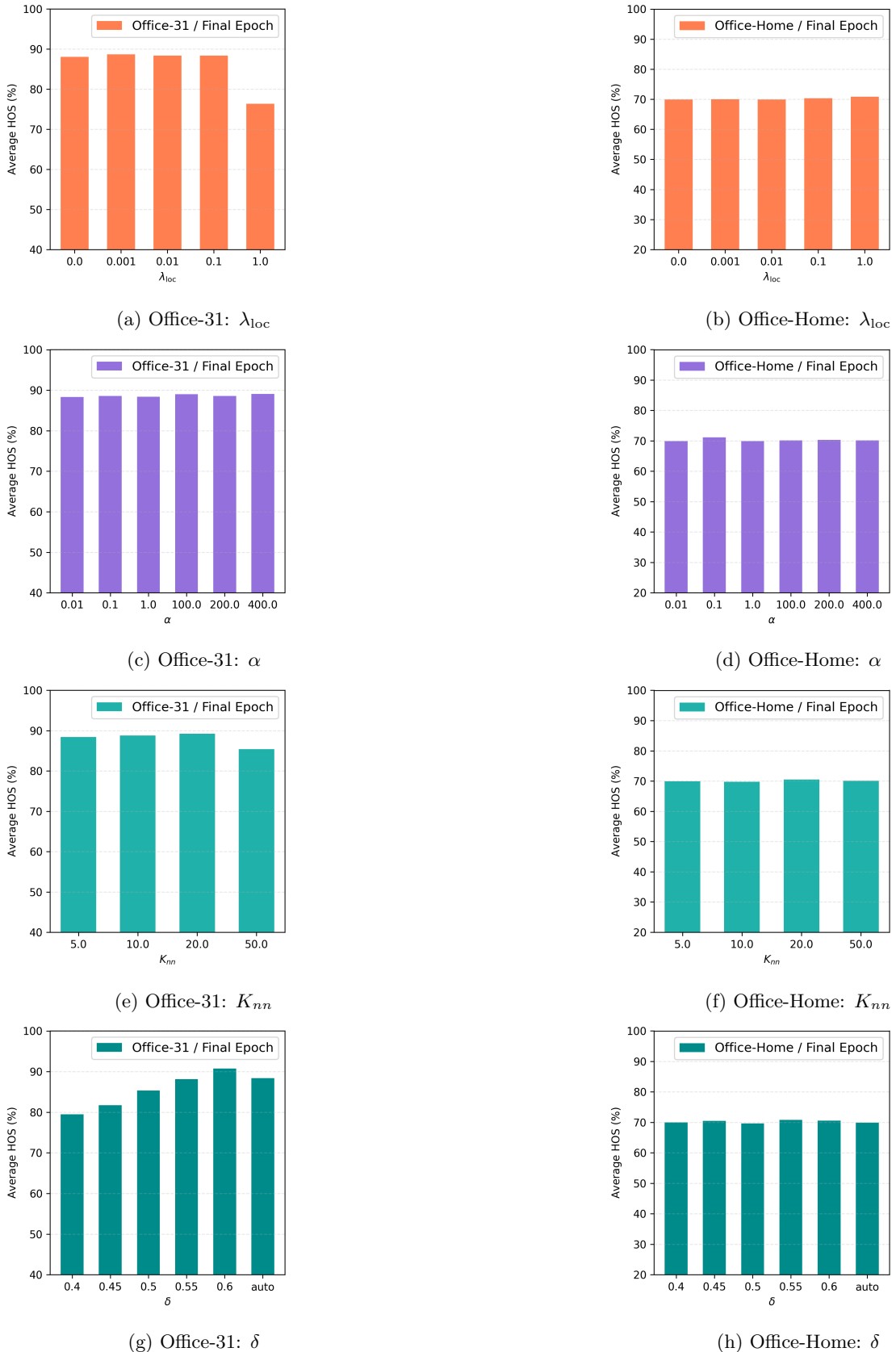

Figure A8: Hyperparameter sensitivity (H-score) of $\lambda_{\text{loc}}$, $\alpha$, $K_{nn}$, and $\delta$ on Office-31 and Office-Home under OPDA.

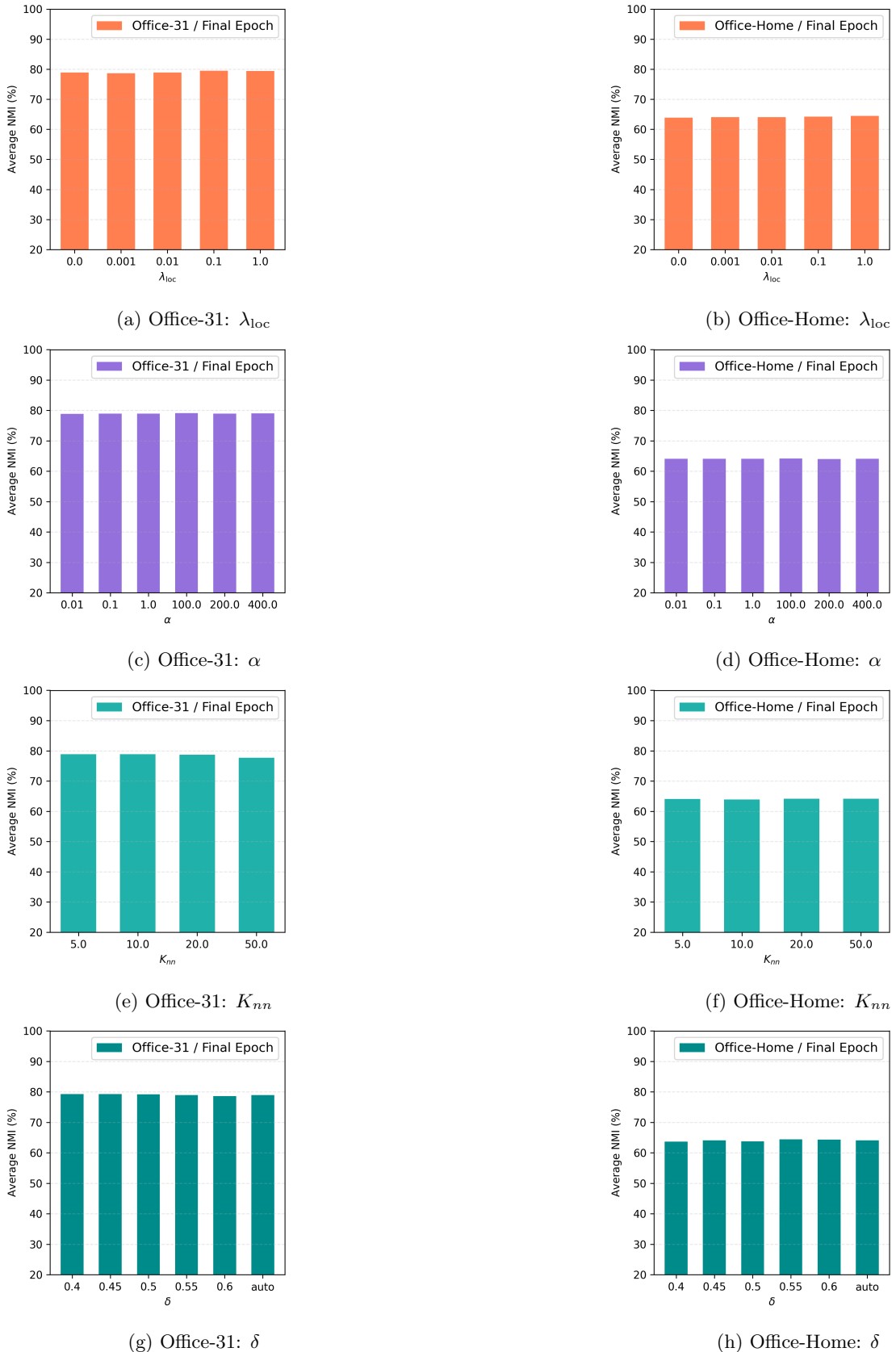

Figure A9: Hyperparameter sensitivity (NMI) of $\lambda_{\mathrm{loc}}$, $\alpha$, $K_{nn}$, and $\delta$ on Office-31 and Office-Home under OPDA.

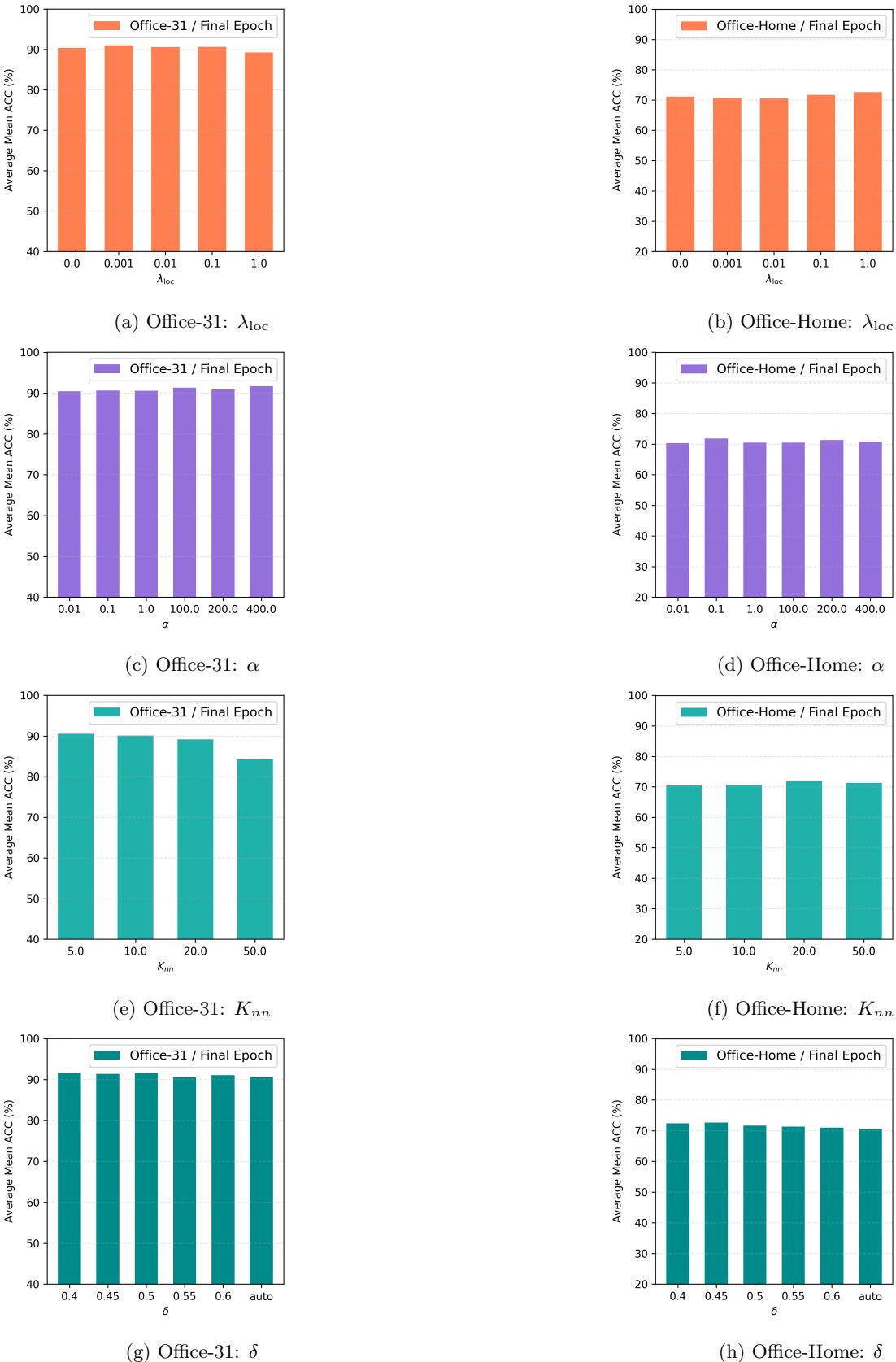

Figure A10: Hyperparameter sensitivity (Mean Accuracy) of $\lambda_{\text{loc}}$, $\alpha$, $K_{nn}$, and $\delta$ on Office-31 and Office-Home under OPDA.

