# OpenReview forum: "Adapt via Bayesian Nonparametric Clustering: Fine-Grained Classification for Model Recycling Under Domain and Category Shift"
_TMLR — Accepted by TMLR_

### Review · Reviewer_wERe · 2025-12-21

**Summary Of Contributions:**

This paper addresses Source-Free Domain Adaptation (SFDA) under category shift, where the target domain contains classes not present in the source domain. The authors propose Adapt via Bayesian Nonparametric Clustering (ABC), a framework that moves beyond treating all unknown classes as a single group. Instead, it performs fine-grained classification of unknown target classes by automatically discovering their latent structure.

**Additional Comments:**

N/A

**Audience:**

Yes

**Audience Explanation:**

The paper addresses the highly relevant and practical problem of Source-Free Domain Adaptation, which is crucial for privacy-preserving machine learning. By introducing fine-grained classification for unknown classes, it pushes the boundaries of standard "unknown detection" and opens new avenues for open-set recognition and deep clustering. Researchers in domain adaptation, transfer learning, and Bayesian nonparametrics would find these results significant.

**Broader Impact Concerns:**

There are no significant ethical concerns. The paper focuses on Source-Free adaptation, which inherently promotes privacy by avoiding the need for raw source data during the adaptation process. The potential applications mentioned (medical imaging, robotics, security) are beneficial, though as with any classification model, users should be cautious about biases in the pretrained source model.

**Claims And Evidence:**

Yes

**Claims Explanation:**

The authors provide extensive experimental evidence across three standard benchmarks: Office-31, Office-Home, and VisDA. They compare ABC against both SFDA and non-SFDA baselines using multiple metrics (H-score, mean accuracy, and NMI). The inclusion of ablation studies and hyperparameter sensitivity analyses demonstrates that the performance gains are due to the proposed components and are robust to parameter settings.

**Requested Changes:**

1. Clarification on Scalability: Provide a more detailed discussion or complexity analysis regarding the Variational Bayes inference compared to simpler SFDA methods, especially for the VisDA dataset where performance was slightly lower than the top baseline.

2. Visualizations of Discovered Clusters: Including t-SNE visualizations of the discovered unknown classes (the fine-grained clusters) would more intuitively demonstrate the "fine-grained" claim.

3. Comparison of Computational Time: A table comparing the per-epoch training time of ABC versus SHOT/GLC would help quantify the "cost" of the nonparametric clustering.

---

> ### Author Response · Authors · 2026-02-20
>
> We sincerely thank the reviewer for the insightful and constructive comments. We have carefully revised the manuscript to incorporate the requested analyses. The main changes in this revised version are highlighted in blue.
>
> **Requested Changes:**
>
> > Clarification on Scalability: Provide a more detailed discussion or complexity analysis regarding the Variational Bayes inference compared to simpler SFDA methods, especially for the VisDA dataset where performance was slightly lower than the top baseline.
>
> **R:** We appreciate the reviewer’s valuable suggestion. In the revised manuscript, we provide a thorough discussion and detailed complexity analysis of our Variational Bayes inference, comparing its scalability with other clustering-based domain adaptation methods. Its time complexity scales linearly with the number of samples, making it practical for large-scale datasets such as VisDA, which we specifically discuss in the revised manuscript to show that our method achieves reasonable runtime despite its larger size. All related analysis and discussion are included in Appendix D.2 (Pages 26-28).
>
> > Visualizations of Discovered Clusters: Including t-SNE visualizations of the discovered unknown classes (the fine-grained clusters) would more intuitively demonstrate the "fine-grained" claim.
>
>
> **R:** We thank the reviewer for the suggestion. In the revised manuscript, we include t-SNE visualizations of the discovered unknown classes across all tasks and datasets in Appendix D.4 (Page 29), showing the fine-grained clusters produced by our method, accompanied by a detailed discussion.
>
> > Comparison of Computational Time: A table comparing the per-epoch training time of ABC versus SHOT/GLC would help quantify the "cost" of the nonparametric clustering.
>
> **R:** In the revised manuscript, we provide Table A7, which compares both per-epoch training time and total running time of ABC versus SHOT/GLC, along with a detailed discussion of the comparison and the scalability of our method in Appendix D.2 (Pages 28–29). In addition to runtime, Table A7 also presents a comparison of memory usage between ABC and SHOT/GLC, further quantifying the computational cost of our method.

---

### Review · Reviewer_B4mX · 2026-01-20

**Summary Of Contributions:**

This paper introduces a unified framework for source-free domain adaptation (SFDA) that leverages Bayesian nonparametric clustering (specifically, Dirichlet Process Mixtures with variational Bayes inference) to discover latent structure within unknown classes. The authors argue that treating all unknown classes as a single category is limiting, and propose integrating clustering and model adaptation in a joint optimization framework. Key strengths include the unified design (which minimizes error propagation between clustering and adaptation), a thorough sensitivity analysis, and a principled theoretical foundation. However, the work does not convincingly demonstrate technical advantages over recent state-of-the-art methods, and the evaluation lacks comparison with the latest USDA benchmarks. The related work section also does not sufficiently position this approach relative to other clustering-based SFDA methods.

**Audience:**

Yes

**Audience Explanation:**

The problem of unknown class discovery in source-free domain adaptation is of practical and theoretical interest to the TMLR audience. The integration of Bayesian nonparametric clustering with model adaptation is a reasonable incremental step, and the sensitivity analysis may provide useful insights for researchers working on related problems. However, the impact of the work is currently limited by the lack of compelling empirical results and insufficient comparison with recent advances.

**Broader Impact Concerns:**

I do not have significant concerns regarding the broader impact or ethical implications of this work. The paper addresses a methodological advance in domain adaptation and does not raise obvious ethical or societal risks. The authors may wish to briefly discuss potential misuse or limitations in the Broader Impact Statement.

**Claims And Evidence:**

No

**Claims Explanation:**

While the paper provides a principled approach and some empirical evidence, the claims regarding the superiority of the proposed method are not fully supported. The evaluation is limited to comparisons with older baselines (2020–2023), omitting recent state-of-the-art methods such as those reported in "[P1] Revisiting Source-Free Domain Adaptation: Insights into Representativeness, Generalization, and Variety" (CVPR 2025), which achieves higher performance on standard benchmarks (e.g., 74.1% on Office-Home with ResNet-50, compared to 72.9%/69.9% for the proposed method). Furthermore, the paper does not provide compelling experimental scenarios where the proposed technique clearly outperforms prior arts, nor does it address or introduce challenging settings (e.g., Class-Incremental Domain Adaptation, ECCV 2020) where its benefits would be more apparent.

**Requested Changes:**

Critical:
1. Provide comprehensive experimental comparisons with recent state-of-the-art SFDA methods, especially those reported in "[P1] Revisiting Source-Free Domain Adaptation: Insights into Representativeness, Generalization, and Variety" (CVPR 2025) and other 2024–2025 works. Clearly state where the proposed method stands relative to these.
2. Demonstrate use cases or experimental scenarios (e.g., class-incremental domain adaptation) where the proposed technique offers clear advantages over prior arts, or provide a discussion if such cases do not exist.
3. Enhance the related work section to explicitly position this approach against other clustering-based SFDA methods, such as "Universal Source-Free Domain Adaptation" and "Domain Consensus Clustering for Universal Domain Adaptation," clarifying the unique contributions and differences.

Would Strengthen the Work:
1. Clarify the rationale for reporting both "best epoch" and "final epoch" results, and discuss the stability/convergence of the method.
2. Consider moving part of the theoretical contributions (e.g., variational Bayes inference) from the appendix to the main text if they represent a good theretical contribution.

---

> ### Author Response · Authors · 2026-02-20
>
> We appreciate the reviewer’s thorough and helpful feedback. We have carefully responded to each point, including the issues raised about weaknesses, and have revised the manuscript accordingly. The main changes in this revised version are highlighted in blue.
>
> > Comment 1: the work does not convincingly demonstrate technical advantages over recent state-of-the-art methods, and the evaluation lacks comparison with the latest USDA benchmarks. The related work section also does not sufficiently position this approach relative to other clustering-based SFDA methods.
>
> **R:** Comment 1 largely summarizes the concerns raised in Comments 3 and 4. As detailed in our responses to those points, we clarify the following:
>
> 1. The suggested CVPR 2025 paper studies closed-set SFDA, which assumes identical label spaces and is therefore not directly comparable to our category-shift setting. Our original manuscript already included strong baselines for category-shift scenarios, and, importantly, unlike prior methods, our approach performs fine-grained classification to identify individual unknown classes—a capability rarely evaluated in previous benchmarks (see response to Comment 3).
>
> 2. Our paper addresses a more challenging scenario where unknown target classes appear with no prior knowledge of their identities or number. The ability to perform fine-grained discovery of individual unknown classes while adapting to the target domain, something prior methods rarely achieve, constitutes a key technical advantage of our approach (see response to Comment 4).
>
> To further address the reviewer’s concern, we have made additional updates in the revised manuscript:
>
> 1. We report the estimated number of unknown clusters and provide t-SNE visualizations of discovered unknown classes across all tasks, offering clearer evidence of the method’s advantages and practical use cases (see response to Comment 4).
>
> 2. We enhance the Related Work section to more clearly position our method relative to relevant clustering-based approaches and to clarify the unique contributions and distinctions (see response to Requested Change 3).
>
> Together, these clarifications and additions strengthen the empirical evidence and positioning of our work, directly addressing the concerns raised in Weakness 1.
>
> > Comment 2: While the paper provides a principled approach and some empirical evidence, the claims regarding the superiority of the proposed method are not fully supported.
>
> **R:** If “superiority” refers to domain adaptation performance, we would like to clarify that our paper does not claim performance superiority, e.g., in terms of H-score, over all prior methods. Instead, we state that we develop a Bayesian nonparametric clustering approach that learns distinct prototypes for both known and unknown classes without requiring the number of unknown classes a priori. As noted in the Abstract, our experiments demonstrate competitive performance on standard benchmarks while simultaneously enabling effective clustering and discovery of unknown classes. Our claims therefore focus on the methodological capability and practical benefits of the approach rather than asserting absolute performance superiority.

---

> > ### Author Response · Authors · 2026-02-20
> >
> > > Comment 3: The evaluation is limited to comparisons with older baselines (2020–2023), omitting recent state-of-the-art methods such as those reported in "[P1] Revisiting Source-Free Domain Adaptation: Insights into Representativeness, Generalization, and Variety" (CVPR 2025), which achieves higher performance on standard benchmarks (e.g., 74.1% on Office-Home with ResNet-50, compared to 72.9%/69.9% for the proposed method).
> >
> > **R:** We carefully reviewed the paper “Revisiting Source-Free Domain Adaptation: Insights into Representativeness, Generalization, and Variety” suggested by the reviewer. Both this work and the methods evaluated and reported therein focus on closed-set domain adaptation, which assumes identical label spaces in the source and target domains and therefore does not address scenarios with category shift.
> >
> > In contrast, our paper studies a more challenging source-free universal domain adaptation setting, where both source-private and target-private (unknown) classes are present, and the goal explicitly includes discovering these unknown classes. Although the same datasets (e.g., Office-Home) are used, the experimental protocols differ substantially. The suggested paper evaluates the standard closed-set setting with all 65 classes shared across domains, whereas our experiments follow the widely adopted universal DA split (e.g., 10 source-private, 5 shared, and 50 target-private classes on Office-Home), consistent with prior work such as GLC. Therefore, the reported performance numbers are not directly comparable due to the fundamentally different problem formulations. We note that our original manuscript already included strong, widely used baselines designed for category-shift scenarios, such as SHOT and GLC. Importantly, unlike prior methods, our approach does not merely detect unknown-class samples; it also performs fine-grained classification to identify individual unknown classes, a capability rarely evaluated in previous benchmarks.
> >
> > > Comment 4: Furthermore, the paper does not provide compelling experimental scenarios where the proposed technique clearly outperforms prior arts, nor does it address or introduce challenging settings (e.g., Class-Incremental Domain Adaptation, ECCV 2020) where its benefits would be more apparent.
> >
> > **R:** Our paper considers a challenging setting in which unknown target classes appear with no prior knowledge of their identities or number, within a source-free domain adaptation scenario. The goal is to perform fine-grained classification by detecting individual unknown target classes while adapting to the target domain. The absence of both class labels and the number of unknown classes makes this task significantly more difficult than standard SFDA problems. We demonstrate that our method can successfully identify individual target-private classes, a capability that most prior techniques cannot achieve, highlighting both the difficulty of the setting and the practical benefit of our approach.
> >
> > We also carefully reviewed the paper “Class-Incremental Domain Adaptation” (CIDA), as suggested by the reviewer. While CIDA considers detecting target-private classes in a source-free setting, it assumes one labeled (one-shot) sample per target-private class, which requires additional supervision and prior knowledge of unknown classes. This deviates from the fully unsupervised setting addressed by our method and by baselines such as GLC and SHOT.
> >
> > To further demonstrate scenarios where our method provides clear advantages, the revised manuscript now includes: a comparison between the estimated number of unknown clusters and the ground truth (Appendix D.3, Page 29), and t-SNE visualizations of the discovered unknown classes across all tasks and datasets (Appendix D.4, Page 29).
> >
> > These additions highlight a key capability of our method: it can automatically infer the number of unknown classes and discover individual unknown clusters directly from unlabeled data, without any target supervision. This use case, unsupervised estimation of previously unseen individual target classes and their unknown cardinality a posteriori, demonstrates the practical advantage of our method over prior approaches.

---

> > > ### Author Response · Authors · 2026-02-20
> > >
> > > **Requested Changes:**
> > >
> > > > Provide comprehensive experimental comparisons with recent state-of-the-art SFDA methods, especially those reported in "[P1] Revisiting Source-Free Domain Adaptation: Insights into Representativeness, Generalization, and Variety" (CVPR 2025) and other 2024–2025 works. Clearly state where the proposed method stands relative to these.
> > >
> > > **R:** We carefully reviewed the paper “Revisiting Source-Free Domain Adaptation: Insights into Representativeness, Generalization, and Variety” suggested by the reviewer. Both this work and the methods evaluated in it focus on closed-set domain adaptation, which does not address scenarios with category shift between source and target domains. In contrast, the problem considered in our paper is more challenging, as it allows for both source-private classes and target-private (unknown) classes, and our method is specifically designed to discover these unknown classes. Moreover, unlike prior methods, our approach performs fine-grained classification of individual unknown classes, a capability rarely evaluated in previous benchmarks.
> > > > Demonstrate use cases or experimental scenarios (e.g., class-incremental domain adaptation) where the proposed technique offers clear advantages over prior arts, or provide a discussion if such cases do not exist.
> > >
> > > **R:** We carefully reviewed the paper “Class-Incremental Domain Adaptation” (CIDA) suggested by the reviewer. Although CIDA also considers detecting target-private classes in a source-free setting, it assumes one-shot labeled sample per target-private class. Consequently, its use case involves additional supervision and presumes prior knowledge of the unknown classes, which deviates from the fully unsupervised setting addressed by our method and by baselines such as GLC and SHOT.
> > >
> > > To better demonstrate practical scenarios where our approach offers clear advantages, the revised manuscript now includes: a comparison between the estimated number of unknown clusters and the ground truth (Appendix D.3, Page 29), and t-SNE visualizations of the discovered unknown classes across all tasks and datasets (Appendix D.4, Page 29). These additions highlight a key capability of our method: it can automatically infer the number of unknown classes and discover individual unknown clusters directly from unlabeled data, without any target supervision. This use case, unsupervised estimation of unseen previously individual target classes and their unknown cardinality a posteriori, illustrates the practical advantage of our method over prior approaches.
> > >
> > > > Enhance the related work section to explicitly position this approach against other clustering-based SFDA methods, such as "Universal Source-Free Domain Adaptation" and "Domain Consensus Clustering for Universal Domain Adaptation," clarifying the unique contributions and differences.
> > >
> > > **R:** We thank the reviewer for the suggestion. Upon careful review, we note that Universal Source-Free Domain Adaptation does not employ clustering; instead, it relies on an artificially generated negative dataset and a Source Similarity Metric to enhance robustness to domain and category shifts, identifying unknown-class samples given the classifier’s softmax outputs. In contrast, Domain Consensus Clustering for Universal Domain Adaptation is a universal domain adaptation method based on clustering, though not source-free, and is highly relevant to our approach.
> > >
> > > To better position our method, we also incorporate the directly relevant clustering-based method Global and Local Clustering and enhance our related work section to clearly compare our method with both of these approaches (Section 2, Pages 3-4). We carefully clarify the unique contributions and differences: Compared to conventional clustering strategies in SFDA methods such as Domain Consensus Clustering and Global and Local Clustering, BNC offers a clear advantage: it can directly estimate the number of unknown target-domain classes while clustering, eliminating the need for repeated model refitting or manual grid searches over K. This allows the unknown class structure to emerge a posteriori, reduces computational overhead, facilitates seamless integration into SFDA pipelines, and makes the approach more robust to datasets where the number of unknown classes varies widely. Unlike other clustering-based approaches, which primarily use clustering to assist in identifying shared classes, our method leverages BNC to directly discover individual unknown classes, supporting more precise and fine-grained adaptation.

---

> > > > ### Author Response · Authors · 2026-02-20
> > > >
> > > > > Clarify the rationale for reporting both "best epoch" and "final epoch" results, and discuss the stability/convergence of the method.
> > > >
> > > > **R:** We thank the reviewer for the comment. In the original manuscript, we already assessed the stability of our method through extensive sensitivity analyses and discussed its convergence in Section 4.3.4 (Page 14). In the revised manuscript, we provide additional clarification regarding the rationale for reporting both Best Epoch and Final Epoch results (Page 11). In SFDA, model selection is inherently challenging due to the absence of both source and target labels. Many prior works report performance at the Best Epoch, the epoch yielding the highest score based on target labels, but this setting is unrealistic in practice and can lead to over-optimistic estimates of model performance. To address this issue and enable fair comparison with prior literature, we present performance at both the Best Epoch and the Final Epoch. Evaluating at the Final Epoch avoids label leakage and provides a more accurate reflection of performance under real-world deployment conditions. By reporting both metrics, we remain consistent with prior literature while also highlighting the robustness of our method in a practical, label-free scenario.
> > > >
> > > > > Consider moving part of the theoretical contributions (e.g., variational Bayes inference) from the appendix to the main text if they represent a good theoretical contribution.
> > > >
> > > > **R:** We appreciate the reviewer’s suggestion. In the revised manuscript, we have moved the update scheme of our Variational Bayes inference algorithm from the appendix to the main text to better present this contribution (Section 3.2, Pages 7-8).

---

### Review · Reviewer_tY5x · 2026-01-31

**Summary Of Contributions:**

This paper proposes Adapt via Bayesian Nonparametric Clustering (ABC) for source-free domain adaptation under category shift, where unknown target classes are present. Unlike prior SFDA methods that treat all unknown classes as a single category, ABC discovers fine-grained clusters among unknown samples without requiring the number of clusters a priori. The method detects high-confidence known-class samples via a Beta mixture model on cosine similarities, applies a guided Dirichlet Process Mixture clustering using these as anchors, and refines the source model through prototype-based and local consistency regularizers. Experiments on Office-31, Office-Home, and VisDA-C show competitive H-scores against SFDA and non-source-free baselines, with additional NMI results demonstrating effective fine-grained clustering of unknowns.

Key Strengths:
1. The paper addresses an underexplored yet practical aspect of SFDA—fine-grained discovery of unknown classes rather than collapsing them into a single category. Integrating Bayesian nonparametric clustering into the SFDA pipeline, guided by high-confidence known-class anchors, is technically sound and represents a meaningful contribution to the field.
2. The evaluation covers three standard benchmarks under OPDA settings, reports both classification (H-score, mean accuracy) and clustering (NMI) metrics, and provides extensive sensitivity analyses across multiple hyperparameters (δ, α, K_nn, λ_loc), supporting robustness claims.
3. The DPM-based clustering, prototype learning, and neighborhood consistency are combined in a unified iterative framework, reducing error propagation between clustering and adaptation compared to post-hoc clustering approaches applied to existing methods.

Key Weaknesses:
1. he paper does not report the estimated number of unknown clusters versus ground truth, nor does it provide ARI or purity metrics specifically on the unknown subset, making it difficult to assess over- or under-clustering behavior. Runtime and memory comparisons against baselines are also absent.
2. The anchor-guided mechanism lacks analysis of failure scenarios when high-confidence labels are incorrect or when the number of discovered common classes diverges from source classes. The remapping procedure between cluster indices and known-class labels is not fully detailed.
3. The Gaussian likelihood with Normal-Wishart prior may not be well-suited for L2-normalized embeddings lying on a hypersphere, and robustness to non-Gaussian cluster structures in the feature space is not empirically examined.

**Audience:**

Yes

**Audience Explanation:**

Fine-grained discovery of unknown classes in source-free domain adaptation is a practical and underexplored problem relevant to researchers in transfer learning, domain adaptation, and Bayesian nonparametrics.

**Claims And Evidence:**

Yes

**Claims Explanation:**

The main claims regarding competitive H-score performance and effective fine-grained clustering are supported by experiments on three standard benchmarks with appropriate metrics and sensitivity analyses. However, the evidence would be more convincing with runtime comparisons, variance reporting across seeds, and diagnostics on the estimated number of unknown clusters versus ground truth.

**Requested Changes:**

Report the estimated number of unknown clusters versus ground truth along with ARI/purity on the unknown subset, and include runtime/memory comparisons against baselines. Strengthening: Add variance reporting across multiple seeds, ablate the Beta-mixture detector against alternative OOD criteria, and clarify the failure modes and remapping procedure for the anchor-guided mechanism.

---

> ### Author Response · Authors · 2026-02-20
>
> We greatly appreciate the reviewer’s detailed and constructive suggestions. We have carefully revised the manuscript to incorporate the requested analyses and have also addressed the noted weaknesses and concerns. The main changes in this revised version are highlighted in blue.
>
> > Weakness 1: Estimated number of unknown clusters versus ground truth, nor does it provide ARI or purity metrics specifically on the unknown subset, making it difficult to assess over- or under-clustering behavior. Runtime and memory comparisons against baselines are also absent.
>
> **R:** We thank the reviewer for highlighting these important aspects. In the revised manuscript, we now report the estimated number of unknown clusters compared to the ground truth (Appendix D.3, Page 29). and provide clustering quality metrics specifically on the unknown subset (Appendix D.1, Page 25) Based on these results, we discuss the over- and under-clustering behavior of the proposed method in detail (Appendix D.1, Page 26). Additionally, we include runtime and memory comparisons against baseline methods (Appendix D.2, Page 28), along with a thorough discussion of the computational complexity of our approach  (Appendix D.2, Page 26),.
>
> > Weakness 2: The anchor-guided mechanism lacks analysis of failure scenarios when high-confidence labels are incorrect or when the number of discovered common classes diverges from source classes. The remapping procedure between cluster indices and known-class labels is not fully detailed.
>
> **R:**  We thank the reviewer for highlighting these issues.
>
> Failure modes. We now explicitly analyze potential failure scenarios of the anchor-guided mechanism in the revision (Section 4.3.1, Page 12). Two situations may occur when high-confidence labels are incorrect: 1.  Misassigned known samples. When samples from two similar source classes are confused, the GNBC step may propagate the incorrect label, biasing the estimation of the corresponding class. 2.          Unknown samples treated as known. If a large number of unknown samples are selected as high-confidence known anchors, some unknown classes may not be discovered.
>
> Empirically, the second case is the dominant one. Based on manual inspection, the first case is rare (≈0% in most tasks on Office-31 and <5% on average on Office-Home). Accordingly, we added an ablation study in which the Beta-mixture detector is replaced with alternative Out-of-Distribution (OOD) detectors, along with an expanded discussion of these two failure modes in the revised manuscript. The results show that weaker detectors substantially degrade performance and, in some tasks, fail to identify any unknown classes, illustrating both the failure scenario and the importance of the chosen Beta-mixture model.
>
> Remapping procedure. We have added a dedicated paragraph in the revised manuscript detailing the mapping between cluster indices and known-class labels, including the matching criterion and implementation details (see the end of Appendix B, Page 24).
>
> > Comment 3: The Gaussian likelihood with Normal-Wishart prior may not be well-suited for L2-normalized embeddings lying on a hypersphere, and robustness to non-Gaussian cluster structures in the feature space is not empirically examined.
>
> **R:** We thank the reviewer for highlighting the concern regarding the use of a Gaussian likelihood for L2-normalized embeddings on a hypersphere. We would like to clarify that L2 normalization is not applied to the embeddings used in the GBNC clustering step, which adopts the Infinite Gaussian Mixture Model assumption. The normalization mentioned in the manuscript is used within the Prototype-based Regularization module during training, where embeddings and class centroids are normalized to compute inner products equivalent to cosine similarity and to remove the influence of vector norms, following standard practice in metric learning. Consequently, the embeddings used for Gaussian mixture modeling are not constrained to a fixed-norm manifold and remain in the Euclidean space, where the Gaussian assumption is appropriate.
>
> Regarding robustness to non-Gaussian cluster structures, we agree that deep feature distributions may deviate from ideal Normality. Our use of the Gaussian mixture is primarily for partitioning rather than exact density modeling, and Gaussian mixture models are known to approximate a wide range of distributions in practice. The consistent performance across datasets in our experiments suggests that the method is not overly sensitive to moderate deviations from the assumption. We also note that Gaussian mixtures are widely used for clustering deep embeddings across diverse tasks and architectures, demonstrating stable empirical performance despite non-Gaussian feature geometries [1–3]. We have clarified this discussion in the revised manuscript (Section 3.2 on Page 7).

---

> > ### Author Response · Authors · 2026-02-20
> >
> > **Requested Changes:**
> >
> > **R:** We appreciate the reviewer’s valuable suggestions.
> >
> > **Report the estimated number of unknown clusters versus ground truth:** We now report the estimated number of unknown clusters compared to the ground truth, along with detailed analysis and discussion in both the main text (Page 12) and Appendix D.3 (Page 29).
> >
> > **Report ARI or purity on the unknown subset:** To address the reviewer’s suggestion, we report Purity on the unknown subsets, along with NMI for consistency with the clustering metrics used in our main experiments and provide detailed analysis in Appendix D.1 (Page 25) discussing the over- and under-clustering behavior of the proposed method.
> >
> > **Include runtime/memory comparisons against baselines:** We provide runtime and memory comparisons with baseline methods, along with a thorough discussion of the computational complexity of our approach in Appendix D.2 (Page 26).
> >
> > **Add variance reporting across multiple seeds:** For all main experiments and the newly added unknown-subset experiments, we report performance metrics including both the mean and standard deviation across three random runs, for our method as well as the reproduced baselines (Page 10).
> >
> > **Ablate the Beta-mixture detector against alternative OOD criteria, and clarify the failure modes:** We added an ablation study replacing the Beta-mixture detector with alternative Out-of-Distribution (OOD) detectors, together with an expanded discussion of the two failure modes in Section 4.3.1 (Page 12).
> >
> > **Remapping procedure for the anchor-guided mechanism:** We include a dedicated paragraph in Appendix B (Page 24) detailing the mapping between cluster indices and known-class labels, including the matching criterion and implementation steps.
> >
> > References:
> >
> > [1] An unsupervised deep learning framework via integrated optimization of representation learning and GMM-based modeling. Asian Conference on Computer Vision  (2018).
> >
> > [2] DNB: A joint learning framework for deep Bayesian nonparametric clustering. IEEE Transactions on Neural Networks and Learning Systems (2021).
> >
> > [3] Deepdpm: Deep clustering with an unknown number of clusters. IEEE/CVF Conference on Computer Vision and Pattern Recognition (2022).

---

### Decision · Action_Editor_siq8 · 2026-03-11

**Recommendation:** Accept as is

**Audience:**

Yes

**Audience Explanation:**

All reviewers agree that the submission meets the bar in terms of interest to at least part of the TMLR audience.

* "The paper addresses an underexplored yet practical aspect of SFDA [... and] represents a meaningful contribution to the field." (tY5x)
* "The problem of unknown class discovery in source-free domain adaptation is of practical and theoretical interest to the TMLR audience." (B4mX)
* "The paper addresses the highly relevant and practical problem of Source-Free Domain Adaptation, which is crucial for privacy-preserving machine learning. [...] Researchers in domain adaptation, transfer learning, and Bayesian nonparametrics would find these results significant." (wERe)

**Claims And Evidence:**

Yes

**Claims Explanation:**

All reviewers agree that the submission meets the bar in terms of claims and evidence.

* "[The proposed approach] is technically sound" and "the evaluation [... supports] robustness claims." (tY5x)
* "[The] authors did an excellent job responding to my request to better demonstrate the practical advantages of their technique. The newly added t-SNE visualizations and the quantitative analysis comparing the estimated versus ground truth number of unknown clusters significantly strengthen the paper's empirical evidence. These additions effectively illustrate the method's capability for the unsupervised estimation of previously unseen classes." (B4mX)
* "The authors provide extensive experimental evidence [... and the] inclusion of ablation studies and hyperparameter sensitivity analyses demonstrates that the performance gains are due to the proposed components and are robust to parameter settings."

Concerns raised by the reviewers were addressed to their satisfaction by the authors' response.